# AGENTMATH: EMPOWERING MATHEMATICAL REASONING FOR LARGE LANGUAGE MODELS VIA TOOL-AUGMENTED AGENT

**Haipeng Luo**[1,3]   **Huawen Feng**[2]   **Qingfeng Sun**[2]   **Can Xu**[2†]   **Kai Zheng**[2]

**Yufei Wang**[2]   **Tao Yang**[2]   **Han Hu**[2]   **Yansong Tang**[1†]

[1]Shenzhen International Graduate School, Tsinghua University
[2]Tencent Hunyuan    [3]Pengcheng Laboratory
{luohp24@mails., tang.yansong@sz.}tsinghua.edu.cn
{bazzfeng,victorqsun,leocaxu,kaivenzhang,garyyfwang}@tencent.com
{rigorosyang,winstony}@tencent.com

## ABSTRACT

Large Reasoning Models (LRMs) like o3 and DeepSeek-R1 have achieved remarkable progress in natural language reasoning with long chain-of-thought. However, they remain computationally inefficient and struggle with accuracy when solving problems requiring complex mathematical operations. In this work, we present AgentMath, an agent framework that seamlessly integrates language models' reasoning capabilities with code interpreters' computational precision to efficiently tackle complex mathematical problems. Our approach introduces three key innovations: (1) An automated method that converts natural language chain-of-thought into structured tool-augmented trajectories, generating high-quality supervised fine-tuning (SFT) data to alleviate data scarcity; (2) A novel agentic reinforcement learning (RL) paradigm that dynamically interleaves natural language generation with real-time code execution. This enables models to autonomously learn optimal tool-use strategies through multi-round interactive feedback, while fostering emergent capabilities in code refinement and error correction; (3) An efficient training system incorporating innovative techniques, including request-level asynchronous rollout scheduling, agentic partial rollout, and prefix-aware weighted load balancing, achieving 4-5× speedup and making efficient RL training feasible on ultra-long sequences with scenarios with massive tool calls. Extensive evaluations show that AgentMath achieves state-of-the-art performance on challenging mathematical competition benchmarks including AIME24, AIME25, and HMMT25, substantially outperforming frontier open-source models of comparable size. Specifically, AgentMath-30B-A3B attains 90.6%, 86.4%, and 73.8% accuracy respectively, surpassing OpenAI-o3-mini and Claude-Opus-4.0-Thinking while remaining competitive with OpenAI-o3, Gemini-2.5-Pro, and DeepSeek-R1-671B-0528. These results validate the effectiveness of our approach and pave the way for building scalable mathematical reasoning agents.

## 1 INTRODUCTION

Large Reasoning Models (LRMs) such as o3 and DeepSeek-R1 have made remarkable progress in natural language reasoning with long chain-of-thought (CoT)(OpenAI et al., 2024; Team et al., 2025a; DeepSeek-AI et al., 2025; xAI, 2023; Claude, 2025; Team et al., 2023; Wei et al., 2022). However, when tackling mathematical problems that demand precise computation or intricate symbolic manipulation, including large-number arithmetic, complex equation solving, and geometric reasoning, pure text-based reasoning still has limitations: frequent computational errors necessitate redundant corrections, which in turn leads to inefficiency and erroneous results.

[†]   Corresponding authors. This work was conducted during Luo's internship at Tencent and was supported by the CIE-Tencent Ph.D. Student Research Incentive Program (Tencent Hunyuan Special Fund).

To enhance computational efficiency and accuracy, recent work has explored incorporating external tools (i.e., code interpreters), delegating complex and error-prone computational steps to external environments(Li et al., 2025e; Zhou et al., 2025a; Lin & Xu, 2025; Zhang et al., 2025b; Chen et al., 2023; Gao et al., 2023; Gou et al., 2023b). For instance, models like o3 and o4-mini have significantly improved mathematical reasoning accuracy through tool invocation. Nevertheless, existing approaches still face three critical challenges. First, high-quality tool-use data remains extremely scarce. While methods like START(Li et al., 2025b) generate tool-augmented trajectories via prompt engineering, they suffer from delayed code computation and code result distrust; CoRT(Li et al., 2025a) employs manual annotation which is effective but lacks scalability; under supervised learning, models struggle to learn autonomous debugging from code execution failures. Second, the potential for continuous performance improvement and tool-use strategy optimization through agentic RL remains unexplored. Third, competition-level mathematical problems typically involve ultra-long reasoning chains with extensive tool invocations (i.e., 96k tokens, 96 tool calls), making traditional batch-synchronous RL training frameworks inadequate for large-scale agent learning, while the rollout time for ultra-long sequences causes severe long-tail effects.

To address these challenges, we propose AgentMath, a tool-augmented agentic framework that seamlessly integrates model reasoning with code execution for efficient and reliable mathematical problem-solving. AgentMath comprises three core components: First, we propose an automated tool-augmented trajectory synthesis method that transforms pure-text long chain-of-thought data into structured training samples containing code execution and authentic feedback. Through code injection, execution verification, and multi-dimensional refinement, this effectively alleviates data scarcity. Second, we design a novel agentic reinforcement learning paradigm that supports dynamic interleaving of natural language generation and code execution during reasoning. Through multi-turn interactive feedback, models autonomously learn optimal tool invocation strategies. Experiments reveal that model accuracy continuously improves with increasing tool invocations, exhibiting emergent code self-correction capabilities. Third, to support large-scale agentic RL training(Schulman et al., 2017; Li et al., 2025e; Feng et al., 2025b; Mai et al., 2025), we develop an efficient training system incorporating key techniques such as request-level asynchronous rollout scheduling, agentic partial rollout, and prefix-aware weighted load balancing. These innovations improve training efficiency by $4$–$5\times$, effectively supporting reinforcement learning in scenarios with ultra-long sequences and extensive tool invocations.

Experimental results demonstrate that AgentMath achieves state-of-the-art performance on challenging mathematical competition benchmarks including AIME24, AIME25, and HMMT25(Balunović et al., 2025), outperforming frontier open-source tool-augmented models and pure-text reasoning models of comparable scale. Specifically, AgentMath-30B-A3B achieves accuracies of 90.6%, 86.4%, and 73.8% respectively, surpassing OpenAI-o3-mini and Claude-Opus-4.0-Thinking, while remaining competitive with Gemini-2.5-Pro and DeepSeek-R1-0528.

Our main contributions include: (1) We propose an efficient automated tool-augmented data synthesis pipeline that effectively alleviates data scarcity issues. (2) We design a novel agentic reinforcement learning paradigm achieving dynamic integration of natural language reasoning and code execution, enabling models to autonomously learn tool-use strategies through multi-turn interactive feedback. (3) We develop an efficient asynchronous training system that provides a scalable solution for ultra-long sequences, multi-turn interaction agent reinforcement learning. (4) We achieve state-of-the-art performance on multiple challenging mathematical competition benchmarks, paving the way for building more complex and scalable mathematical reasoning agents.

## 2 METHOD

### 2.1 OVERVIEW

This section presents **AgentMath**, a tool-augmented agent framework designed to enhance complex mathematical reasoning by tightly integrating the emergent reasoning capabilities of Large Language Models (LLMs) with the precise arithmetic and symbolic computation facilitated by an external code execution environment. The architecture operates in two stages: (i) supervised fine-tuning (SFT) on curated, synthetic tool-invocation trajectories to establish initial competence in invoking tools appropriately, and (ii) large-scale reinforcement learning (RL) driven by outcome feedback to incentivize exploration and mastery of optimal, self-corrective tool-use strategies.

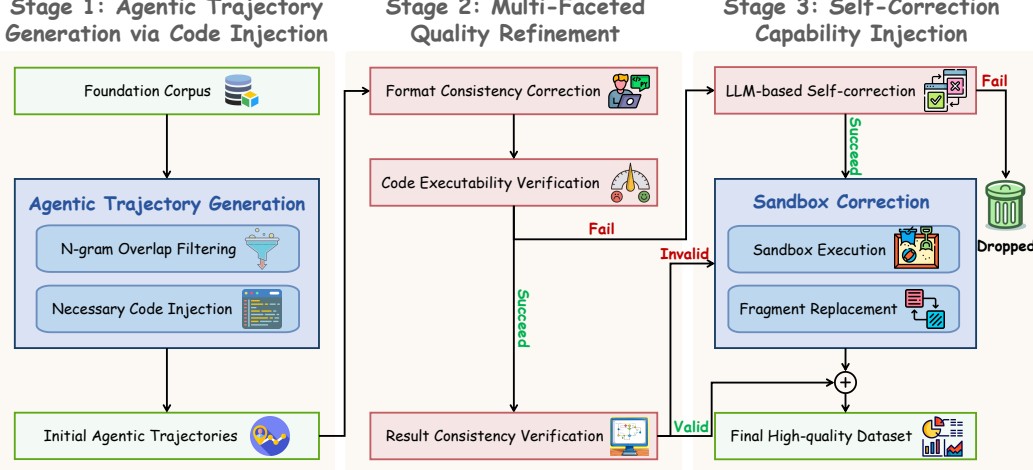

Figure 1: This diagram outlines a three-stage pipeline for creating a high-quality tool-augmented trajectories for training agents, including Agentic Trajectory Generation via Code Injection, Multi-Faceted Quality Refinement and Self-Correction Capability Injection. This automated process transforms pure-text reasoning into verified, executable agentic trajectories.

**Problem Formulation and Interaction Protocol**: We formulate tool-augmented mathematical reasoning as a Markov Decision Process (MDP). The LLM-based policy generates interleaved reasoning segments and executable code blocks through interaction with a sandboxed execution environment. Each trajectory consists of action-observation pairs, where state transitions result from the policy's conditional generation and deterministic code execution. We define a structured markup protocol for agent-environment communication: <think> denotes natural language reasoning,  delimits executable code blocks, and <interpreter> encapsulates execution feedback. This bidirectional exchange mechanism incorporates execution results into the generation context, enabling adaptive strategy refinement and planning. See Appendix A.2 for more details.

## 2.2 TOOL-DRIVEN DATA SYNTHESIS

The scarcity of high-quality training data that captures both complex reasoning patterns and strategic tool utilization remains a fundamental bottleneck in developing code-enabled agents. This work introduces a three-stage automated synthesis-and-refinement pipeline that transforms pure-text long CoT into agent-style demonstrations with executable code invocations and authentic interpreter feedback, yielding a compact and efficient instruction dataset for SFT, as shown in Figure 1.

**Stage 1: Agentic Trajectory Generation via Code Injection.** We assemble a large-scale corpus from public mathematical reasoning sources (i.e., AM-Thinking, Open-Thoughts(Ji et al., 2025; Guha et al., 2025)), which distill responses from DeepSeek-R1-0528. To prevent evaluation contamination, we apply n-gram overlap filtering against benchmark datasets (i.e, AIME24/25, HMMT25), yielding a high-quality pure-text reasoning dataset $\mathcal{D}_{\text{text}}$.

Direct Manual annotation of agent trajectories is both costly and susceptible to noise. To address this, we propose an efficient code-injection strategy that leverages a powerful teacher model (i.e., DeepSeek-V3(DeepSeek-AI et al., 2025)), guided by carefully crafted prompts presented in Appendix A.6.2. Each extensive Chain-of-Thought (CoT) sequence $\tau_{\text{text}} \in \mathcal{D}_{\text{text}}$ is systematically partitioned into multiple segments. Subsequently, each segment undergoes transformation via the injection function $\mathcal{F}_{\text{inject}}$, which substitutes computationally intensive reasoning steps $s_{\text{calc}}$ with executable code blocks and their corresponding execution outputs:

$$\tau'_{\text{agent}} = \mathcal{F}_{\text{inject}}(\tau_{\text{text}}), \quad \text{where } \mathcal{F}_{\text{inject}} : \tau_{\text{text}} \mapsto \left(\tau_{\text{text}} \text{ with } s_{\text{calc}} \Rightarrow (c, o_{\text{sim}})\right).$$

where $c$ denotes the injected code segment, $s_{\text{calc}}$ represents the replaced computational step, and $o_{\text{sim}}$ is the teacher-simulated execution result. This injection targets complex operations (exponential computations, matrix manipulations, equation solving) while preserving elementary calculations in

textual form to maintain the model's understanding of tool invocation rationale and prevent over-dependence. Code blocks are delimited by   tags, with execution results enclosed in <interpreter> </interpreter> tags referring to (Feng et al., 2025b).

**Stage 2: Multi-Faceted Quality Refinement.** Automatically synthesized trajectories can contain formatting issues, code defects, and logical inconsistencies. We apply four complementary procedures to ensure high quality and effectiveness:

*(i) Format consistency correction:* We employ regular-expression normalization and teacher model regeneration for complex cases to enforce strict adherence to the –<interpreter> structural compliance.

*(ii) Code executability verification:* Each embedded code snippet is executed within a controlled sandbox environment. For any failures, we initiate a bounded resampling loop to generate equivalent but executable alternatives. If execution remains unsuccessful within a predefined compute budget, the block is reverted to its original textual step $s_{\text{calc}}$ to preserve logical soundness.

*(iii) Environmental feedback alignment:* Simulated outputs $o_{\text{sim}}$ from the teacher are systematically replaced with ground-truth execution results $o_{\text{real}} = \mathcal{E}(c)$, where $\mathcal{E}$ denotes the interpreter environment. A dedicated verifier model (i.e., Qwen3-32B) is employed to perform this judgment, guided by a specific judge-prompt detailed in Appendix A.6.3, then assesses contextual consistency. Incoherent samples are removed or downgraded to text-only variants to maintain narrative integrity.

*(iv) Tool-usage rationality assessment:* Heuristic constraints on code complexity metrics (i.e., line count, abstract syntax tree depth) are enforced to eliminate instances of unnecessary code invocation, thereby reinforcing necessity-aware tool utilization patterns.

**Self-Correction Capability Injection.** Beyond correct tool invocation, a robust agent need also recover from erroneous tool feedback. We sample trajectories that were excluded during refinement due to execution failures, and for each failed program $c_{\text{fail}}$ with error output $o_{\text{error}} = \mathcal{E}(c_{\text{fail}})$, we prompt the teacher model to generate a structured self-correction trace (diagnose the error → repair the code → re-execute → continue reasoning). The detailed prompt can be found in Appendix A.6.1. A small fraction of these negative-to-positive corrections is injected to strengthen debugging robustness. The final instruction set $\mathcal{D}_{\text{SFT}}$ combines validated, tool-augmented trajectories with diagnostic correction traces and serves as the foundation for SFT.

## 2.3 AGENTIC REINFORCEMENT LEARNING

We present an agentic reinforcement learning (RL) framework that advances code-integrated reasoning capabilities beyond supervised fine-tuning (SFT). This stage pursues two objectives: (i) to quantify the incremental gains of RL over an SFT baseline, and (ii) to elucidate how RL reshapes tool-usage strategies under interleaved natural language generation and program execution. Additionally, the detailed construction of the RL data is described in Appendix A.5.2.

### 2.3.1 AGENT-SPECIFIC REINFORCEMENT LEARNING

Our framework employs Group Relative Policy Optimization (GRPO)(Shao et al., 2024) as the core optimization algorithm, which obviates critic models while enhancing training efficiency through group-wise trajectory sampling and reward normalization described in Appendix A.3. We employed multi-stage RL training, as detailed in Section 3.5. Following DAPO (Yu et al., 2025), we incorporate dynamic sampling, asymmetric gradient clipping, token-level loss computation, and KL divergence removal. We introduce three system innovations tailored for code-integrated agents:

**Agentic trajectories with interleaved code execution.** During rollout, trajectories are constructed through a *generate–pause–execute–resume* loop (See Appendix A.2), yielding hybrid traces composed of chain-of-thought reasoning, inline code snippets, and real-time interpreter feedback. Tool invocations are bounded by a per-instance cap $T$, enabling fine-grained control over agent-environment interactions and promoting sample efficiency.

**Loss Masking for Policy Gradient Updates.** To focus learning on the agent's decision-making process, the advantage signal is applied exclusively to tokens within <think> and  segments. Tokens generated by the environment, specifically within <interpreter>, are masked during

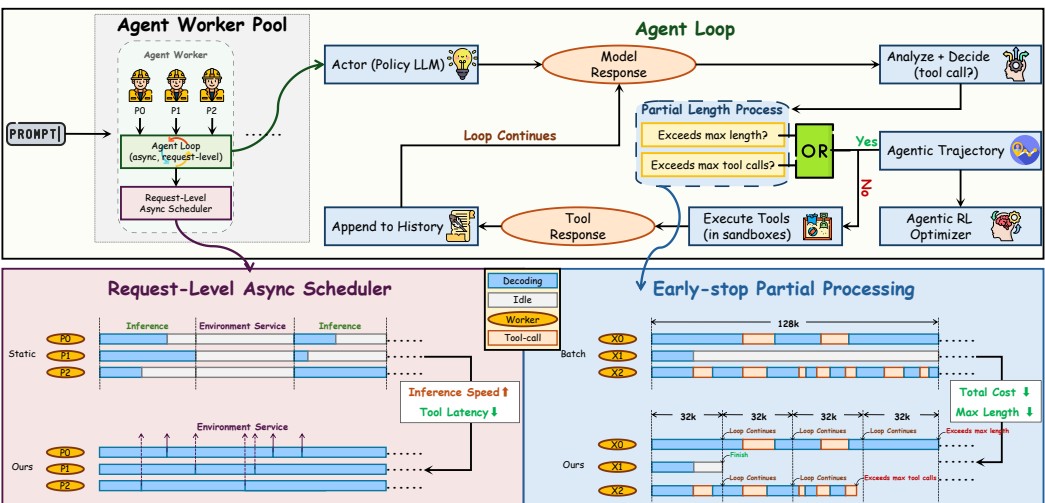

Figure 2: The diagram of agentic reinforcement learning. It depicts the structure and workflow of our agentic reinforcement learning system with core functions including Agent Loop, Asynchronous Scheduler, and Partial Rollout, along with key performance improvement. Based on the Asynchronous Scheduler, the Agent Loop continues running by default. It will stop early only when conditions are met: either the content length exceeds the max length (i.e., 32k) or the number of tool calls exceeds the maximum constraint.

optimization, ensuring that gradient updates are driven by the agent's own actions rather than deterministic environmental responses.

**Adaptive Batch Construction with Filtering and Backfilling.** For each problem instance, we sample $G$ trajectories. Batches are filtered to exclude problems where all trajectories yield either uniformly correct or uniformly incorrect answers, which offer limited learning signal. To maintain consistent batch sizes, we backfill by randomly sampling additional filtered instances from the same pool, thus avoiding inefficient resampling loops while preserving distributional diversity.

### 2.3.2 REWARD DESIGN

Our reward function integrates answer correctness with tool-usage efficiency. The accuracy component $R_{\text{acc}}$ provides binary feedback based on mathematical equivalence, validated via the math_verify library:

$$R_{\text{acc}} = \begin{cases} 1, & \text{if is\_equivalent}(\hat{a}, a), \\ 0, & \text{otherwise}, \end{cases}$$

where $\hat{a}$ denotes the predicted answer and $a$ represents ground truth. Conditioned on correctness, the tool-usage reward $R_{\text{tool}}$ incentivizes efficient computational resource utilization:

$$R_{\text{tool}} = \min\big(R_{\max}, \alpha + \beta \cdot N_{\text{code}}\big) \quad \text{if } N_{\text{code}} > 0,$$

where $\alpha$ represents the base tool-usage reward, $\beta$ scales with invocation count, and $R_{\max}$ caps the maximum tool-usage reward. The composite reward function becomes:

$$R_{\text{total}} = R_{\text{acc}} + \mathbb{I}(R_{\text{acc}} = 1) \cdot R_{\text{tool}}.$$

### 2.4 SCALABLE AGENTIC RL INFRASTRUCTURE

Agentic Reinforcement Learning for complex mathematical reasoning poses significant infrastructural challenges. Empirical analysis reveals that, during RL training with the temperature set to 1.0, complex problems yield trajectories averaging 24k tokens and involve approximately 27 tool invocations. This combination of long-context generation and high-frequency external interactions produces heterogeneous computational workloads. Traditional synchronous batch rollouts exhibit substantial inefficiencies due to synchronization overhead and resource underutilization.

To address these challenges, we design a high-performance, scalable training system tailored to Agentic RL. Through an asynchronous decoupled architecture, an Agentic Partial Rollout algorithm, and prefix-aware load balancing, the system mitigates performance bottlenecks induced by long-tail effects and concurrent tool invocations, achieving $4 \sim 5\times$ improvement in end-to-end training throughput, as shown in Figure 2.

### 2.4.1 DECOUPLED AND ASYNCHRONOUS SYSTEM ARCHITECTURE

The architecture is founded on the principle of decoupling GPU-intensive model inference from CPU/IO-intensive agent logic and environment interactions.

**Distributed code execution sandbox cluster.** We deploy a distributed cluster of isolated worker pods to serve concurrent tool invocations at scale. This design offloads CPU-bound code execution from the training loop while enabling dynamic load distribution. Parallelization reduces tool-call latency from 175 s to 1.2 s and removes inference blocking, substantially improving GPU utilization.

**Request-Level Asynchronous Rollout Scheduling.** Static batch-synchronous processing is replaced with a coroutine-driven, request-level asynchronous scheduler. Each trajectory rollout is treated as an independent long-running request, with the inference engine (server) and agents (clients) fully decoupled via asynchronous communication. When requests suspend for tool invocations, the inference engine immediately processes other ready requests. This fine-grained scheduling eliminates head-of-line blocking and maximizes GPU parallelism across heterogeneous workloads.

### 2.4.2 AGENTIC PARTIAL ROLLOUT

Agentic RL suffers long-tail latency from both sequence length and tool-invocation counts. We introduce an Agentic Partial Rollout mechanism that decomposes each trajectory $\tau$ into budget-limited segments:

$$\tau = \tau^{(1)} \oplus \tau^{(2)} \oplus \ldots \oplus \tau^{(N)},$$

where $\oplus$ denotes sequence concatenation. Each segment is constrained by a maximum generation length $L_{\text{seg}}$ and a maximum number of tool invocations $T_{\text{seg}}$.

At each training iteration, the scheduler samples from an unfinished pool $\mathcal{U}$ and a set of new tasks $\mathcal{P}$, generating one segment per task. Segment generation terminates when: (i) an EOS token is produced; (ii) segment length reaches $L_{\text{seg}}$; (iii) tool invocations reach $T_{\text{seg}}$; or (iv) cumulative trajectory metrics reach global limits $L_{\text{global}}$ or $T_{\text{global}}$. This segmentation prevents individual trajectories from monopolizing resources and smooths computational load, yielding a 2.2–2.5x speedup. Algorithm 1 outlines the procedure.

### 2.4.3 PREFIX-AWARE WEIGHTED LOAD BALANCING

Partial rollouts alleviate long-tail latency but introduce requests with long prefixes, increasing KV-cache memory and prefill cost. Therefore, We design a Prefix-Aware Weighted Load Balancing strategy that assigns dynamic weights based on prefix length and routes requests to the least-loaded inference engine.

Each request $R_j$ with prefix length $L_j$ receives a weight

$$w_j = \left\lfloor \frac{L_j}{L_{\text{base}}} \right\rfloor + w_{\text{base}},$$

where $L_{\text{base}}$ (i.e., 16k tokens) normalizes length and $w_{\text{base}}$ quantifies prefill overhead. For $M$ engines $S_1, \ldots, S_M$ with loads $W_k$, a new request $R_j$ is routed to

$$k^* = \arg \min_{k \in \{1,\ldots,M\}} W_k, \quad \text{and} \quad W_{k^*} \leftarrow W_{k^*} + w_j.$$

To maximize KV-cache reuse, we implement sticky sessions via a LRU(Least-Recently-Used) caching, ensuring consecutive segments from the same trajectory preferentially route to the same engine, thereby avoiding redundant context transfer and recomputation. This combination of dynamic weighting and cache-affinity scheduling maintains load balance under heterogeneous traffic patterns while maximizing system throughput.

## 3 EXPERIMENTS

### 3.1 SUPERVISED FINE-TUNING (SFT) DATA CONSTRUCTION

**Stage 1: Foundational Data Curation and Filtering.** We aggregate a raw corpus from multiple public math reasoning datasets (i.e., AM-Thinking, OpenThoughts, and AceReason). After problem-level deduplication, we apply N-gram ($N = 4$) and MinHash LSH algorithms to eliminate overlaps with all evaluation sets, including AIME24, AIME25 and HMMT25 with 0.6 similarity threshold. To further prevent data leakage, we compute semantic similarities between training data and evaluation sets using gte-large model(Zhang et al., 2024a) , filtering out the top-5 most similar samples. We then annotate each problem with difficulty scores from 0 to 10 using Qwen3-30B, retaining data with scores above 5, which yields 392k samples. Finally, using DeepSeek-R1-0528 to generate solutions and removing instances with incorrect answers, we obtain 346k data.

**Stage 2: Tool-Augmented Data Synthesis.** We first decompose the problem-solving process into discrete reasoning segments, each with a fixed length of 3k tokens and perform tool-augmented synthesis for each segment using the Prompt presented in Appendix A.6.2 via DeepSeek-V3-0324. During synthesis, we filter out samples with synthesis format error, tool execution failures, or incorrect final answer, yielding 302k high-fidelity, tool-augmented solution trajectories.

**Stage 3: Self-Correction Data Generation.** To incorporate self-correction mechanisms, we sample 30k instances from trajectories with unsuccessful code execution and leverage the Self-correction Prompt presented in Appendix A.6.1 to guide DeepSeek-V3-0324 in generating correction processes, producing 14k valid self-correction trajectories.

Through this comprehensive pipeline, we construct a 316k tool-augmented synthetic training set with an average of 8.3 tool calls and an average sequence length of 16.9K tokens per sample.

### 3.2 REINFORCEMENT LEARNING (RL) DATA CONSTRUCTION

For RL, we collect problems from multiple public high-quality RL datasets (i.e., DeepScaler, Skywork-OR1, Retool, POLARIS). We apply the same deduplication strategy as for SFT data, ensuring no overlap with evaluation sets through N-gram (N=4), MinHash LSH, and semantic similarity computation with 0.6 similarity threshold. To identify challenging problems, we use AgentMath-8B-SFT to perform 8 inference attempts on all data, filtering out problems that are solved correctly 8 times, culminating in a final set of 42k high-difficulty RL training data. This focuses training on hard instances that push the model's strategic capabilities and maximize potential gains from RL.

### 3.3 TRAINING SETTINGS

**Base Models.** The experiments utilize four base models from the Qwen3 series: Qwen3-1.7B-Base and Qwen3-8B-Base are pre-trained models without post-training; Qwen3-30B-A3B-Instruct-2507 (Non-Thinking mode, 30B total parameters, 3B activated) and Qwen3-235B-A22B-Instruct-2507 (Non-Thinking mode, 235B total parameters, 22B activated) are instruction-tuned Mixture-of-Experts (MoE) models without long chain-of-thought training.

**SFT Training.** We employ the Llama-Factory framework, training for 6 epochs with learning rates of $6 \times 10^{-5}$ (1.7B, 8B, A3B models) and $2 \times 10^{-5}$ (A22B model), using cosine decay with 10% warmup, batch size of 512, and maximum sequence length of 32k tokens.

**RL Training.** RL training is built on the verl 0.5.0.dev0 framework(Sheng et al., 2024), initializing from the best SFT checkpoint and using VLLM(Kwon et al., 2023) as the inference engine, with a 128-node Sandbox cluster for large-scale code execution. We use a constant learning rate of $1 \times 10^{-6}$, a batch size of 64, and a temperature of 1.0, performing 8 rollouts per problem. Training progresses through three stages, dynamically adjusted to maintain length truncation and tool-call excess rates below 10%: maximum response length increases from 48k to 72k to 96k tokens, with corresponding tool invocation limits of 48, 72, and 96 calls, and partial rollout counts of 2, 3, and 4, ensuring each segment rollout remains within 24k tokens and 24 tool invocations. For reward design, we set $\alpha = 0.1$ and $\beta = 0.01$, and constrain the maximum tool reward to $R_{max} = 1$ (For details refer to Appendix A.16). Due to computational constraints, AgentMath-235B-A22B is trained solely via SFT. More details for data synthesis, SFT, RL, and evaluation are provided in Appendix A.5

## 3.4 MAIN RESULTS

In this section, we comprehensively evaluate AgentMath by comparing it against the advanced reasoning models on three challenging math competition benchmarks: AIME24, AIME25, and HMMT25. Results are presented in Table 1 with more extensive model comparisons available in Appendix Table 4. To ensure robust evaluation, we perform 32 independent inference runs per test sample, using avg@32 as the pass@1 metric. We use a consistent configuration: 96K maximum sequence length, 96 maximum tool calls, and code interpreter output is limited 1024 tokens, and we set 0.6 temperature, and 0.95 top-p.

The results show that AgentMath significantly outperforms existing tool-augmented and text-only frontier reasoning models across all three benchmarks at comparable parameter scales. Among small-scale models (1B ~ 2B), AgentMath-1.7B attains 59.6%, 48.1%, and 40.2% accuracy on AIME24, AIME25, and HMMT25 respectively, substantially surpassing both the tool-augmented CoRT-1.5B (43.1%, 30.2%, 20.1%) and the text-only OpenReasoning-1.5B (55.5%, 45.6%, 31.5%). At the medium scale (7B ~ 8B), AgentMath-8B achieves 89.8%, 84.7%, and 71.3%, significantly outperforming the tool-augmented CIR-Qwen3-NT8-8B and text-only DS-0528-Qwen3-8B (86.0%, 76.3%, 61.5%). For larger-scale models (30B ~ 32B), AgentMath-30B-A3B reaches 90.6%, 86.4%, and 73.8%, exceeding the tool-augmented STILL-3-TOOL-32B (81.7%, 64.2%, 45.4%) and text-only Qwen3-30B-A3B-Thinking-2507 (87.7%, 85.0%, 74.3%).

Table 1: Performance of AgentMath on AIME24/25, and HMMT25. Our model (highlighted in blue) is compared against other leading models, with accuracy (avg@32) as the evaluation metric. Due to space limitations, we use DS, QW2.5, and QM2.5 to denote DeepSeek-R1, Qwen2.5, and Qwen-2.5-Math, respectively. For a more detailed and comprehensive performance table, refer to Table 4 in the Appendix.

| Models | Base Model | Tool Use | AIME24 | AIME25 | HMMT25 |
|---|---|---|---|---|---|
| Proprietary models | | | | | |
| OpenAI-o4-mini-w/tools | - | ✓ | 98.7 | 99.5 | - |
| OpenAI-o3-w/tools | - | ✓ | 95.2 | 98.4 | - |
| OpenAI-o4-mini | - | ✗ | 93.4 | 92.7 | 83.0 |
| Gemini-2.5-Pro | - | ✗ | 92.0 | 88.0 | 82.5 |
| OpenAI-o3 | - | ✗ | 91.6 | 88.9 | 77.5 |
| OpenAI-o3-mini | - | ✗ | 87.3 | 86.3 | 53.0 |
| Claude-Opus-4.0-Thinking | - | ✗ | 83.0 | 72.0 | 58.3 |
| Frontier Models (1B ~ 2B) | | | | | |
| ToRL-1.5B | QM2.5-1.5B-Base | ✓ | 26.7 | 26.7 | - |
| DS-Distill-Qwen-1.5B | QM2.5-1.5B-Base | ✗ | 28.8 | 21.8 | 15.3 |
| CoRT-1.5B | DS-Distill-Qwen-1.5B | ✓ | 43.1 | 30.2 | 20.1 |
| Qwen3-1.7B Thinking | Qwen3-1.7B-Base | ✗ | 52.0 | 35.3 | 23.3 |
| OpenThinker3-1.5B | QW2.5-1.5B-Instruct | ✗ | 52.0 | 41.7 | 27.3 |
| OpenReasoning-1.5B | QW2.5-1.5B-Instruct | ✗ | 55.5 | 45.6 | 31.5 |
| AgentMath-1.7B | Qwen3-1.7B-Base | ✓ | 59.6 | 48.1 | 40.2 |
| Frontier Models (7B ~ 8B) | | | | | |
| ToRL-7B | QM2.5-7B-Base | ✓ | 43.3 | 30.0 | - |
| ZeroTIR-7B | QW2.5-7B-Base | ✓ | 46.7 | 30.0 | 22.5 |
| SimpleTIR-7B | QW2.5-7B-Base | ✓ | 50.5 | 30.9 | 29.7 |
| AFM-7B | QW2.5-7B-Instruct | ✓ | 51.9 | 37.8 | - |
| rStar-Math-Qwen-7B | QM2.5-7B-Base | ✓ | 53.3 | - | - |
| DS-Distill-Qwen-7B | QM2.5-7B-Base | ✗ | 55.0 | 39.7 | - |
| CIR-Qwen3-NT8-8B | Qwen3-8B | ✓ | 61.5 | 46.3 | - |
| AReal-boba-7B | DS-Distill-Qwen-7B | ✗ | 61.9 | 48.3 | 29.4 |
| Skywork-OR1-7B | DS-Distill-Qwen-7B | ✗ | 70.2 | 54.6 | 35.7 |
| POLARIS-7B-Preview | DS-Distill-Qwen-7B | ✗ | 72.6 | 52.6 | - |
| Qwen3-8B-Thinking | Qwen3-8B-Base | ✗ | 76.0 | 67.3 | 44.7 |
| OpenReasoning-7B | QW2.5-7B-Instruct | ✗ | 84.7 | 78.2 | 63.5 |
| DS-0528-Qwen3-8B | Qwen3-8B-Base | ✗ | 86.0 | 76.3 | 61.5 |
| AgentMath-8B | Qwen3-8B-Base | ✓ | 89.8 | 84.7 | 71.3 |
| Frontier Models (30B ~ 32B) | | | | | |
| ZeroTIR-32B | Q2.5-32B-Base | ✓ | 56.7 | 33.3 | 20.0 |
| START-32B | QwQ-32B | ✓ | 66.7 | 47.1 | - |
| AFM-32B | QW2.5-32B-Instruct | ✓ | 66.7 | 59.8 | - |
| ReTool-32B | QW2.5-32B-Instruct | ✓ | 67.0 | 49.3 | |
| rStar2-Agent-32B | QW2.5-32B-instruct | ✓ | 69.4 | 57.3 | - |
| ReTool-R1-32B-distill | DS-Distill-Qwen-32B | ✓ | 72.5 | 54.3 | - |
| DS-Distill-Qwen-32B | QW2.5-32B-Base | ✗ | 72.9 | 59.0 | 33.0 |
| Qwen3-30B-A3B-Instruct-2507 | Qwen3-30B-A3B-Base | ✗ | 72.9 | 61.3 | 43.0 |
| CoRT-32B | DS-Distill-Qwen-32B | ✓ | 76.7 | 67.1 | - |
| QwQ-32B | - | ✗ | 79.5 | 65.3 | 48.0 |
| STILL-3-TOOL-32B | DS-Distill-Qwen-32B | ✓ | 81.7 | 64.2 | 45.4 |
| Skywork-OR1-32B | DS-Distill-Qwen-32B | ✗ | 82.2 | 73.3 | - |
| AM-Thinking-v1-32B | Qwen 2.5-32B-Base | ✗ | 85.3 | 74.4 | - |
| Qwen3-30B-A3B-Thinking-2507 | Qwen3-30B-A3B-Base | ✗ | 87.7 | 85.0 | 71.4 |
| AgentMath-30B-A3B | Qwen3-30B-A3B-Instruct-2507 | ✓ | 90.6 | 86.4 | 73.8 |
| Frontier Models (>32B) | | | | | |
| Qwen3-235B-A22B-Instruct-2507 | Qwen3-235B-A22B-Base | ✗ | 79.2 | 70.3 | 55.4 |
| DS-671B | DeepSeek-V3-Base | ✗ | 79.8 | 70.0 | 44.4 |
| Qwen3-235B-A22B-Thinking | Qwen3-235B-A22B-Base | ✗ | 85.7 | 81.5 | 62.5 |
| DS-671B-0528 | DeepSeek-V3-Base | ✗ | 91.4 | 87.5 | 77.0 |
| Qwen3-235B-A22B-Thinking-2507 | Qwen3-235B-A22B-Base | ✗ | 94.2 | 92.3 | 83.9 |
| AgentMath-235B-A22B-SFT | Qwen3-235B-A22B-Instruct-2507 | ✓ | 93.4 | 90.8 | 81.7 |

Notably, the Mixture-of-Experts model AgentMath-30B-A3B with 3B active parameters and 30B total parameters outperforms most dense 30B models on AIME24 and AIME25, approaching the performance of DS-671B-0528. This demonstrates that our approach achieves competitive performance with substantially larger models while maintaining computational efficiency.

At ultra-large scale (>32B), AgentMath-235B-A22B-SFT achieves 93.4%, 90.8%, and 81.7% across the three benchmarks, surpassing DS-671B-0528 (91.4%, 87.5%, 77.0%) and achieving performance on par with Qwen3-235B-A22B-Thinking-2507. Compared to proprietary models, AgentMath-30B-A3B outperforms OpenAI-o3-mini and Claude-Opus-4.0-Thinking on three benchmarks and is competitive with OpenAI-o3 and Gemini-2.5-Pro. Furthermore, AgentMath-235B-A22B-SFT exceeds OpenAI-o3, approaching OpenAI-o4-mini and Gemini-2.5-Pro. Notably, due to computational constraints, AgentMath-235B-A22B is trained solely via SFT.

These results validate the effectiveness of our tool-augmented data synthesis method and large-scale reinforcement learning training strategy, yielding consistent improvements in math reasoning capabilities across diverse model scales. We provide the case study of AgentMath in Appendix A.8.

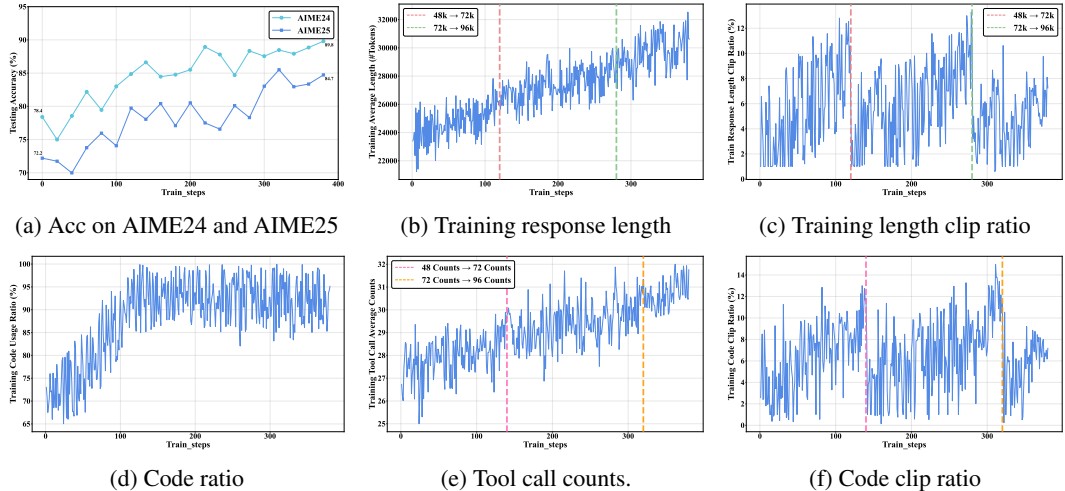

Figure 3: Evolution of key metrics during multi-stage RL training: (a-c) accuracy on AIME24 and AIME25, response length, length clip ratio; (d-f) code ratio, tool call counts, code clip ratio.

## 3.5 COGNITION ANALYSIS

**Tool-Augmented Synthetic Data vs. Text-Based Data.** To assess the effectiveness of tool-augmented synthetic data method, we conduct experiments addressing two core questions: the comparative advantage of tool-augmented synthetic data over text-based data in SFT, and the impact of tool augmentation on performance and efficiency during RL. We employ Qwen3-8B-Base as the backbone model and an identi-

Table 2: Performance comparison between AgentMath and Text-Based Model in SFT and RL stages.

| Models | AIME24 | AIME25 |
|---|---|---|
| Text-Based-SFT-20k | 57.1% | 49.2% |
| AgentMath-SFT-20k | 60.5% | 53.3% |
| Text-Based-RL | 68.7% | 57.5% |
| AgentMath-RL | 76.2% | 67.5% |

cal 20k data. As shown in Table 2, in SFT stage, AgentMath-SFT achieves accuracies of 60.5% on AIME24 and 53.3% on AIME25, surpassing the text-based baseline by 3.4% and 4.1%, validating our method of converting computation-intensive steps into executable code. The benefits are further amplified in RL: as detailed in Figure 4 and Table 2, AgentMath-RL requires only ~400 steps to reach 76.2% (AIME24) and 67.5% (AIME25), a 4.0× efficiency improvement over the ~1600 steps needed by the Text-Based-SFT model to achieve inferior results (68.7% and 57.5%). Notably, it matches the Text-Based model's final performance in just 100–200 steps. Additionally, inference efficiency improves substantially, as indicated in Figure 5, with sequence lengths reduced by ~4k tokens (~14%) and slower length growth, attributable to precise code execution replacing verbose manual calculations. Collectively, AgentMath demonstrates superior accuracy, training efficiency, and inference scalability, confirming the power of interleaving natural language reasoning with computational tools. See Appendix A.7.1 for more details.

**Multi Stage RL Training.** Following the supervised fine-tuning phase, we observed that the model frequently generated responses exceeding 32k tokens for complex mathematical problems, with the most challenging instances surpassing 64k tokens. To effectively balance training efficiency with model capacity, we developed an adaptive, multi-stage RL strategy that progressively unlocks the model's potential by dynamically expanding the sequence length and tool-call budget. This process is triggered automatically when truncation rates for either response length or tool usage exceed 10%, incrementally increasing the context length from 48k to 72k (at step 120) and finally to 96k (at step 280), while the tool-call limit expands from 48 to 72 (step 140) and then to 96 (step 320), as illustrated in Figure 3c and 3f. The training progression, detailed in Figure 3, reveals significant trends: generated trajectory average lengths increased from 24k to 30k (Figure 3b), tool average invocation frequency rose from 27 to 31 calls per problem (Figure 3e), and code utilization improved markedly from 70% to 95% (Figure 3d), indicating enhanced proficiency in multi-step reasoning. Consequently, accuracy on the AIME24 benchmark rose from 78.4% to 89.8% (+11.4%) and on AIME25 from 72.2% to 84.7% (+12.5%) (Figure 3a), with consistent improvements following each capac-

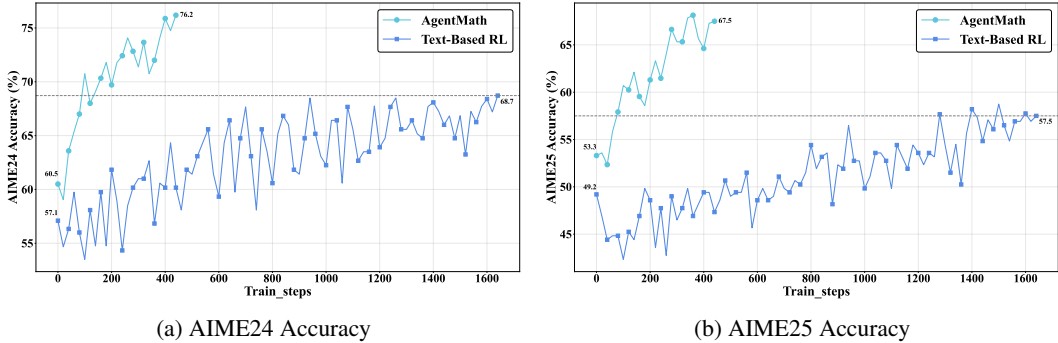

| (a) AIME24 Accuracy | (b) AIME25 Accuracy |

Figure 4: Performance Comparison of AgentMath vs. Text-Based Model in the RL phase on AIME24/25. Both models were initialized from their best SFT checkpoint trained on 20k data.

ity expansion. Notably, the model exhibited emergent code self-correction capabilities as shown in Appendix A.8.2 Figure 9. These results, along with the performance of AgentMath with different backbones detailed in Table 5, confirm the efficacy of our strategy. The experiments establish three key insights: (1) expanded capacity is crucial for facilitating deeper reasoning chains; (2) The composite reward effectively guides the model's tool-call decisions; and (3) the stable training under extreme configurations (96k tokens, 96 tool calls) underscores the robustness of the AgentMath framework and its asynchronous training infrastructure. See Appendix A.7.2 for more details.

**Synthetic Data Refinement and Scaling Law.** As detailed in Table 3, we conduct a systematic evaluation of AgentMath's data synthesis pipeline, revealing that progressive multi-dimensional refinement is critical for performance. The initial unrefined synthetic data yielded suboptimal results (AIME24: 35.3%; AIME25: 25.7%), primarily due to formatting inconsistencies and non-executable code. By systematically ap-

Table 3: Performance improvements on AIME24/25 through progressive refinement steps.

| Models / Refinement Steps | AIME24 | AIME25 |
|---|---|---|
| Initial Unrefined CI-Synthetic Data (20k) | 35.3% | 25.7% |
| + Format consistency correction | 47.4% | 40.1% |
| + Code executability verification | 52.8% | 44.8% |
| + Environmental feedback alignment | 56.3% | 48.3% |
| + Self-correction capability injection | 58.6% | 50.8% |
| + SFT with selective feedback masking | 60.5% | 53.3% |

plying refinements, including format consistency, code executability verification, and environment feedback alignment, performance substantially improved to 58.6% on AIME24 and 50.8% on AIME25. The subsequent integration of a self-correction mechanism, combined with supervised fine-tuning using selective feedback masking guided by code execution results, culminated in final accuracies of 60.5% on AIME24 and 53.3% on AIME25, underscoring the necessity of each refinement stage. Furthermore, scaling the tool-augmented dataset from 2k to 300k (Figure 7) yielded significant performance gains, improving accuracy from 27.2% to 78.4% on AIME24 and from 21.1% to 72.2% on AIME25. This combination of rigorous quality control and effective data scaling effectively mitigates data scarcity in tool-augmented mathematical reasoning, establishing a robust foundation for high-performance reasoning agents. Further details are provided in Appendix A.7.3.

Owing to space constraints, a comprehensive analysis of the AgentMath framework's training efficiency and the impact of the partial rollout segment count is deferred to Appendix A.7.4.

## 4 CONCLUSION

This paper introduces AgentMath, a tool-augmented agent framework that seamlessly integrates language model reasoning with the precision of code interpreters to tackle complex mathematical problems. Extensive evaluations show that AgentMath achieves state-of-the-art performance on challenging mathematical competition benchmarks, including AIME24, AIME25, and HMMT25. Remarkably, AgentMath-30B-A3B with only 3B active parameters achieves 90.6%, 86.4%, and 78.9% accuracy, outperforming OpenAI-o3-mini and Claude-Opus-4.0-Thinking while remaining competitive with OpenAI-o3, Gemini-2.5-Pro, and DeepSeek-R1-671B. Furthermore, our work highlights the essential role of automated tool-augmented data synthesis and a scalable asynchronous training infrastructure in enabling effective and efficient agentic learning for mathematical reasoning.

ACKNOWLEDGMENTS

This work was supported by the Major Key Project of Pengcheng Laboratory (Grant No. PCL2025A12) and the CIE-Tencent Ph.D. Student Research Incentive Program (Tencent Hunyuan Special Fund). Sincere thanks to Dr. Wenfeng Deng, Team Leader Di Wang and other colleagues for their valuable guidance and insightful suggestions throughout this research.

REPRODUCIBILITY STATEMENT

To ensure the reproducibility of AgentMath, we provide comprehensive details on algorithms, data synthesis procedures, Agent RL training, and experimental configurations throughout the main paper and appendices. The data synthesis pipeline, multi-dimensional quality refinement mechanisms, and the design of synthesis and judge prompts are detailed in Sections 2.1 and 2.2, and Appendices A.2 and A.6. The RL training framework and implementation details are presented in Section 2.3 and Appendix A.3. Optimization strategies for RL training efficiency and the design of Algorithm 1 are described in Section 2.4 and Appendix A.4. We systematically evaluate the individual contributions of tool-augmented data synthesis, multi-stage RL training, scaling laws of synthetic data, and multi-dimensional quality refinement through ablation experiments in Section 3.5 and Appendix A.7. Case analyses are provided in Appendix A.8. Detailed experimental settings, including SFT and RL training data, training strategies, evaluation settings and main results, are documented in Sections 3.1 , 3.2, 3.3, 3.4 and Appendix A.5. We hope these will help researchers better understand and reproduce our work.

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

# A  APPENDIX

## A.1  RELATED WORK

**Mathematical Reasoning in LLMs.** Large Language Models (LLMs) have made remarkable progress in mathematical reasoning(Wei et al., 2022; Yao et al., 2023; Luong et al., 2024; OpenAI et al., 2024; Team et al., 2025a; DeepSeek-AI et al., 2025; xAI, 2023; Claude, 2025; Team et al., 2023; Yang et al., 2024a;b; He et al., 2025b; Fang et al., 2025; Zhang et al., 2025a; Wu, 2025b;a). The introduction of Chain-of-Thought (CoT)(Wei et al., 2022; Yao et al., 2023) prompting enabled models to decompose complex problems into intermediate reasoning steps, substantially enhancing their problem-solving capabilities. Subsequently, research has shifted from a singular focus on model scaling towards optimizing the reasoning process itself(Snell et al., 2024). This paradigm shift has spurred the development of Large Reasoning Models (LRMs) trained with advanced methods like Reinforcement Learning(Schulman et al., 2017; Li et al., 2024; Shao et al., 2024), Direct Preference Optimization(Rafailov et al., 2023; Lai et al., 2024; Pang et al., 2024; Luo et al., 2024b; Feng et al., 2024; 2025c; Zhou et al., 2025b; 2026), and Monte Carlo Tree Search(Xie et al., 2024; Wang et al., 2023b; Guo et al., 2025). State-of-the-art models such as OpenAI's o1 and DeepSeek-R1(OpenAI et al., 2024; DeepSeek-AI et al., 2025; Team, 2025d; Team et al., 2023; 2025a;b) exhibit human-like cognitive planning on long-chain reasoning tasks, pushing the frontiers of mathematical performance. Despite these advances, reasoning purely within natural language is constrained by inherent limitations: complex arithmetic and symbolic manipulations are prone to error, and self-correction is often inefficient. It may cause hallucinations and risks(Feng et al., 2025a; 2026). These shortcomings fundamentally limit their accuracy and efficiency on competition-level mathematical problems.

**Tool-Augmented LLM Reasoning.** Tool-augmented reasoning has emerged as a promising solution to the limitations of text-based approaches(Li et al., 2025e; Zhou et al., 2025a; Lin & Xu, 2025; Zhang et al., 2025b; Luo et al., 2024a; Jin et al., 2025b; Liu et al., 2025b; Luo et al., 2023; Azerbayev et al., 2023; Yu et al., 2023). Program-of-Thought (PoT)Chen et al. (2023); Gao et al. (2023); Yue et al. (2023); Jin et al. (2025a); Wang et al. (2024; 2023a); Cobbe et al. (2021); Hendrycks et al. (2021); Lightman et al. (2023); Tian et al. (2025); Shao et al. (2024); OpenAI (2024; 2025); Wang et al. (2025b); Chang et al. (2024); Gou et al. (2023a); Liu et al. (2024); Qu et al. (2025); Song et al. (2025); Li et al. (2025d); Schick et al. (2023); Zhang et al. (2024b;c) pioneered delegating computational steps to external code interpreters, enhancing numerical accuracy. ToRA(Gou et al., 2023b) subsequently developed code-integrated reasoning frameworks tailored for mathematical problems, demonstrating the efficacy of specialized tools in complex computations. START(Li et al., 2025b) generates tool-augmented trajectories via prompt engineering, though random code insertion often yields inefficient utilization. STILL3(Chen et al., 2025b) relies on prompt-based data construction, and CoRT(Li et al., 2025a) employs high-quality human annotations but faces scalability constraints. These approaches predominantly depend on supervised fine-tuning, preventing models from learning debugging strategies from execution failures or adaptively mastering tool invocation timing and methods. While Retool(Feng et al., 2025b) combines data rewriting with reinforcement learning for optimization, improvements over existing LRMs (i.e, DeepSeek-R1-Distill-Qwen-32B(DeepSeek-AI et al., 2025)) remain marginal. Current tool-augmented methods thus face three critical challenges: scarcity of high-quality data, inadequate policy learning, and inefficient training on long sequences. AgentMath mitigates these limitations by automating tool-augmented data synthesis and employing reinforcement learning to enable autonomous exploration of optimal tool-use strategies, including tool invocation and code self-correction.

**Agentic Reinforcement Learning.** Reinforcement learning (RL) offers a powerful framework for cultivating autonomous, decision-making agents from LLMs (Dong et al., 2025; Zhang et al., 2025c; Xue et al., 2025; Li et al., 2025c; Singh et al., 2025; Shang et al., 2025; Nguyen et al., 2025; Wei et al., 2025; Luo et al., 2025b; Hao et al., 2025; Agarwal et al., 2025; Lu et al., 2025b; Zeng et al., 2025a; Liu et al., 2025d; Du et al., 2025b; Chang et al., 2025; Feng et al., 2025d; Li et al., 2025d; Song et al., 2025; Nakano et al., 2022; Mai et al., 2025; Wang et al., 2025a; Du et al., 2025a; Gao et al., 2025a; Mei et al., 2025; Wang et al., 2025c; Liu et al., 2025a; Li et al., 2025f; Lu et al., 2025a; Gao et al., 2025b; Liu et al., 2025e; Hu et al., 2024; Shen et al., 2024; Wang et al., 2025d; Zhu et al., 2025; Mei et al., 2024; Fan et al., 2025). In information retrieval, models like Search-R1 and R1-Searcher (Li et al., 2025d; Song et al., 2025; Nakano et al., 2022) have demonstrated how outcome-based rewards can successfully guide agents to query search engines. In mathematical

reasoning, recent work has explored RL for emergent tool use. ToRL (Li et al., 2025e) utilizes RL to train an agent to operate a code interpreter without predefined patterns, while concurrent work on the scaling laws of agentic RL has revealed that simple, outcome-based rewards often foster greater exploration and policy innovation than complex process-based rewards(Lightman et al., 2023; Wang et al., 2023b). Similarly, ReTool(Feng et al., 2025b) leverages RL to teach models strategic tool call, significantly outperforming SFT baselines and uncovering cognitive patterns in code-invocation decisions. Nevertheless, existing RL methods face a critical bottleneck when applied to competition-level mathematics. These problems can generate exceptionally long reasoning chains (i.e., 64k tokens) with dense tool interactions (e.g., 64 calls), a scale that overwhelms conventional batch-synchronous training architectures. AgentMath alleviates this scalability challenge through a suite of technical innovations, including request-level asynchronous rollout scheduling, agentic partial rollouts, and prefix-aware weighted load balancing. These techniques enable efficient RL training on ultra-long sequences with massive tool usage, boosting training throughput by 4–5x and paving the way for developing more sophisticated and scalable mathematical reasoning agents.

## A.2 PROBLEM FORMULATION AND INTERACTION PROTOCOL

### A.2.1 PROBLEM FORMULATION

Tool-augmented mathematical reasoning is formalized as a Markov Decision Process (MDP), wherein the LLM-based policy agent iteratively interacts with a sandboxed execution environment. Given a problem statement $P$, the policy $\pi_\theta$ generates trajectories comprising interleaved reasoning segments and executable code blocks, while the environment $\mathcal{E}$ deterministically executes submitted code and returns corresponding outputs.

The objective is to construct an optimal trajectory $\tau^* = \{(z_1, o_1), \ldots, (z_T, o_T)\}$, where $(z_t, o_t)$ denotes the action-observation pair at timestep $t$. The state transition dynamics are characterized by:

$$
\begin{aligned}
z_t &\sim \pi_\theta(\cdot \mid s_t), \quad s_t = (P, \tau_{t-1}) \\
o_t &= \begin{cases} \mathcal{E}(c_t), & \text{if } z_t = c_t \in \mathcal{C} \\ \varnothing, & \text{if } z_t \in \mathcal{T} \end{cases} \\
\tau_t &= \tau_{t-1} \cup \{(z_t, o_t)\}
\end{aligned}
\tag{1}
$$

where $s_t$ represents the current state comprising the problem and interaction history, $\mathcal{C}$ and $\mathcal{T}$ denote the code and thought action spaces respectively, and $\mathcal{E}(c_t)$ returns the execution result of code block $c_t$. The interaction terminates upon generation of a terminal token or exhaustion of the computational budget.

### A.2.2 STRUCTURED INTERACTION PROTOCOL

The implementation employs a structured markup protocol to delineate reasoning and tool invocation boundaries. Natural language reasoning is encapsulated within <think> ...</think> tags, executable code is delimited by  ... tags, and execution feedback is injected through <interpreter> ...</interpreter> tags.

The generation-execution cycle operates through bidirectional information exchange: upon completion of a  segment, generation is suspended while the extracted code undergoes execution in the sandboxed environment. The resulting output, whether successful execution, error message, or timeout notification, is subsequently incorporated into the context as an <interpreter> segment. This feedback mechanism enables adaptive strategy refinement, wherein the model conditions its subsequent generation on execution outcomes to perform error correction, strategy adjustment, or continued reasoning. Such fine-grained interaction traces provide rich supervision signals amenable to reinforcement learning optimization.

### A.2.3 SUPERVISED FINE-TUNING WITH SELECTIVE FEEDBACK MASKING

During supervised fine-tuning on $\mathcal{D}_{\text{SFT}}$, the model must learn to generate reasoning and code while avoiding memorization of deterministic interpreter outputs. Consequently, tool outputs are masked during loss computation. For a training sample $\tau = (z_1, o_1, \ldots, z_T, o_T)$, where $z_t$ represents model-generated segments and $o_t$ denotes external feedback, the standard autoregressive loss is expressed

---

**Algorithm 1** Agentic Reinforcement Learning with Partial Rollouts

---

1: **Initialize:** Unfinished pool $\mathcal{U} \leftarrow \emptyset$; Experience buffer $\mathcal{B} \leftarrow \emptyset$; Global limits $L_{\text{global}}$, $T_{\text{global}}$;
    Segment limits $L_{\text{seg}}$, $T_{\text{seg}}$.
2: **for** each training iteration $k = 1, 2, \ldots$ **do**
3:    tasks_to_process $\leftarrow$ Sample($\mathcal{P} \cup \mathcal{U}$)
4:    new_segments $\leftarrow$ **Rollout**(tasks_to_process, $L_{\text{seg}}$, $T_{\text{seg}}$)         ▷ Asynchronous generation
5:    finished_trajectories $\leftarrow \emptyset$,    next_unfinished_pool $\leftarrow \emptyset$
6:    **for** each trajectory $\tau$ in new_segments **do**
7:        **if** $\tau$ ends with EOS **or** length($\tau$) $\geq L_{\text{global}}$ **or** tools($\tau$) $\geq T_{\text{global}}$ **then**
8:            Add $\tau$ to finished_trajectories
9:        **else**
10:           Add $\tau$ to next_unfinished_pool
11:    $\mathcal{U} \leftarrow$ next_unfinished_pool
12:    Add finished_trajectories to $\mathcal{B}$; **UpdatePolicy**($\mathcal{B}$)

---

as:

$$\mathcal{L}_{\text{SFT}}(\theta) = -\sum_{t=1}^{T} \log \pi_\theta(z_t \mid P, \tau_{<t}).$$

A masking function $\mathbb{I}(\cdot)$ is introduced to identify tokens originating from $<$interpreter$>$ segments, yielding the modified loss:

$$\mathcal{L}_{\text{SFT-masked}}(\theta) = -\sum_{t=1}^{T} \sum_{k=1}^{|z_t|} \left(1 - \mathbb{I}(z_{t,k})\right) \log \pi_\theta(z_{t,k} \mid P, \tau_{<t}, z_{t,<k}),$$

where $z_{t,k}$ denotes the $k$-th token of segment $z_t$, and $\mathbb{I}(z_{t,k}) = 1$ if and only if $z_{t,k}$ resides within $<$interpreter$>$ tags. This selective masking ensures that gradient updates originate exclusively from model-generated reasoning and code, thereby shaping intrinsic reasoning capabilities and decision-making processes while treating external feedback as non-trainable contextual information.

## A.3   GROUP RELATIVE POLICY OPTIMIZATION

We employs Group Relative Policy Optimization (GRPO) as the core optimization algorithm. GRPO eliminates the requirement for value function approximation, thereby reducing computational complexity through group-wise trajectory sampling and intra-group reward normalization. The optimization objective is formulated as:

$$\mathcal{J}_{\text{GRPO}}(\theta) = \mathbb{E}_{\substack{P \sim \mathcal{D}, \\ \{T_i\}_{i=1}^{G} \sim \pi_{\theta_{\text{old}}}(\cdot|P)}} \left[ \frac{1}{G} \sum_{i=1}^{G} \frac{1}{|T_i|} \sum_{t=1}^{|T_i|} \min\left(r_{i,t}(\theta)\,\hat{A}_i, \text{clip}(r_{i,t}(\theta), 1-\varepsilon, 1+\varepsilon)\,\hat{A}_i\right) \right],$$

where $r_{i,t}(\theta)$ denotes the importance sampling ratio. The advantage estimate $\hat{A}_i$ is computed through within-group normalization:

$$\hat{A}_i = \frac{R(T_i) - \mu_\mathcal{R}}{\sigma_\mathcal{R} + \delta},$$

with $\mu_\mathcal{R}$ and $\sigma_\mathcal{R}$ representing the group mean and standard deviation, respectively, and $\delta$ serving as a numerical stability constant. Following recent advances in DAPO, the KL divergence penalty is omitted to facilitate exploration, while the Clip-Higher strategy is adopted to enhance learning of high-entropy, low-probability tokens critical for complex reasoning tasks.

## A.4   AGENTIC PARTIAL ROLLOUT ALGORITHM

## A.5   EXPERIMENTAL DETAILS

This section describes the training data construction, model training, and evaluation settings.

### A.5.1 Supervised Fine-tuning (SFT) Data Construction

The supervised fine-tuning (SFT) data construction pipeline consists of three phases.

**Stage 1: Foundational Data Curation and Filtering.** We aggregate a raw corpus from multiple public math reasoning datasets (i.e., AM-Thinking, OpenThoughts, and AceReason). After problem-level deduplication, we apply N-gram ($N = 4$) and MinHash LSH algorithms to eliminate overlaps with all evaluation sets, including AIME24, AIME25 and HMMT25 with 0.6 similarity threshold. To further prevent data leakage, we compute semantic similarities between training data and evaluation sets using gte-large model(Zhang et al., 2024a) , filtering out the top-5 most similar samples. We then annotate each problem with difficulty scores from 0 to 10 using Qwen3-30B, retaining data with scores above 5, which yields 392k samples. Finally, using DeepSeek-R1-0528 to generate solutions and removing instances with incorrect answers, we obtain 346k high-quality data.

**Stage 2: Tool-Augmented Data Synthesis.** We firstly decompose the problem-solving process into discrete reasoning segments and perform tool-augmented synthesis for each segment using the Prompt presented in Appendix A.6.2 via DeepSeek-V3-0324. During synthesis, we filter out samples with synthesis format error, tool execution failures, or incorrect final answer, yielding 302k high-fidelity, tool-augmented solution trajectories.

**Stage 3: Self-Correction Data Generation.** To incorporate self-correction mechanisms, we sample 30k instances from trajectories with unsuccessful code execution and leverage the Self-correction Prompt presented in Appendix A.6.1 to guide DeepSeek-V3-0324 in generating correction processes, producing 14k valid self-correction trajectories.

Through this comprehensive pipeline, we construct a 316k tool-augmented synthetic training set with an average of 8.3 tool calls and an average sequence length of 16.9K tokens per sample.

### A.5.2 Reinforcement Learning (RL) Data Construction

For RL, we collect problems from multiple public high-quality RL datasets (i.e., DeepScaler, Skywork-OR1, Retool, POLARIS). We apply the same deduplication strategy as for SFT data, ensuring no overlap with evaluation sets through N-gram (N=4), MinHash LSH, and semantic similarity computation with 0.6 similarity threshold. To identify challenging problems, we use AgentMath-8B-SFT to perform 8 inference attempts on all data, filtering out problems that are solved correctly 8 times, culminating in a final set of 42k high-difficulty RL training data. This focuses training on hard instances that push the model's strategic capabilities and maximize potential gains from RL.

### A.5.3 Training Settings

**Base Models.** Our experiments utilize four base models from the Qwen3 series: Qwen3-1.7B-Base and Qwen3-8B-Base are pre-trained models without post-training; Qwen3-30B-A3B-Instruct-2507 (Non-Thinking mode, 30B total parameters, 3B activated) and Qwen3-235B-A22B-Instruct-2507 (Non-Thinking mode, 235B total parameters, 22B activated) are instruction-tuned Mixture-of-Experts (MoE) models without long chain-of-thought training.

**SFT Training.** We employ the Llama-Factory framework, training for 6 epochs with learning rates of $6 \times 10^{-5}$ (1.7B, 8B, A3B models) and $2 \times 10^{-5}$ (A22B model), using cosine decay with 10% warmup, batch size of 512, and maximum sequence length of 32k tokens.

**RL Training.** Our RL training is built on the verl 0.5.0.dev0 framework(Sheng et al., 2024), initializing from the best SFT checkpoint and using VLLM(Kwon et al., 2023) as the inference engine, with a 128-node Sandbox cluster for large-scale code execution. We use a constant learning rate of $1 \times 10^{-6}$, a batch size of 64, and a temperature of 1.0, performing 8 rollouts per problem. Training progresses through three stages, dynamically adjusted to maintain length truncation and tool-call excess rates below 10%: maximum response length increases from 48k to 72k to 96k tokens, with corresponding tool invocation limits of 48, 72, and 96 calls, and partial rollout counts of 2, 3, and 4,

ensuring each segment rollout remains within 24k tokens and 24 tool invocations. Due to computational constraints, AgentMath-235B-A22B is trained solely via supervised fine-tuning (SFT).

### A.5.4 EVALUATION SETTINGS

**Benchmarks.** We primarily evaluate on AIME24, AIME25, and HMMT25. These challenging U.S. high school math competitions feature problems in algebra, number theory, combinatorics, and geometry, providing a robust test of advanced mathematical modeling, multi-step logical reasoning, and strategic problem-solving.

**Evaluation Metrics.** To ensure robust evaluation, we perform 32 independent inference runs per test sample, using avg@32 as the pass@1 metric.

**Inference Parameters.** We use a consistent configuration: maximum sequence length = 96K tokens, maximum tool calls = 96, code interpreter output limit = 1024 tokens, temperature = 0.6, and top-p = 0.95.

**Answer Extraction and Validation.** We extract final answers from `\boxed{}` markers in model responses and employ Math-Verify library for exact comparison with ground truth answer, determining correctness only when verification returns True.

### A.5.5 DETAIL RESULTS

Table 4: Performance comparison (avg@32 accuracy) of AgentMath against state-of-the-art models on AIME24, AIME25, and HMMT25 benchmarks. Evaluation follows DeepSeek-R1 framework (temperature=0.6, top_p=0.95). AgentMath models (highlighted in blue) achieve superior results across all scales, with the 30B variant competitive against 671B models.

| Models | Base Model | Tool Use | AIME24 | AIME25 | HMMT25 |
|---|---|---|---|---|---|
| *Proprietary models* | | | | | |
| OpenAI-o4-mini-w/tools(OpenAI, 2025) | - | ✓ | 98.7 | 99.5 | - |
| Grok-4-w/tools(xAI, 2023) | - | ✓ | - | 98.8 | - |
| OpenAI-o3-w/tools(OpenAI, 2025) | - | ✓ | 95.2 | 98.4 | - |
| OpenAI-o4-mini(OpenAI, 2025) | - | ✗ | 93.4 | 92.7 | 83.0 |
| Gemini-2.5-Pro(Team et al., 2023) | - | ✗ | 92.0 | 88.0 | 82.5 |
| OpenAI-o3(OpenAI, 2025) | - | ✗ | 91.6 | 88.9 | 77.5 |
| Seed-1.6-thinking(Seed et al., 2025) | - | ✗ | 90.3 | 86.0 | - |
| OpenAI-o3-mini(OpenAI, 2025) | - | ✗ | 87.3 | 86.3 | 53.0 |
| Claude-Opus-4.0-Thinking(Claude, 2025) | - | ✗ | 83.0 | 72.0 | 58.3 |
| Grok-3-Beta Thining(xAI, 2023) | - | ✗ | 83.9 | 77.3 | - |
| Kimi-k1.5(Team et al., 2025a) | - | ✗ | 77.5 | - | - |
| *Frontier Models (1B ∼ 2B)* | | | | | |
| ToRL-1.5B(Li et al., 2025e) | Qwen2.5-Math-1.5B-Base | ✓ | 26.7 | 26.7 | - |
| DeepSeek-R1-Distill-Qwen-1.5B(DeepSeek-AI et al., 2025) | Qwen2.5-Math-1.5B-Base | ✗ | 28.8 | 21.8 | 15.3 |
| DeepScaleR-1.5B-Preview(Luo et al., 2025a) | DeepSeek-R1-Distill-Qwen-1.5B | ✗ | 40.0 | 30.0 | - |
| CoRT-1.5B(Li et al., 2025a) | DeepSeek-R1-Distill-Qwen-1.5B | ✗ | 43.1 | 30.2 | 20.1 |
| Nemontron-Research-Reasoning-Qwen-1.5B(Liu et al., 2025c) | DeepSeek-R1-Distill-Qwen-1.5B | ✗ | 49.6 | 36.0 | 21.7 |
| Qwen3-1.7B Thinking(Team, 2025c) | Qwen3-1.7B-Base | ✗ | 52.0 | 35.3 | 23.3 |
| OpenThinker3-1.5B(Guha et al., 2025) | Qwen2.5-1.5B-Instruct | ✗ | 52.0 | 41.7 | 27.3 |
| OpenReasoning-Nemotron-1.5B(Ahmad et al., 2025) | Qwen2.5-1.5B-Instruct | ✗ | 55.5 | 45.6 | 31.5 |
| AgentMath-1.7B | Qwen3-1.7B-Base | ✓ | 59.6 | 48.1 | 40.2 |
| *Frontier Models (7B ∼ 8B)* | | | | | |
| Qwen2.5-7B-Math-Instruct-TIR(Yang et al., 2024b) | Qwen2.5-Math-7B-Base | ✓ | 20.0 | 26.7 | - |
| Eurus-2-PRIME-7B(Cui et al., 2025) | Qwen-2.5-Math-7B-Base | ✗ | 26.7 | 13.3 | - |
| SimpleRL-Zero-7B(Zeng et al., 2025b) | Qwen-2.5-Math-7B-Base | ✗ | 33.3 | 6.7 | - |
| ToRL-7B(Li et al., 2025e) | Qwen2.5-Math-7B-Base | ✓ | 43.3 | 30.0 | - |
| ZeroTIR(Mai et al., 2025) | Qwen-2.5-7B-Base | ✓ | 46.7 | 30.0 | 22.5 |
| SimpleTIR-7B(Xue et al., 2025) | Qwen2.5-7B-Base | ✓ | 50.5 | 30.9 | 29.7 |
| AFM-7B(Li et al., 2025c) | Qwen2.5-7B-Instruct | ✓ | 51.9 | 37.8 | - |
| rStar-Math-Qwen-7B(Zhang et al., 2024b) | Qwen2.5-Math-7B-Base | ✓ | 53.3 | - | - |
| DeepSeek-R1-Distill-Qwen-7B(DeepSeek-AI et al., 2025) | Qwen2.5-Math-7B-Base | ✗ | 55.0 | 39.7 | - |
| OpenR1-Distill-7B(Face, 2025) | Qwen2.5-Math-7B-Base | ✗ | 57.7 | 39.7 | 25.7 |
| Light-R1-7B-DS(Wen et al., 2025) | DeepSeek-R1-Distill-Qwen-7B | ✗ | 59.1 | 44.3 | 27.6 |
| CIR-Qwen3-NT8-8B(Bai et al., 2025) | Qwen3-8B | ✓ | 61.5 | 46.3 | - |
| AReal-boba-7B(Fu et al., 2025) | DeepSeek-R1-Distill-Qwen-7B | ✗ | 61.9 | 48.3 | 29.4 |
| Skywork-OR1-7B(He et al., 2025a) | DeepSeek-R1-Distilled-Qwen-7B | ✗ | 70.2 | 54.6 | 35.7 |
| POLARIS-7B-Preview(An et al.) | DeepSeek-R1-Distill-Qwen-7B | ✗ | 72.6 | 52.6 | - |
| AceReason-Nemotron-1.1-7B(Chen et al., 2025a) | DeepSeek-R1-Distill-Qwen-7B | ✗ | 72.6 | 64.8 | 42.9 |
| OpenMath-Nemotron-7B(Moshkov et al., 2025) | Qwen2.5-Math-7B | ✗ | 74.8 | 61.2 | - |
| Qwen3-8B Thinking(Team, 2025c) | Qwen3-8B-Base | ✗ | 76.0 | 67.3 | 44.7 |
| MiMo-7B(Xiaomi, 2025) | MiMo-7B-Base | ✗ | 80.1 | 70.2 | 35.7 |
| OpenReasoning-Nemotron-7B(Ahmad et al., 2025) | Qwen2.5-7B-Instruct | ✗ | 84.7 | 78.2 | 63.5 |
| DeepSeek-R1-0528-Qwen3-8B(DeepSeek-AI et al., 2025) | Qwen3-8B-Base | ✗ | 86.0 | 76.3 | 61.5 |
| AgentMath-8B | Qwen3-8B-Base | ✓ | 89.8 | 84.7 | 71.3 |
| *Frontier Models (30B ∼ 32B)* | | | | | |
| Sky-T1-32B-Preview(Team, 2025b) | Qwen2.5-32B-Instruct | ✗ | 43.3 | - | - |
| Open-Reasoner-Zero-Qwen-32B(Hu et al., 2025) | Qwen2.5-32B-Base | ✗ | 48.1 | 36.0 | - |
| DAPO-Qwen-32B(Yu et al., 2025) | Qwen2.5-32B-Base | ✗ | 50.0 | 32.1 | - |
| s1-32B(Muennighoff et al., 2025) | Qwen2.5-32B-Instruct | ✗ | 56.7 | 50.0 | 37.0 |
| ZeroTIR-32B(Mai et al., 2025) | Qwen-2.5-32B-Base | ✓ | 56.7 | 33.3 | 20.0 |
| START-32B(Li et al., 2025b) | QwQ-32B | ✓ | 66.7 | 47.1 | - |
| AFM-32B(Li et al., 2025c) | Qwen2.5-32B-Instruct | ✓ | 66.7 | 59.8 | - |
| ReTool-32B(Feng et al., 2025b) | Qwen2.5-32B-Instruct | ✓ | 67.0 | 49.3 | - |
| rStar2-Agent-Qwen2.5-32B(Shang et al., 2025) | Qwen2.5-32B-instruct | ✓ | 69.4 | 57.3 | - |
| ReTool-R1-32B-distill(Feng et al., 2025b) | DeepSeek-R1-Distill-Qwen-32B | ✓ | 72.5 | 54.3 | - |
| DeepSeek-R1-Distill-Qwen-32B(DeepSeek-AI et al., 2025) | Qwen2.5-32B-Base | ✗ | 72.9 | 59.0 | 33.0 |
| Qwen3-30B-A3B-Instruct-2507(Team, 2025c) (Non-Thinking) | Qwen3-30B-A3B-Base | ✗ | 72.9 | 61.3 | 43.0 |
| Light-R1-32B(Wen et al., 2025) | Qwen2.5-32B-Instruct | ✗ | 76.6 | 64.6 | - |
| CoRT-32B(Li et al., 2025a) | DeepSeek-R1-Distill-Qwen-32B | ✓ | 76.7 | 67.1 | - |
| TinyR1-32B-Preview(Team, 2025e) | DeepSeek-R1-Distill-Qwen-32B | ✗ | 78.1 | 65.3 | - |
| QwQ-32B(Team, 2025d) | - | ✗ | 79.5 | 65.3 | 48.0 |
| Qwen3-30B-A3B-Thinking(Team, 2025c) | Qwen3-30B-A3B-Base | ✗ | 80.4 | 70.9 | 51.0 |
| Qwen3-32B-Thinking(Team, 2025c) | Qwen3-32B-Base | ✗ | 81.4 | 72.9 | - |
| STILL-3-TOOL-32B(Chen et al., 2025c) | DeepSeek-R1-Distill-Qwen-32B | ✓ | 81.7 | 64.2 | 45.4 |
| Skywork-OR1-32B(He et al., 2025a) | DeepSeek-R1-Distill-Qwen-32B | ✗ | 82.2 | 73.3 | - |
| AM-Thinking-v1-32B(Ji et al., 2025) | Qwen 2.5-32B-Base | ✗ | 85.3 | 74.4 | - |
| AM-DeepSeek-R1-0528-Distill-32B(a-m team, 2025) | Qwen 2.5-32B-Base | ✗ | 87.1 | - | - |
| Qwen3-30B-A3B-Thinking-2507(Team, 2025c) | Qwen3-30B-A3B-Base | ✗ | 87.7 | 85.0 | 71.4 |
| OpenReasoning-Nemotron-32B(Ahmad et al., 2025) | Qwen2.5-32B-Instruct | ✗ | 89.2 | 84.0 | 73.8 |
| AgentMath-30B-A3B | Qwen3-30B-A3B-Instruct-2507 | ✓ | 90.6 | 86.4 | 73.8 |
| *Frontier Models (>32B)* | | | | | |
| Qwen3-235B-A22B-Instruct-2507(Non-Thinking)(Team, 2025c) | Qwen3-235B-A22B-Base | ✗ | 79.2 | 70.3 | 55.4 |
| DeepSeek-R1-671B(DeepSeek-AI et al., 2025) | DeepSeek-V3-Base | ✗ | 79.8 | 70.0 | 44.4 |
| Qwen3-235B-A22B-Thinking(Team, 2025c) | Qwen3-235B-A22B-Base | ✗ | 85.7 | 81.5 | 62.5 |
| DeepSeek-R1-671B-0528(DeepSeek-AI et al., 2025) | DeepSeek-V3-Base | ✗ | 91.4 | 87.5 | 77.0 |
| Seed-Oss-36B-Instruct(Team, 2025a) | Seed-OSS-36B-Base | ✗ | 91.7 | 84.7 | - |
| Qwen3-235B-A22B-Thininking-2507(Team, 2025c) | Qwen3-235B-A22B-Base | ✗ | 94.2 | 92.3 | 83.9 |
| AgentMath-235B-A22B-SFT | Qwen3-235B-A22B-Instruct-2507 | ✓ | 93.4 | 90.8 | 81.7 |

### A.6 PROMPT

#### A.6.1 DATA SYNTHESIS FOR CODE SELF-CORRECTION PROMPT

---

**Prompt 1: Data Synthesis For Code Self-correction Prompt**

The following is Your Response based on User Instruction. But there was a code interpreter execution error during the process. Please do the following:

Please:

1. Based on the interpreter's failed execution output, identify the exact code segment that caused the error and explain the reason for the failure.

2. Immediately after the interpreter's failed output, add a transition sentence , such as: "Oops, the code above appears to be throwing an error. I need to fix this to ensure it runs successfully."

3. Correct the erroneous code to ensure it runs successfully.

4. Continue the process from where you left off in your response, completing the remaining steps as planned.

5. Wrap the final output in <output></output>tags.

**User Instruction:** {Input}
**Your Response:** <revised_thinking_process>{output} </revised_thinking_process>

---

#### A.6.2 TOOL-AUGMENTED DATA SYNTHESIS PROMPT

**Prompt 2: Tool-Augmented Data Synthesis Prompt**

You are a professional assistant with expertise in mathematics and Python programming, and a multiple gold medalist in mathematics olympiads and programming competitions. You have the capability to write code and use a code interpreter for calculation. The code will be executed in a sandbox environment, and the results will enhance the reasoning process.

**Task Objective**

Enhance the provided mathematical problem-solving process by replacing complex manual calculations with Python code and their execution results, while preserving the original reasoning logic and structure.

**Instructions**

**1. Identify Computational Steps for Code Replacement**

Identify steps that would benefit from code execution, including:

- Complex symbolic algebra: polynomial expansion, factorization, solving equations
- Advanced calculus: differentiation, integration, evaluating definite integrals
- Probability and combinatorics: complex counting, probability distributions
- Linear algebra: matrix operations, inversion, eigenvalue decomposition
- Numerical computations: approximations, large number calculations, geometric calculations
- Any error-prone or computationally intensive calculations

**Important**: Simple calculations (basic arithmetic like $2 \times 3 = 6$) should remain as text. Only use code for complex computations that require at least 5 lines of implementation.

**2. Code Implementation Requirements**

Each code snippet must:

- Be complete and executable, including all necessary imports
- Use appropriate libraries (sympy, numpy, scipy, etc.)
- Include clear variable definitions and comments
- Explicitly use print() for all outputs
- Demonstrate the computation process, not just final results
- Contain at least 5 lines of code

**3. Integration Guidelines**

- Preserve the original reasoning flow: Keep all logical steps, explanations, and even failed attempts unchanged
- Seamless integration: Code and results should naturally fit within the surrounding text
- Context preservation: Maintain semantic coherence and logical consistency
- No unnecessary changes: Do not modify, delete, or polish unrelated content

**4. Formatting Requirements**

Wrap each code snippet as follows:

```
\n```python\n```\n
```

Follow immediately with execution results:

```
<interpreter>\n</interpreter>
```

**5. Input Format**

User Question:

```
{question}\n<original_thinking_process>\n</original_thinking_process>
```

**6. Output Format**

Provide the enhanced thinking process wrapped in:

```
<revised_thinking_process>\n</revised_thinking_process>
```

**7. Key Principles**

1. Code Necessity: Use code only for complex calculations that warrant automation.
2. Minimal Changes: Modify only the computational steps, preserving all other text.
3. Complete Scripts: Each code block must be self-contained and executable.
4. Accuracy: Execution results must match exactly what the code produces.

### A.6.3 CONSISTENCY JUDGMENT PROMPT

---

**Prompt 3: Consistency Judgment Prompt**

**Role and Objective**

You are a meticulous scientific analyst. Your mission is to determine if **Text A** and **Text B** are substantively equivalent in their core results.

Focus strictly on **numerical values, mathematical expressions, and final conclusions.** Ignore all differences in wording, formatting, and sentence structure.

**Core Principles**

1. **Content Over Form:** Prioritize the core message and data. Completely ignore stylistic choices like wording, formatting (bolding, spacing), and sentence construction.

2. **Standardize Before Comparing:** Before comparison, you must normalize all values to a common standard. Convert percentages to decimals ($50\% \rightarrow 0.5$), unify units ($1m \rightarrow 100cm$), and resolve scientific notation.

3. **Compare Overlapping Information Only:** If one text contains data information absent in the other, ignore the missing parts. Base your comparison solely on the data and claims present in *both* texts. Asymmetry is not a basis for inequivalence.

**Evaluation Criteria**

1. **Numerical Equivalence**

   - **Rule:** Two numbers, A and B, are equivalent if their absolute difference is less than or equal to $1.0$ after normalization.
   - **Formula:** $|A - B| \leq 1.0$
   - **Examples:**
     - **Equivalent**: $|1.453125 - 1.390625| = 0.0625 \ (\leq 1.0)$
     - **Equivalent**: $|-32.015744 - (-32.515744)| = 0.5 \ (\leq 1.0)$
     - **Inequivalent**: $|12.91829 - 11.78172| = 1.136 \ (> 1.0)$
   - **Clarifications:**
     - Differences in significant figures or decimal places are acceptable as long as the absolute difference rule is met.
     - Numerical signs must match (e.g., $5$ and $-5$ are not equivalent).

2. **Mathematical Expression Equivalence**

   - **Rule:** Assess if expressions are mathematically equivalent, not just if they are written identically.
   - **Examples of Equivalence:**
     - **Commutativity/Associativity:** `a + b` is equivalent to `b + a`
     - **Alternative Forms:** `x/2` is equivalent to `0.5x`
     - **Factoring/Expansion:** $x^2 - 1$ is equivalent to $(x - 1)(x + 1)$
     - **Variable Renaming:** $f(x) = x^2$ is equivalent to $g(y) = y^2$
   - **Important Caveat:** If a transformation alters the domain in a way that impacts the conclusion (e.g., introduces division by zero), the expressions are not equivalent.

3. **Conclusion Equivalence**

   - **Rule:** The final answers or main conclusions must align.
     - **Numerical Conclusions:** Must meet the numerical equivalence standard defined above.
     - **Categorical Conclusions:** Must be identical (e.g., "Positive" vs. "Positive"; "Category A" vs. "Category A").

**Strict Output Format**

1. **Reasons:** Provide a concise, objective, bulleted list explaining your rationale.

2. **Verdict:** State the final decision: True (for equivalent) or False (for inequivalent).

3. **Boxed:** Wrap the final boolean value in \\boxed{}, for example \\boxed{True} or \\boxed{False}.

**Texts to Evaluate:**

[**Text A**] {Text_A}

[**Text B**] {Text_B}

---

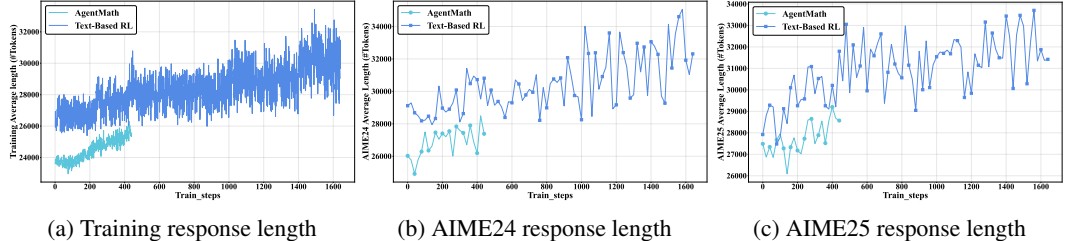

(a) Training response length  (b) AIME24 response length  (c) AIME25 response length

Figure 5: The evolution of sequence lengths for AgentMath and Text-Based model during RL training and on the AIME24 and AIME25. Both models started from their best SFT checkpoints trained on 20k data.

Table 5: Performance of different backbone models in SFT and RL stage on AIME24/25

| Models | AIME24 | AIME25 |
|---|---|---|
| Qwen3-1.7B-SFT-30w | 44.5% | 34.8% |
| Qwen3-1.7B-RL | 59.6% | 48.1% |
| Qwen3-8B-SFT-30w | 78.4% | 72.2% |
| Qwen3-8B-RL | 89.8% | 84.7% |
| Qwen3-30B-A3B-SFT-30w | 83.9% | 80.5% |
| Qwen3-30B-A3B-RL | 90.6% | 86.4% |

## A.7 Detailed analysis

### A.7.1 Tool-Augmented Synthetic Data vs. Text-Based Data

To assess the effectiveness of **AgentMath**, our proposed tool-augmented agent framework for complex mathematical reasoning, we conduct experiments addressing two key questions: (1) What performance advantages do tool-augmented synthetic data provide over text-only data in SFT phase? (2) Can tool augmentation enhance both model performance and training efficiency in RL phase? All experiments employ Qwen3-8B-Base as the backbone model with a maximum sequence length of 64k and a limit of 64 tool invocations in RL.

**Supervised Fine-Tuning Performance.** As shown in Table 2, when trained on an identical 20k SFT data, the tool-augmented model achieved accuracies of 60.5% on AIME24 and 53.3% on AIME25, surpassing the plain-text baseline (57.1% and 49.2%) by margins of 3.4% and 4.1%, respectively. This result confirms the effectiveness of our data synthesis method, which transforms computation-intensive reasoning steps into executable code.

**Agent RL Efficiency.** The benefits of tool augmentation are further amplified during RL. As detailed in Figure 4 and Table 2, the tool-augmented model required only approximately 400 training steps to improve from 60.5% to 76.2% on AIME24 and from 53.3% to 67.5% on AIME25. In contrast, the text-based model needed around 1,600 steps to reach 68.7% and 57.5%. This represents a 4.0× improvement in training efficiency. Notably, the tool-augmented model matched the final performance of the text-based model within just 100–200 steps, underscoring the advantage of dynamically interleaving natural language reasoning with code execution for accelerated policy optimization.

**Improved Inference Efficiency and Scalability.** As indicated in Figure 5, the tool-augmented model also demonstrated superior inference efficiency. During RL training and inference, its sequence length ranged from 24k to 29k tokens, compared to 28k–34k for the text-based model, with a reduction of roughly **4k tokens (∼14%)**. Furthermore, the growth in sequence length was significantly slower for the tool-augmented model as training progressed. These efficiency gains stem from precise tool-based computations replacing verbose and error-prone manual calculation steps.

In conclusion, **AgentMath**, by seamlessly integrating natural language reasoning with precise computational tools, demonstrates substantial improvements across all critical metrics (accuracy, training efficiency, and inference cost). These findings validate the effectiveness of both our tool-augmented data synthesis method and the agent-based RL framework.

Table 6: Efficiency evaluation of AgentMath RL training framework

| Method | Time per step (s) | Speedup |
|---|---|---|
| Static Batch Synchronous Rollout | 3600–4000 | – |
| + Request-Level Asynchronous Rollout | 2100–2500 | 1.5–1.8× |
| + Agentic Partial Rollout | 1100–1300 | 3.0–3.3× |
| + Prefix-Aware Weighted Load Balancing | 750–900 | 4.0–5.0× |

Table 7: Impact of the number of partial rollout segments ($N$) on training efficiency and model performance.

| Partial Rollout ($N$) | Time (100 steps) | AIME24 | AIME25 |
|---|---|---|---|
| Partial Rollout N = 1 | 62h | 70.5% | 60.5% |
| Partial Rollout N = 2 | 28h | 70.1% | 60.7% |
| Partial Rollout N = 4 | 22h | 70.8% | 60.7% |
| Partial Rollout N = 6 | 22h | 69.8% | 60.1% |
| Partial Rollout N = 8 | 23h | 69.5% | 60.5% |

### A.7.2 MULTI STAGE RL TRAINING

Following SFT, the model frequently produced responses exceeding 32k tokens on complex mathematical problems, with particularly challenging instances surpassing the 64k tokens. To balance training efficiency with model capacity, we developed an **adaptive, multi-stage reinforcement learning strategy**. This method progressively unlocks the model's potential by dynamically expanding the sequence length and tool-call budget. A truncation rate exceeding 10% for either response length or tool usage triggers an automatic budget increase: context length increases from 48k to 72k (step 120) and finally to 96k (step 280), while the tool-call limit expands from 48 to 72 (step 140) and then to 96 (step 320) as shown in Figure 3c and 3f.

As illustrated in Figure 3, the training progression reveals significant trends: As training progresses, generated trajectory lengths increase from 24k to 30k (Figure 3b), tool invocation frequency rises from 27 to 31 calls per problem (Figure 3e), and code utilization improved markedly from 70% to 95% (Figure 3d). These metrics indicate the model's growing proficiency in complex, multi-step reasoning and sophisticated tool use. Correspondingly, accuracy on the AIME24 benchmark rose from 78.4% to 89.8% (+11.4%), and on AIME25 72.2% to 84.7% (+12.5%), as shown in Figure 3a. Accuracy consistently improves following each capacity expansion. Crucially, the model exhibited emergent capabilities in self-correcting its generated code (Figure 9). These results confirm the efficacy of our multi-stage reinforcement learning strategy, which strikes an optimal balance between computational efficiency and model capability. Additionally, Table 5 details the performance of AgentMath with different backbones on AIME24 and AIME25. It shows that our approach brings significant enhancement in both SFT and RL stages, demonstrating the robustness and effectiveness of the data synthesis method and the multi-stage RL training strategy.

**Key Findings.** The experiments establish three critical insights: (1.) Expanded capacity is crucial: Increasing the sequence length and tool-call budget is essential for facilitating deeper reasoning chains. (2.) Effective reward shaping: The sustained growth in tool usage confirms that our composite reward function successfully guides the model's tool-call decisions. (3.) Framework robustness: Stable training under the extreme configuration of 96k tokens and 96 tool calls underscores the robustness of both the AgentMath framework and its asynchronous training infrastructure.

### A.7.3 SYNTHETIC DATA REFINEMENT AND SCALING LAW

Table 3 presents a comprehensive evaluation of AgentMath's data synthesis pipeline. The initial, unrefined synthetic data yielded suboptimal results (AIME24: 35.3%; AIME25: 25.7%), primarily due to formatting inconsistencies and non-executable code. By progressively applying multi-dimensional quality refinement, including format consistency, code executability verification, and environment feedback alignment, model performance improved substantially, achieving accuracies of 58.6% on AIME24 and 50.8% on AIME25. The subsequent integration of self-correction ca-

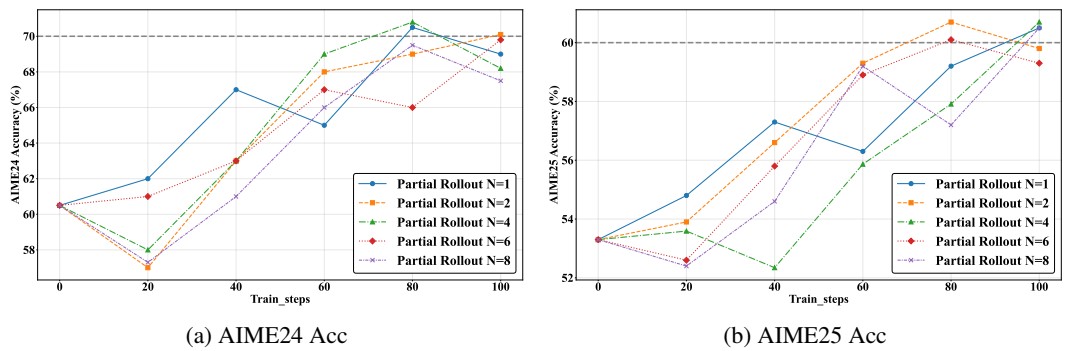

(a) AIME24 Acc

(b) AIME25 Acc

Figure 6: Exploring the performance impact of agent partial rollout segment count on AIME24/25, we adopt the Qwen3-8B-2w-SFT model as the RL initial point.

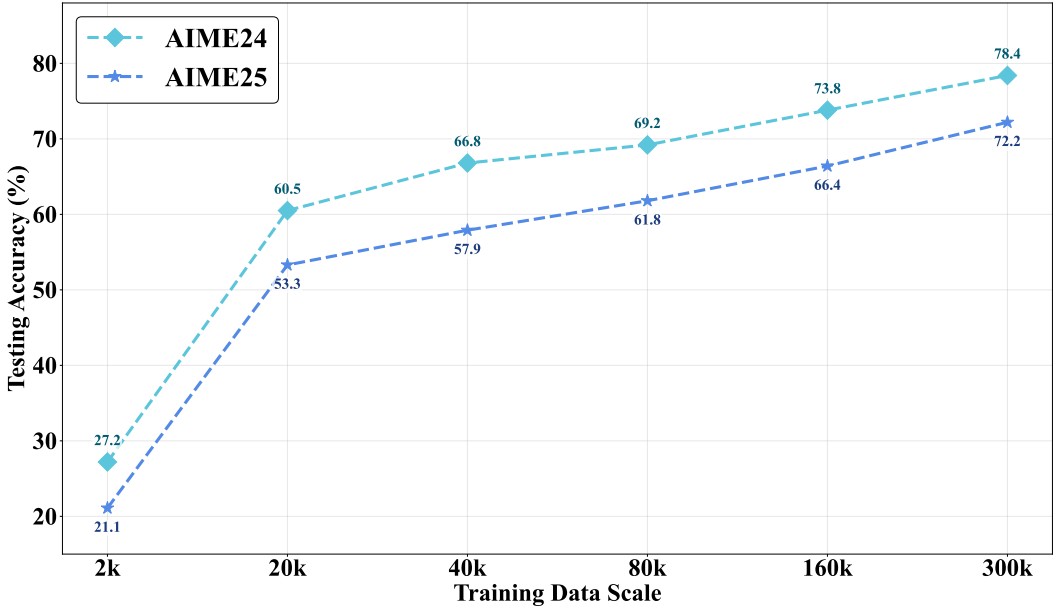

Figure 7: Exploring the scaling laws of tool-enhanced synthetic data, with performance evaluated on AIME24 and AIME25, using the Qwen3-8B-Base model as the backbone.

pabilities, combined with supervised fine-tuning using selective feedback masking based on code execution results, yielded final performance of 60.5% on AIME24 and 53.3% on AIME25. These results underscore the critical contribution of each refinement operation.

Building upon this validated data synthesis pipline, we further explored the impact of scaling the tool-augmented dataset, as shown in Figure 7. Scaling the dataset from 2k to 300k led to a performance increase from 27.2% to 78.4% on AIME24 and from 21.1% to 72.2% on AIME25, demonstrating the effective scalability of our approach. By combining rigorous quality control with effective scaling, AgentMath effectively alleviates the data scarcity in tool-augmented mathematical reasoning, laying a robust foundation for developing high-performance reasoning agents.

### A.7.4 EFFICIENCY OF AGENTMATH RL TRAINING FRAMEWORK

To alleviate the computational bottlenecks in agent reinforcement learning caused by ultra-long sequences and frequent tool use, we evaluated the efficiency of our AgentMath training framework. As shown in Table 6, a conventional static, batch-synchronous rollout approach required 3600–4000 s per training step. By introducing request-level asynchronous rollout scheduling, we cut this latency to 2100–2500 s (a $1.5$–$1.8\times$ speedup), mitigating head-of-line blocking from tool invocations. Incorporating agentic partial rollouts further reduced latency to 1100–1300 s (a $3.0$–$3.3\times$ speedup). Finally, adding prefix-aware weighted load balancing brought the per-step latency down to just 750–900 s, achieving a total $4.0$–$5.0\times$ speedup and demonstrating AgentMath's advantages for long-sequence, tool-interactive tasks.

We also investigated how the number of partial rollout segments ($N$) affects training efficiency. As shown in Table 7 and Figure 6, training 100 steps took 62 hours with $N = 1$, but this was reduced to 28 hours with $N = 2$ and 22 hours with $N = 4$. However, the benefits plateaued for $N \geq 6$ due to the scheduling overhead from excessive segmentation. Critically, these optimizations did not harm performance; the model maintained consistent accuracy of approximately $70\%$ on AIME24 and $60\%$ on AIME25 across all segmentation strategies. These results confirm that AgentMath effectively resolves the efficiency challenges of long-sequence agent RL, offering a scalable solution for scenarios that require extended sequences and intensive tool use.

### A.8 CASE STUDY

The following example illustrates the dynamic interaction between text reasoning and tool use in AgentMath's problem-solving process. Notably, the model also exhibits an emergent capability for code self-correction. Code blocks highlighted in red indicate an execution error.

### A.8.1 AGENTMATH CASE 1

### A.8.2 AGENTMATH CASE 2

### A.9 LLM USAGE STATEMENT:

LLM is employed solely for grammar checking and expression polishing to enhance the readability of the text.

**(AIME24-22) Question:** Let $ABC$ be a triangle inscribed in circle $\omega$. Let the tangents to $\omega$ at $B$ and $C$ intersect at point $D$, and let $\overline{AD}$ intersect $\omega$ at $P$. If $AB=5$, $BC=9$, and $AC=10$, $AP$ can be written as the form $\frac{m}{n}$, where $m$ and $n$ are relatively prime integers. Find $m + n$. **Ground Truth Answer: 113**

**AgentMath Response: I need to solve this geometry problem. We have a triangle ABC inscribed in a circle ω. Tangents to ω at points B and C intersect at point D. Line AD intersects the circle again at point P. Given the sides AB = 5, BC = 9, AC = 10, I need to find AP as a fraction m/n in simplest terms and then find m + n.\n\nFirst, ... So equations:\n\n1) 252x + 33 s y = 0\n\n2) 252x - 33 s y = 2268\n\nNow, add them together: (252x + 33s y) + (252x - 33s y) = 0 + 2268 => 504x = 2268 => x = 2268 / 504. Let\'s use code to simplify this fraction.\n \n**  \n```python\nfrom fractions import Fraction\n\n# We need to solve for x from the equation: 504x = 2268\n # The initial fraction for x is 2268/504.\n x_fraction = Fraction(2268, 504)\n \n print(f"The simplified value for x is: {x_fraction}")\n ```\n  \n \n <interpreter> \n The simplified value for x is: 9/2 \n\ n </interpreter>n \n So x = 9/2 = 4.5. Now from equation 1: 252x + 33 s y =0, so 33 s y = -252 x, y = -252 x / (33 s). Let\'s calculate this with code.\n \n  \n ```python \n from fractions import Fraction \n \n# We substitute x = 9/2 into the equation y = -252x / (33*s)\n# where s = sqrt(14).\n# We calculate the coefficient of 1/s.\n\ n coefficient = Fraction(-252) * Fraction(9, 2) / 33 \n \n print(f"The coefficient for 1/s is: {coefficient}")\n```\n\n\n<interpreter>\nThe coefficient for 1/s is: -378/11\n\n</interpreter>\\n\nSo y = -378 / (11 sqrt(14)). Rationalize: multiply numerator and denominator by sqrt(14): [-378 sqrt(14)] / (11 * 14) = -378 sqrt(14) / 154. Let\'s use code to simplify this fraction.\n\n  \n ```python \n from fractions import Fraction \n \n # We need to simplify the fraction -378/154. \n # The full expression for y is -378 * sqrt(14) / 154. \n simplified_fraction = Fraction(-378, 154) \n \n print(f"The simplified fraction is: {simplified_fraction}") \n ``` \n  \n \n \n <interpreter> \n The simplified fraction is: -27/11\n\n</interpreter>\n …Now, we can compute the squares and the denominator.\n\n\n```python\nimport math\n\n# Numerator parts\nval1 = 264**2\nval2 = 225**2\n\n# Denominator parts\nval3 = 22**2\nval4 = 157\nval5 = 107\n\n# Calculate the full numerator and denominator\nnumerator = val1 * val2\ndenominator = val3 * val4 * val5\nprint(f"264^2 = {val1}")\nprint(f"225^2 = {val2}")\nprint(f"22^2 = {val3}")\nprint(f"Numerator (264^2 * 225^2) = {numerator}")\nprint(f"Denominator (22^2 * 157 * 107) = {denominator}")\n```\n\n\n<interpreter>\n264^2 = 69696\n225^2 = 50625\n22^2 = 484\nNumerator (264^2 * 225^2) = 3528360000\nDenominator (22^2 * 157 * 107) = 8130716\n\n</interpreter>.... 100 and 13 coprime, yes.\n\nSo m/n = 100/13, m+n=113.\n\nBut let me confirm.\n\nIn the problem, sides are 5,9,10, which is almost isosceles, but not, and AP=100/13≈7.692, which is plausible.\n\nNow, to box the answer.\n\nSo m+n=100+13=113.\n\nBut let me make sure.\n\nSince we have AP, and it\'s rational, and matches.\n\nAnother way to confirm: using the formula.\n\nI recall that in some geometries, the length can be found using formulas, but I think it\'s correct.\n\nSo the answer is 113.**

Figure 8: AgentMath Case study

**(AIME24-29) Question:** Let $\omega \neq 1$ be a 13th root of unity. Find the remainder when $\prod_{k=0}^{12}(2-2\omega^k+\omega^{2k})$ is divided by 1000.

**Ground Truth Answer: 321**

**AgentMath Response: The problem states: Let ω ≠ 1 be a 13th root of unity. I need to find the remainder when the product from k=0 to 12 of (2 - 2ω^k + ω^{2k}) is divided by 1000. Since it\'s a product over all 13th roots of unity except 1, but k goes from 0 to 12, which includes all roots. ω is a primitive 13th root of unity, I assume, since it\'s specified ≠1, but k=0 gives ω^0=1, so it includes k=0. \n \n Let me denote the expression inside as f(k) = 2 - 2ω^k + ω^{2k}. ...... \n Now, let me compute the product numerically to check. \n \n  \n ```python \n import cmath \n import math \n \n # Let omega be the primitive 5th root of unity, omega = exp(2\*pi\*i / 5) \n omega = cmath.exp(2 \* cmath.pi \* 1j / 5) \n \n # Calculate the five terms g(omega^k) \n g_0 = (omega\*\*0)\*\*2 - 2 \* (omega\*\*0) + 2 \n g_1 = omega\*\*2 - 2 \* omega + 2 \n g_2 = omega\*\*4 - 2 \* omega\*\*2 + 2 \n g_3 = omega\*\*6 - 2 \* omega\*\*3 + 2 \n g_4 = omega\*\*8 - 2 \* omega\*\*4 + 2 \n \n # Calculate the product of the five terms \n total_product = g_0 \* g_1 \* g_2 \* g_3 \* g_4 \n \n # Print the individual terms and the final product \n # Using round to get cleaner numbers as the imaginary parts might be very small \n print(f"g(1) = {round(g_0, 5)}") \n print(f"g(omega) = {round(g_1, 5)}") \n print(f"g(omega^2) = {round(g_2, 5)}") \n print(f"g(omega^3) = {round(g_3, 5)}") \n print(f"g(omega^4) = {round(g_4, 5)}") \n print(f"Total product = {round(total_product, 5)}") \n ``` \n ** \n\n \<interpreter\> \n Traceback (most recent call last): \n    File "/tmp/tmpwjpzt94l/tmp2mcwv09t.py", line 19, in \<module\> \n print(f"g(1) = {round(g_0, 5)}") \n    ^^^^^^^^^^^^ \n **TypeError: type complex doesn\'t define \_\_round\_\_ method** \n    \n \</interpreter\> \n \n Oops, the code above appears to be throwing an error. I need to fix this to ensure it runs successfully.< The error occurs because the `round()` function cannot handle complex numbers directly. I will modify the code to handle complex numbers correctly. \n \n \<code\> \n ```python \n import cmath \n import math \n \n # Let omega be the primitive 5th root of unity, omega = exp(2\*pi\*i / 5) \n omega = cmath.exp(2 \* cmath.pi \* 1j / 5) \n \n # Calculate the five terms g(omega^k) \n g_0 = (omega\*\*0)\*\*2 - 2 \* (omega\*\*0) + 2 \n g_1 = omega\*\*2 - 2 \* omega + 2 \n g_2 = omega\*\*4 - 2 \* omega\*\*2 + 2 \n g_3 = omega\*\*6 - 2 \* omega\*\*3 + 2 \n g_4 = omega\*\*8 - 2 \* omega\*\*4 + 2 \n \n # Calculate the product of the five terms \n total_product = g_0 \* g_1 \* g_2 \* g_3 \* g_4 \n \n # Print the individual terms and the final product \n print(f"g(1) = {g_0.real:.5f} {g_0.imag:+.5f}j") \n print(f"g(omega) = {g_1.real:.5f} {g_1.imag:+.5f}j") \n print(f"g(omega^2) = {g_2.real:.5f} {g_2.imag:+.5f}j") \n print(f"g(omega^3) = {g_3.real:.5f} {g_3.imag:+.5f}j") \n print(f"g(omega^4) = {g_4.real:.5f} {g_4.imag:+.5f}j") \n print(f"Total product = {total_product.real:.5f} {total_product.imag:+.5f}j") \n ``` \n \</code\> \n \<interpreter\>\ng(1) = 1.00000+0.00000j \n g(omega) = 0.57295 - 1.31433j \n g(omega^2) = 3.92705 - 2.12663j \n g(omega^3) = 3.92705+2.12663j \n g(omega^4) = 0.57295+1.31433j \n Total product = 41.00000-0.00000j \n    \n \</interpreter\> **\n\n The numerical calculation shows the total product is 41. The small imaginary part is due to floating-point inaccuracies… So here f=g(x), so f2=1, f1=-2, f0=2 \n\n g=r(x), so g1=-64, g0=-1 \n\n ......Thus P=8321. \n\n Now, 8321 mod 1000 is 321. \n\n So the remainder is 321.**

Figure 9: AgentMath Case study For code self-correction

## A.10 Motivation and Core Contributions

While agentic RL training—including coding-assisted math reasoning—has been explored in prior works such as ReTool and ToRL, the field currently faces two critical challenges that remain inadequately addressed.

**Challenge 1: Lack of efficient tool-augmented data synthesis methods and scaling-law studies for long-chain reasoning.** Existing approaches face substantial limitations when synthesizing tool-augmented data in long-chain reasoning scenarios. For instance, CoRT relies on manual annotation, making it costly and difficult to scale. While ReTool employs LLM-based synthesis, it produces only 2k samples and provides insufficient details about the synthesis process. Moreover, existing research lacks systematic investigation of scaling laws for agent-synthesized data: in code-assisted mathematical reasoning, what is the relationship between data scale and performance, and can superior performance be achieved through data scaling alone? These fundamental questions have rarely been systematically studied in the open-source community.

**Challenge 2: Agent RL training encounters severe efficiency bottlenecks in ultra-long sequence and large-scale tool invocation scenarios.** Even when models perform well on code-assisted mathematical reasoning after supervised fine-tuning, it remains an open question whether performance can be continuously improved through large-scale Agent RL to achieve adaptive tool invocation. Under conditions involving ultra-long reasoning chains and large-scale tool invocations (*e.g.*, 96k token length, 96 tool calls), training efficiency becomes a critical bottleneck. Frequent environment interactions and ultra-long trajectory generation cause traditional batch-synchronous rollout training on Qwen3-8B to require 3,600–4,000 seconds per training step, with rollout generation accounting for 70%–80% of total training time. This inefficiency severely constrains researchers' exploration of Agent RL training in long-horizon, large-scale tool-calling scenarios. We note that training efficiency is crucial for RL training: the ability to train for thousands of steps and break through performance plateaus directly impacts final results. For instance, DeepSeek-R1-Zero requires 4,000–8,000 training steps to observe the model's "Aha moment" phenomenon and continuous performance improvements.

Furthermore, while leading proprietary models (e.g., GPT-5, Gemini 3.0, Claude 4.5) demonstrate exceptional performance on agent tasks such as code-assisted mathematical reasoning, the lack of technical details and training methods hinders the open-source community's ability to reproduce and advance this research. By providing a clear, transparent, and complete pipeline for data synthesis and large-scale Agent RL training that achieves competitive performance with proprietary models (*e.g.*, Gemini 2.5-Pro, OpenAI-o3), our work aims to offer the open-source community a reproducible technical roadmap, thereby advancing the development of the Agent LLM field.

To address these challenges, we highlight four core technical contributions.

### A.10.1 Efficient Tool-Augmented Data Synthesis Method

**Segmented synthesis with focused tool invocation.** We observe that previous methods (e.g., ReTool) suffer from *tool overuse* issues, including invoking tools for simple calculations and performing redundant verification of already-verified correct results, leading to an increase in ineffective tokens. To address this, AgentMath focuses on applying tool invocations to genuinely necessary complex computational scenarios (e.g., solving complex equations, performing large-number arithmetic, and handling advanced linear algebra and calculus operations). We propose a *segmented synthesis strategy* that partitions DeepSeek-R1's long chain-of-thought process into fixed-length segments (e.g., 3K tokens) and transforms them into tool-augmented chains-of-thought, incorporating automated multi-dimensional quality refinement to ensure high data quality.

Under identical training settings—using the same 2k synthetic data problems from ReTool's official open-source release, the same DeepSeek-R1 teacher model responses, the same base model (Qwen2.5-32B-Instruct), and the same number of RL training steps (400 steps)—AgentMath's synthesis method significantly outperforms ReTool's original approach. As shown in Table 8, AgentMath-32B-SFT-2k achieves 44.1% on AIME24 and 37.3% on AIME25, outperforming ReTool-32B-SFT-2k (40.9% and 34.5%). After RL training, AgentMath-32B-RL reaches 74.8% on AIME24 and 56.6% on AIME25, similarly outperforming ReTool-32B-RL (67.0% and 49.3%).

These results validate the effectiveness of our data synthesis method and the superior quality of the synthesized data.

Table 8: Performance comparison between AgentMath and ReTool on AIME24 and AIME25 under identical training configurations, demonstrating the superiority of AgentMath's tool-augmented data synthesis method in both SFT and RL stages.

| Model | AIME24 Acc | AIME25 Acc |
|---|---|---|
| ReTool-32B-SFT-2k | 40.9% | 34.5% |
| AgentMath-32B-SFT-2k | 44.1% | 37.3% |
| ReTool-32B-RL | 67.0% | 49.3% |
| AgentMath-32B-RL | 74.8% | 56.6% |

**Multi-dimensional quality refinement pipeline.** We further emphasize the multi-dimensional quality refinement pipeline in our data synthesis process, which includes correcting format inconsistencies, excluding samples with failed code execution, performing environmental feedback alignment, filtering low-complexity code (e.g., $\leq 5$ lines), and injecting self-correction capabilities. As shown in Table 9, through this pipeline, accuracy on AIME24 improves from 35.3% with initially unrefined synthetic data to 60.5%, and on AIME25 from 25.7% to 53.3%, substantially demonstrating the critical role and effectiveness of each refinement module in optimizing data quality. We are committed to maintaining transparency and clarity in our synthesis and refinement procedures, whereas some recent works provide limited details on this crucial aspect, thereby posing challenges for research reproducibility.

Table 9: Progressive improvement in model accuracy on AIME24 and AIME25 through the multi-dimensional quality refinement pipeline. Each refinement step is applied cumulatively, demonstrating substantial accuracy gains from 35.3% to 60.5% on AIME24 and from 25.7% to 53.3% on AIME25.

| Refinement Steps | AIME24 | AIME25 |
|---|---|---|
| Initial Unrefined CI-Synthetic Data (20k) | 35.3% | 25.7% |
| + Format consistency correction | 47.4% | 40.1% |
| + Code executability verification | 52.8% | 44.8% |
| + Environmental feedback alignment | 56.3% | 48.3% |
| + Tool-usage rationality assessment | 57.2% | 48.9% |
| + Self-correction capability injection | 58.6% | 50.8% |
| + SFT with selective feedback masking | 60.5% | 53.3% |

### A.10.2 SCALING LAWS FOR TOOL-AUGMENTED SYNTHETIC DATA

We systematically validate the scaling laws associated with tool-augmented synthetic data. Through AgentMath's segmented data synthesis strategy and multi-dimensional quality refinement, we progressively scale the dataset from 2k to 300k samples. Fine-tuning the Qwen3-8B-Base model, as shown in Table 10, we observe consistent performance improvements: accuracy on AIME24 increases from 27.2% to 78.4%, and on AIME25 from 21.1% to 72.2%. Performance continues to improve with data scale, ultimately achieving excellent results. These findings robustly demonstrate that scaling laws remain effective for tool-augmented synthetic data. In contrast, some recent studies (e.g., ReTool) utilize only limited data (e.g., 2k samples) and do not sufficiently explore the potential benefits of data scaling.

### A.10.3 EFFICIENT AGENT RL TRAINING FOR ULTRA-LONG SEQUENCES AND LARGE-SCALE TOOL CALLING

In Agent RL, the combination of long-context generation and frequent external tool interactions creates heterogeneous computational workloads, posing significant challenges to training efficiency. To address this, we design targeted optimization strategies comprising three core improvements:

Table 10: Data scaling law of AgentMath-8B models as a function of SFT training data volume, demonstrating consistent improvements from 2k to 300k samples on AIME24, AIME25, and Google-IMO-AnswerBench.

| SFT Training Data Volume | AIME24 | AIME25 | Google-IMO-AnswerBench | Avg Score |
|---|---|---|---|---|
| AgentMath-8B-SFT-2k | 27.2 | 21.1 | 5.3 | 17.9 |
| AgentMath-8B-SFT-20k | 60.5 | 53.3 | 12.8 | 42.2 |
| AgentMath-8B-SFT-40k | 66.8 | 57.9 | 17.6 | 47.4 |
| AgentMath-8B-SFT-80k | 69.2 | 61.8 | 21.4 | 50.8 |
| AgentMath-8B-SFT-160k | 73.8 | 66.4 | 25.6 | 55.3 |
| AgentMath-8B-SFT-300k | 78.4 | 72.2 | 28.8 | 59.8 |

1. **Request-Level Asynchronous Rollout Scheduling.** We replace the conventional static batch synchronous processing with a coroutine-driven, request-level asynchronous scheduler. Each trajectory rollout is treated as an independent long-running request, where the inference engine (server) and the agent (client) are fully decoupled through asynchronous communication. This effectively mitigates the efficiency bottleneck caused by synchronous waiting.

2. **Agentic Partial Rollout.** To alleviate the long-tail latency issues arising from ultra-long sequences and large-scale tool calls, we propose an agentic partial rollout mechanism, including *Length Partial Rollout* and *Tool Partial Rollout*. This mechanism decomposes each trajectory $\tau$ into budget-constrained segments, where each segment is limited to a fixed maximum sequence length (e.g., 32k tokens) and a fixed maximum number of tool calls (e.g., 32). Incomplete sequences or tool calls are carried over to subsequent batches for continued execution, while completed trajectories immediately participate in training, thereby significantly improving training efficiency.

3. **Prefix-Aware Weighted Load Balancing.** Since Partial Rollout introduces ultra-long sequences, leading to a substantial increase in KV cache memory consumption and prefill computational overhead, we design a prefix-aware weighted load balancing strategy. This strategy dynamically assigns weights based on the prefix length of requests and intelligently routes them to the least-loaded inference engine instances, effectively alleviating memory and computational pressure.

We systematically evaluate the efficiency improvements of the AgentMath training framework. As shown in Table 11, the traditional static batch synchronous rollout method requires 3,600–4,000 seconds per training step. With the introduction of request-level asynchronous rollout scheduling, latency is reduced to 2,100–2,500 seconds (a 1.5–1.8× speedup). Further incorporating the Agentic Partial Rollout mechanism reduces latency to 1,100–1,300 seconds (a 3.0–3.3× speedup). Finally, with the incorporation of prefix-aware weighted load balancing, the per-step latency drops to 750–900 seconds, achieving an overall training speedup of 4.0–5.0×. This framework enables efficient Agent RL training in extreme scenarios (e.g., 96k sequence length, 96 tool calls), for which training with such ultra-long sequences and large-scale tool calls has few precedents in the open-source community.

Table 11: Progressive training efficiency improvements in the AgentMath framework through cumulative optimization strategies, demonstrating up to 5.0× speedup over baseline static batch synchronous rollout.

| Method | Time per step (s) | Speedup |
|---|---|---|
| Static Batch Synchronous Rollout | 3,600–4,000 | — |
| + Request-Level Asynchronous Rollout | 2,100–2,500 | 1.5–1.8× |
| + Agentic Partial Rollout | 1,100–1,300 | 3.0–3.3× |
| + Prefix-Aware Weighted Load Balancing | 750–900 | 4.0–5.0× |

A.10.4 STATE-OF-THE-ART PERFORMANCE VIA LARGE-SCALE AGENT RL

AgentMath undergoes a three-stage curriculum learning process, with sequence lengths progressively increasing from 48k to 72k to 96k tokens and tool invocations scaling from 48 to 72 to 96. Across diverse model backbones spanning 1.5B to 32B parameters, we successfully conduct stable Agent RL training on ultra-long sequences with large-scale tool invocations, achieving consistent performance improvements. As detailed in Table 12, we achieve state-of-the-art performance on challenging mathematical competition benchmarks including AIME24, AIME25, and HMMT25, significantly outperforming leading open-source models of comparable size. Specifically, AgentMath-30B-A3B achieves accuracies of 90.6%, 86.4%, and 73.8% on AIME24, AIME25, and HMMT25 respectively, surpassing OpenAI-o3-mini and Claude-Opus-4.0-Thinking while remaining competitive with OpenAI-o3, Gemini-2.5-Pro, and DeepSeek-R1-671B-0528. These results validate the effectiveness and scalability of our data synthesis method and Agent RL training strategy, paving the way for building more sophisticated and scalable mathematical reasoning agents.

Table 12: Performance comparison of AgentMath against proprietary and frontier open-source models on AIME24, AIME25, and HMMT25. AgentMath models are highlighted in blue.

| Models | AIME24 | AIME25 | HMMT25 |
|---|---|---|---|
| **Proprietary Models** | | | |
| OpenAI-o4-mini-w/tools | 98.7 | 99.5 | - |
| Gemini-2.5-Pro | 92.0 | 88.0 | 82.5 |
| OpenAI-o3 | 91.6 | 88.9 | 77.5 |
| OpenAI-o3-mini | 87.3 | 86.3 | 53.0 |
| Claude-Opus-4.0-Thinking | 83.0 | 72.0 | 58.3 |
| **Frontier Models (1B–2B)** | | | |
| DeepSeek-R1-Distill-Qwen-1.5B | 28.8 | 21.8 | 15.3 |
| Qwen3-1.7B Thinking | 52.0 | 35.3 | 23.3 |
| OpenReasoning-Nemotron-1.5B | 55.5 | 45.6 | 31.5 |
| AgentMath-1.7B | 59.6 | 48.1 | 40.2 |
| **Frontier Models (7B–8B)** | | | |
| DeepSeek-R1-Distill-Qwen-7B | 55.0 | 39.7 | - |
| Qwen3-8B Thinking | 76.0 | 67.3 | 44.7 |
| OpenReasoning-Nemotron-7B | 84.7 | 78.2 | 63.5 |
| DeepSeek-R1-0528-Qwen3-8B | 86.0 | 76.3 | 61.5 |
| AgentMath-8B | 89.8 | 84.7 | 71.3 |
| **Frontier Models (30B–32B)** | | | |
| ReTool-32B | 67.0 | 49.3 | - |
| DeepSeek-R1-Distill-Qwen-32B | 72.9 | 59.0 | 33.0 |
| Qwen3-30B-A3B-Thinking-2507 | 87.7 | 85.0 | 71.4 |
| OpenReasoning-Nemotron-32B | 89.2 | 84.0 | 73.8 |
| AgentMath-30B-A3B | 90.6 | 86.4 | 73.8 |
| **Frontier Models (>32B)** | | | |
| DeepSeek-R1-671B-0528 | 91.4 | 87.5 | 77.0 |
| Qwen3-235B-A22B-Thinking-2507 | 94.2 | 92.3 | 83.9 |
| AgentMath-235B-A22B-SFT | 93.4 | 90.8 | 81.7 |

**Summary of contributions.** In summary, our core contributions are as follows:

1. We propose an efficient tool-augmented data synthesis method that focuses on applying tool invocation to genuinely demanding complex computational scenarios (*e.g.*, solving complex equations, large-number arithmetic, advanced linear algebra, and calculus). By combining a segmented synthesis strategy with a multi-dimensional quality refinement

mechanism, AgentMath significantly outperforms ReTool in both SFT and RL stages on AIME24 and AIME25 under identical training configurations.

2. We systematically validate the scaling laws associated with tool-augmented synthetic data, scaling the dataset from 2K to 300K samples. Building on the Qwen3-8B-Base model, accuracy on AIME24 improves from 27.2% to 78.4%, and on AIME25 from 21.1% to 72.2%, demonstrating consistent performance gains with increased data scale.

3. To alleviate the training efficiency bottleneck in Agent RL under ultra-long sequence and large-scale tool invocation scenarios, we introduce request-level asynchronous rollout scheduling, propose Agentic Partial Rollout, and design prefix-aware weighted load balancing. Through these technical innovations, our framework achieves a 4–5× speedup in training efficiency, reducing the time per training step from 3,600–4,000 seconds to 750–900 seconds, thereby enabling efficient Agent RL training in extreme scenarios (*e.g.*, 96K sequence length with 96 tool invocations). To the best of our knowledge, such exploration of large-scale Agent RL training with extremely long sequences and massive tool invocations is rarely conducted in the open-source community.

4. Through three-stage curriculum Agent RL training, AgentMath achieves state-of-the-art performance on challenging mathematical competition benchmarks including AIME24, AIME25, and HMMT25 across model backbones ranging from 1.5B to 32B parameters, significantly outperforming leading open-source models of comparable size. Notably, AgentMath-30B-A3B surpasses OpenAI-o3-mini while remaining competitive with Gemini-2.5-Pro and DeepSeek-R1-671B-0528. We are committed to open-sourcing our code, data synthesis pipeline, and model training workflow to advance the LLM reasoning community.

## A.11 RELIABILITY OF EVALUATION BENCHMARKS

The math benchmarks used in our main evaluation (AIME24, AIME25, HMMT25) each contain only about 30 questions, and prior work has raised concerns that these datasets may have leaked into open-source model training corpora. To systematically validate the reliability of our results and mitigate potential evaluation benchmark leakage concerns, we conducted supplementary experiments and analyses from three complementary perspectives.

### A.11.1 IMO-ANSWERBENCH: A LATEST HIGH-DIFFICULTY BENCHMARK

Considering that datasets such as AIME24, AIME25, and HMMT25 may have partially appeared in the pre-training corpora of open-source models, we evaluate AgentMath on IMO-AnswerBench (Luong et al., 2025), a recent mathematics olympiad benchmark released by Google DeepMind in November 2025. This benchmark comprises 400 carefully curated problems spanning four domains—algebra, combinatorics, geometry, and number theory—with 100 problems per category, all sourced from national, regional, and international olympiad competitions.

To mitigate data memorization and leakage, IMO-AnswerBench systematically rewrites original competition problems through several strategies: renaming points or lines in geometry problems, rephrasing problem statements, modifying numerical values and/or introducing distractors, and reformulating problems using entirely different yet semantically equivalent expressions. This design substantially reduces the probability of verbatim matches in pre-training corpora and minimizes the potential for memorization-based advantages, thereby enhancing evaluation robustness and fairness.

The performance of current frontier models on this benchmark—DeepSeek-V3 at 37.0%, Qwen3-235B at 53.8%, and DeepSeek-R1 at 60.8%—indicates that the benchmark possesses sufficient discriminative power and challenge. Moreover, the benchmark's release date (November 2025) is notably later than the public release of Qwen3-series models (May 2025), further minimizing the likelihood of contamination in their training corpora.

Following the experimental setup specified in the IMO-AnswerBench paper, we report results averaged over 8 independent runs. As detailed in Table 13, AgentMath achieves strong performance at comparable parameter scales across all four categories:

• **At the 1.5B–1.7B scale:** AgentMath-Qwen3-1.7B achieves 20.3%, outperforming OpenReasoning-Nemotron-1.5B (17.8%).

- **At the 7B–8B scale:** AgentMath-Qwen2.5-7B achieves 35.1% and AgentMath-Qwen3-8B achieves 37.9%, both substantially outperforming OpenReasoning-Nemotron-7B (33.0%).

- **At the 30B–32B scale:** AgentMath-30B-A3B achieves 51.2%, surpassing Qwen3-30B-A3B-Thinking-2507 (41.4%), Claude Sonnet 4, DeepSeek-V3, and Kimi-K2-Instruct, while approaching the performance of Qwen3-235B.

- **At scales larger than 32B:** AgentMath-235B-A22B-SFT achieves 55.4%, surpassing Qwen3-235B and approaching DeepSeek-R1.

These results demonstrate that AgentMath maintains strong performance and robustness across different parameter scales on a challenging evaluation set that is demonstrably free from data leakage concerns.

Table 13: Performance of AgentMath on Google-IMO-AnswerBench, a recent mathematics olympiad benchmark released by Google DeepMind in November 2025. Our model (highlighted in blue) is compared against other leading models, with accuracy (avg@8) as the evaluation metric.

| Model (Google-IMO-AnswerBench) | Algebra | Combinatorics | Geometry | Number Theory | Avg Score |
|---|---|---|---|---|---|
| *Proprietary Models* | | | | | |
| Gemini Deep Think (IMO Gold) | 85.0% | 69.0% | 88.0% | 78.0% | 80.0% |
| Grok 4 | 75.5% | 55.9% | 80.1% | 80.9% | 73.1% |
| Gemini 2.5 Deep Think | 78.0% | 49.0% | 83.0% | 77.0% | 71.8% |
| Gemini 2.5 Pro | 73.4% | 48.0% | 74.3% | 77.1% | 68.2% |
| o4-mini (high reasoning) | 71.3% | 46.6% | 78.4% | 75.3% | 67.9% |
| GPT-5 | 69.9% | 46.4% | 74.8% | 71.2% | 65.6% |
| o3 | 62.8% | 43.0% | 70.6% | 68.0% | 61.1% |
| Claude Sonnet 4 | 20.6% | 17.8% | 26.0% | 27.6% | 23.0% |
| Claude Opus 4 | 19.4% | 20.0% | 23.3% | 26.6% | 22.3% |
| *Frontier Models (1B–2B)* | | | | | |
| DeepSeek-R1-Distill-Qwen-1.5B | 5.5% | 9.9% | 14.5% | 8.7% | 9.7% |
| Qwen3-1.7B Thinking | 11.4% | 12.5% | 21.5% | 17.5% | 15.7% |
| OpenReasoning-Nemotron-1.5B | 16.3% | 14.0% | 22.6% | 18.4% | 17.8% |
| AgentMath-Qwen3-1.7B | 18.6% | 16.2% | 24.4% | 21.8% | 20.3% |
| *Frontier Models (7B–8B)* | | | | | |
| DeepSeek-R1-Distill-Llama3.1-8B | 10.2% | 17.4% | 21.6% | 17.0% | 16.6% |
| DeepSeek-R1-Distill-Qwen-7B | 12.8% | 20.6% | 22.8% | 18.0% | 18.5% |
| Qwen3-8B-Thinking | 17.0% | 26.8% | 38.0% | 25.8% | 26.9% |
| DeepSeek-R1-0528-Distill-Qwen3-8B | 21.2% | 31.7% | 43.3% | 27.3% | 30.9% |
| OpenReasoning-Nemotron-7B | 23.6% | 35.2% | 43.2% | 29.9% | 33.0% |
| AgentMath-Qwen2.5-7B | 24.9% | 37.4% | 45.6% | 32.5% | 35.1% |
| AgentMath-Qwen3-8B | 28.3% | 39.8% | 49.5% | 33.8% | 37.9% |
| *Frontier Models (30B–32B)* | | | | | |
| DeepSeek-R1-Distill-Qwen-32B | 17.2% | 25.1% | 33.0% | 22.5% | 24.4% |
| Qwen3-30B-A3B-Instruct-2507 | 27.8% | 29.9% | 36.2% | 27.4% | 30.3% |
| AM-Thinking-v1-32B | 27.9% | 36.2% | 50.1% | 29.2% | 35.8% |
| Qwen3-30B-A3B-Thinking-2507 | 34.7% | 44.3% | 53.2% | 33.3% | 41.4% |
| AgentMath-30B-A3B | 43.4% | 40.0% | 69.6% | 51.9% | 51.2% |
| *Frontier Models (¿32B)* | | | | | |
| DeepSeek V3 | 39.0% | 26.0% | 35.0% | 48.0% | 37.0% |
| Kimi-K2-Instruct | 45.6% | 31.1% | 49.3% | 56.9% | 45.8% |
| Qwen3-235B | 57.6% | 37.5% | 57.6% | 62.3% | 53.8% |
| AgentMath-235B-A22B-SFT | 49.4% | 43.2% | 73.5% | 55.3% | 55.4% |
| DeepSeek R1 | 65.0% | 40.0% | 73.0% | 65.0% | 60.8% |

### A.11.2 VALIDATION USING BASE MODELS RELEASED PRIOR TO BENCHMARK PUBLICATION

To further mitigate concerns regarding potential evaluation data leakage into the pre-training corpus, we employ open-source base models released before the publication of the evaluation benchmarks for our SFT and RL training. The relevant timeline is as follows:

- Qwen2.5-7B-Base release: September 2024
- Llama3.1-8B-Base release: July 2024
- AIME25 publication: February 2025
- HMMT25 publication: February 2025

- Google-IMO-AnswerBench publication: November 2025

All benchmarks were published after model pre-training was completed, thereby precluding any possibility of data contamination. As shown in Tables 14 and 15, AgentMath trained on these earlier base models achieves superior performance across multiple benchmarks:

- Building on Llama3.1-8B-Base, AgentMath-Llama-8B achieves 66.2% and 53.1% on AIME25 and HMMT25, respectively, significantly outperforming Llama3.1-Nemotron-Nano-8B-v1 (48.0% and 26.7%).
- Building on Qwen2.5-7B-Base, AgentMath-Qwen2.5-7B attains 79.8% and 65.9% on AIME25 and HMMT25, respectively, surpassing OpenReasoning-Nemotron-7B (78.2% and 63.5%).
- On the more challenging Google-IMO-AnswerBench, AgentMath-Llama-8B achieves 25.2%, outperforming DeepSeek-R1-Distill-Llama3.1-8B (16.6%), while AgentMath-Qwen2.5-7B (35.1%) also exceeds OpenReasoning-Nemotron-7B (33.0%).

By leveraging base models released in 2024 (Qwen2.5-7B-Base and Llama3.1-8B-Base), Agent-Math demonstrates consistent and leading performance across multiple mathematical reasoning benchmarks released in 2025 (AIME25, HMMT25, and IMO-AnswerBench). These results not only eliminate data contamination risks but also establish the robustness and reliability of our approach.

Table 14: Performance of AgentMath on AIME25 and HMMT25 based on Llama3.1-8B-Base and Qwen2.5-7B-Base. Our model (highlighted in blue) is compared against other leading models, with accuracy (avg@32) as the evaluation metric.

| Models | Base Model | Tool Use | AIME25 | HMMT25 |
|---|---|---|---|---|
| *Based on Llama3.1-8B-Base/Instruct* | | | | |
| DeepSeek-R1-Distill-Llama-8B | Llama-3.1-8B-Base | ✗ | 28.7 | 13.8 |
| Llama3.1-Nemotron-Nano-8B-v1 | Llama-3.1-8B-Instruct | ✗ | 48.0 | 26.7 |
| AgentMath-Llama-8B | Llama-3.1-8B-Base | ✓ | 66.2 | 53.1 |
| *Based on Qwen2.5-7B-Base/Instruct* | | | | |
| ZeroTIR-7B | Qwen-2.5-7B-Base | ✓ | 30.0 | 22.5 |
| SimpleTIR-7B | Qwen2.5-7B-Base | ✓ | 30.9 | 29.7 |
| DeepSeek-R1-Distill-Qwen-7B | Qwen2.5-Math-7B-Base | ✗ | 39.7 | 16.3 |
| OpenR1-Distill-7B | Qwen2.5-Math-7B-Base | ✗ | 39.7 | 25.7 |
| Light-R1-7B-DS | DeepSeek-R1-Distill-Qwen-7B | ✗ | 44.3 | 27.6 |
| AReal-boba-7B | DeepSeek-R1-Distill-Qwen-7B | ✗ | 48.3 | 29.4 |
| Skywork-OR1-7B | DeepSeek-R1-Distilled-Qwen-7B | ✗ | 54.6 | 35.7 |
| AceReason-Nemotron-1.1-7B | DeepSeek-R1-Distill-Qwen-7B | ✗ | 64.8 | 42.9 |
| OpenReasoning-Nemotron-7B | Qwen2.5-7B-Instruct | ✗ | 78.2 | 63.5 |
| AgentMath-Qwen2.5-7B | Qwen2.5-7B-Base | ✓ | 79.8 | 65.9 |

Table 15: Performance of AgentMath on Google-IMO-AnswerBench based on Llama3.1-8B-Base and Qwen2.5-7B-Base. Our model (highlighted in blue) is compared against other leading models, with accuracy (avg@8) as the evaluation metric.

| Model (Google-IMO-AnswerBench) | Base Model | Algebra | Combinatorics | Geometry | Number Theory | Avg Score |
|---|---|---|---|---|---|---|
| *Based on Llama3.1-8B-Base* | | | | | | |
| DeepSeek-R1-Distill-Llama3.1-8B | Llama3.1-8B-Base | 10.2% | 17.4% | 21.6% | 17.0% | 16.6% |
| AgentMath-Llama-8B | Llama3.1-8B-Base | 18.8% | 24.6% | 33.7% | 23.7% | 25.2% |
| *Based on Qwen2.5-7B-Base/Instruct* | | | | | | |
| OpenReasoning-Nemotron-7B | Qwen2.5-7B-Instruct | 23.6% | 35.2% | 43.2% | 29.9% | 33.0% |
| AgentMath-Qwen2.5-7B | Qwen2.5-7B-Base | 24.9% | 37.4% | 45.6% | 32.5% | 35.1% |

### A.11.3 SYSTEMATIC COMPARISON OF SFT AND RL CONTRIBUTIONS

To rigorously evaluate the independent contributions of SFT and RL in AgentMath while mitigating potential data contamination, we conduct systematic comparisons exclusively on benchmarks confirmed to be released after the pre-training cutoff date of their corresponding backbone models, as shown in Tables 16 and 17:

- For Llama3.1-8B-Base and Qwen2.5-7B-Base, we report results on AIME25, HMMT25, and Google-IMO-AnswerBench.

- For Qwen3-1.7B-Base, Qwen3-8B-Base, and Qwen3-30B-A3B-Instruct-2507, we report results on Google-IMO-AnswerBench only.

The results reveal consistent improvements across all base models. Based on Llama3.1-8B-Base, AgentMath-Llama-8B-SFT achieves an average score of 39.2% across the three benchmarks, while AgentMath-Llama-8B-RL reaches 48.2%. Based on Qwen2.5-7B-Base, AgentMath-Qwen2.5-7B-SFT attains 49.6%, with AgentMath-Qwen2.5-7B-RL achieving 60.3%.

On the Qwen3 series, we observe similar trends on Google-IMO-AnswerBench: based on Qwen3-1.7B-Base, AgentMath-Qwen3-1.7B-SFT scores 11.7% while the RL variant scores 20.3%; based on Qwen3-8B-Base, AgentMath-Qwen3-8B-SFT scores 28.8% while the RL variant scores 37.9%; based on Qwen3-30B-A3B-Instruct-2507, AgentMath-30B-A3B-SFT scores 43.5% while the RL variant scores 51.2%.

These results demonstrate that tool-augmented SFT substantially enhances mathematical reasoning capabilities across all base models, while RL consistently delivers an additional 7%–10% performance gain over SFT.

Table 16: SFT and RL performance of AgentMath on AIME25, HMMT25, and Google-IMO-AnswerBench based on Llama3.1-8B-Base and Qwen2.5-7B-Base.

| Model | Base Model | AIME25 | HMMT25 | Google-IMO-AnswerBench | Avg Score |
|---|---|---|---|---|---|
| *Based on Llama3.1-8B-Base* | | | | | |
| AgentMath-Llama-8B-SFT | Llama3.1-8B-Base | 53.5% | 45.8% | 18.3% | 39.2% |
| AgentMath-Llama-8B-RL | AgentMath-Llama-8B-SFT | 66.2% | 53.1% | 25.2% | 48.2% |
| *Based on Qwen2.5-7B-Base* | | | | | |
| AgentMath-Qwen2.5-7B-SFT | Qwen2.5-7B-Base | 66.4% | 55.1% | 27.3% | 49.6% |
| AgentMath-Qwen2.5-7B-RL | AgentMath-Qwen2.5-7B-SFT | 79.8% | 65.9% | 35.1% | 60.3% |

Table 17: SFT and RL performance of AgentMath on Google-IMO-AnswerBench based on Qwen3-series base models (1.7B, 8B, 30B).

| Model | Base Model | Google-IMO-AnswerBench |
|---|---|---|
| *Based on Qwen3-1.7B-Base* | | |
| AgentMath-Qwen3-1.7B-SFT | Qwen3-1.7B-Base | 11.7% |
| AgentMath-Qwen3-1.7B-RL | AgentMath-Qwen3-1.7B-SFT | 20.3% |
| *Based on Qwen3-8B-Base* | | |
| AgentMath-Qwen3-8B-SFT | Qwen3-8B-Base | 28.8% |
| AgentMath-Qwen3-8B-RL | AgentMath-Qwen3-8B-SFT | 37.9% |
| *Based on Qwen3-30B-A3B-Instruct-2507* | | |
| Qwen3-30B-A3B-Instruct-2507 | Qwen3-30B-A3B-Base | 30.3% |
| AgentMath-Qwen3-30B-A3B-SFT | Qwen3-30B-A3B-Instruct-2507 | 43.5% |
| AgentMath-Qwen3-30B-A3B-RL | AgentMath-Qwen3-30B-A3B-SFT | 51.2% |

### A.11.4 SUMMARY

We mitigate potential benchmark contamination through both *temporal controls*—utilizing a latest high-difficulty benchmark (Google-IMO-AnswerBench, November 2025) together with earlier pre-trained base models (Llama3.1-8B-Base and Qwen2.5-7B-Base, 2024)—and *model diversity*—validating across architectures spanning 1.7B to 30B parameters. The consistent and substantial performance improvements across all configurations validate the effectiveness of our tool-augmented data synthesis method and large-scale reinforcement learning on ultra-long sequences with massive tool calls, while confirming the reliability and robustness of our approach.

## A.12 TRAINING-TESTING OVERLAP ANALYSIS

### A.12.1 DATA DECONTAMINATION

To rigorously prevent data contamination between our training corpora (346K SFT and 42K RL samples) and the evaluation benchmarks (AIME24, AIME25, HMMT25), we employed a three-stage progressive deduplication pipeline.

**Stage 1: Problem-Level Exact Deduplication.** We first remove all training samples whose problem statements are exact string matches with those in the test sets.

**Stage 2: 4-gram + MinHash LSH Similarity-Based Deduplication.** To further eliminate problems with highly similar surface forms, we employ a combination of 4-gram analysis and MinHash Locality-Sensitive Hashing (LSH):

- **4-gram construction:** For each problem, we construct an $n$-gram set ($n=4$) by first tokenizing the text (word-level tokenization for English; Jieba tokenizer for Chinese) and then extracting all consecutive 4-token sequences.
- **Similarity measurement:** For any test problem $A$ and training problem $B$, we compute the Jaccard similarity of their 4-gram sets:

$$J(A, B) = \frac{|A \cap B|}{|A \cup B|},$$

  where $J(A, B) \in [0, 1]$, with higher values indicating greater $n$-gram-level overlap.
- **Filtering criterion:** We set the Jaccard similarity threshold at **0.6**. Any training sample with $J > 0.6$ relative to any test problem is removed.
- **Computational optimization:** To efficiently handle large-scale deduplication, we employ MinHash LSH, which maps MinHash signatures into hash buckets so that similar texts are clustered together, transforming the problem from global pairwise comparisons into localized bucket-wise searches and substantially reducing computational complexity. This procedure is implemented using NLTK and the `MinHashLSH` module from the `datasketch` library.

**Stage 3: Semantic Embedding-Based Deduplication.** To account for problems that exhibit substantial lexical differences yet remain semantically equivalent, we perform an additional semantic deduplication step:

- We generate embeddings for all problems in both training and test sets using the gte-large model.
- We compute cosine similarity between each test problem and all training problems using the sentence_transformers library.
- For each test problem, we rank all training problems by descending similarity and remove the top5 most semantically similar training samples.

Through this comprehensive three-stage pipeline, we removed approximately **8.3K** samples from the training set, ensuring that AgentMath's SFT and RL training data contain no overlapping or highly similar samples with respect to any evaluation benchmark.

### A.12.2 EFFECT OF TRAINING DATA VOLUME

We further investigate how training data volume affects downstream performance through systematic scaling experiments for both the SFT and RL stages.

**SFT Data Scaling.** Using Qwen3-8B-Base as the backbone, we vary the SFT training data from 2K to 300K samples. As shown in Table 18, the model exhibits significant and consistent performance gains across all benchmarks: AIME24 accuracy increases from 27.2% to 78.4%, AIME25 from 21.1% to 72.2%, and the overall average improves from 17.9% to 59.8% (+41.9 pp). These results demonstrate that our tool-augmented data synthesis approach exhibits strong scalability.

Table 18: Effect of SFT training data volume on benchmark performance.

| SFT Data Volume | AIME24 | AIME25 | IMO-AnswerBench | Avg |
|---|---|---|---|---|
| SFT-2k | 27.2 | 21.1 | 5.3 | 17.9 |
| SFT-20k | 60.5 | 53.3 | 12.8 | 42.2 |
| SFT-40k | 66.8 | 57.9 | 17.6 | 47.4 |
| SFT-80k | 69.2 | 61.8 | 21.4 | 50.8 |
| SFT-160k | 73.8 | 66.4 | 25.6 | 55.3 |
| SFT-300k | 78.4 | 72.2 | 28.8 | 59.8 |

**RL Data Scaling.** Starting from the best SFT checkpoint (300K), we vary the RL training data from 10K to 42K samples while keeping the number of training steps (200) and all hyperparameters fixed. As shown in Table 19, the model demonstrates consistent improvements: AIME24 accuracy increases from 80.8% to 85.3%, AIME25 from 75.5% to 80.4%, and the overall average improves from 62.4% to 66.8%.

Table 19: Effect of RL training data volume on benchmark performance. The first row shows the SFT-only baseline.

| RL Data Volume | AIME24 | AIME25 | IMO-AnswerBench | Avg |
|---|---|---|---|---|
| SFT-300k only | 78.4 | 72.2 | 28.8 | 59.8 |
| RL-10k | 80.8 | 75.5 | 30.9 | 62.4 |
| RL-20k | 82.7 | 77.8 | 32.4 | 64.3 |
| RL-30k | 84.2 | 79.0 | 33.7 | 65.6 |
| RL-42k | 85.3 | 80.4 | 34.7 | 66.8 |

Combining both sets of scaling experiments, we observe that AgentMath exhibits sustained and consistent performance improvements as the tool-augmented training data size increases across both the SFT and RL stages, further validating the strong scalability of our approach.

## A.13    DATA SYNTHESIS METHODOLOGY AND ANALYSIS

The data synthesis process in AgentMath involves several key design choices, including computational component segmentation and code complexity filtering. In this section, we provide detailed analysis and empirical justification for these design decisions, demonstrate scalability to new domains, and analyze the computational costs of our pipeline.

### A.13.1    COMPUTATIONAL COMPONENT SEGMENTATION

Our segmentation strategy divides each long chain-of-thought response generated by DeepSeek-R1 into fixed-length chunks (e.g., 3k tokens), yielding $N$ segments $(S_1, S_2, S_3, \ldots, S_N)$. We then use DeepSeek-V3-0324 to independently perform tool-augmented rewriting on each segment. In our 346k-example synthetic SFT dataset, responses have an average length of 18.3k tokens; thus, each response is on average split into 6 segments.

**Motivation.** In early experiments, we attempted to rewrite the full DeepSeek-R1 response without segmentation. This led to several issues:

- DeepSeek-R1 responses are excessively long.

- The instruction-following ability of DeepSeek-V3-0324 degrades markedly under ultra-long inputs, and the model becomes more prone to hallucinations.

- Consequently, the rewritten data exhibits a low frequency of tool usage, a high code execution failure rate, and extensive abbreviation or omission of intermediate natural language reasoning steps (which we refer to as the "textual reasoning omission rate").

To quantitatively assess the effect of segment length on data quality and downstream performance, we conducted controlled experiments on 20k synthetic examples and performed SFT on Qwen3-8B-Base. As shown in Table 20, the results demonstrate clear trends:

- **No segmentation:** The average number of tool calls is 1.6; the code execution failure rate is 31.8%; the textual reasoning omission rate is 78.5%; and AIME24/AIME25 accuracies are 40.3%/35.6%.

- **Segmentation with 9k tokens:** This setting improves over no segmentation, but segments remain relatively long, so tool usage and evaluation performance are still limited.

- **Segmentation with 3k tokens:** The average number of tool calls increases to 7.8; the code execution failure rate drops to 6.2%; the textual reasoning omission rate decreases to 2.3%; and AIME24/AIME25 accuracies reach 60.5%/53.3%.

- **Segmentation with 2k tokens:** Although the number of tool calls further increases, downstream performance does not improve significantly and appears to saturate.

Based on this analysis, we adopt a 3k-token segment length as the default configuration, as it strikes a favorable balance between data-quality metrics and downstream task performance.

Table 20: Effect of computational component segmentation on data quality and model performance using 20k AgentMath synthetic data.

| Configuration | Mean Tool Calls | Code Execution Failure Rate | Textual Reasoning Omission Rate | AIME24 Acc | AIME25 Acc |
|---|---|---|---|---|---|
| No Segmentation | 1.6 | 31.8% | 78.5% | 40.3% | 35.6% |
| Segment-9k-token | 2.1 | 28.4% | 67.9% | 44.8% | 38.7% |
| Segment-6k-token | 3.2 | 23.3% | 38.6% | 49.2% | 44.3% |
| Segment-4k-token | 5.7 | 12.5% | 8.8% | 57.3% | 51.6% |
| Segment-3k-token | 8.3 | 6.2% | 2.3% | 60.5% | 53.3% |
| Segment-2k-token | 10.1 | 5.9% | 2.1% | 59.4% | 51.9% |

### A.13.2 CODE COMPLEXITY FILTERING

We conducted a statistical analysis of code-length distributions over 20k synthetic instances. The results showed that samples with fewer than 5 lines of code constituted 5% of the dataset, with exactly 5 lines representing 7%, 6 lines 13%, 7 lines 24%, 8 lines 29%, 9 lines 13%, and 10 or more lines comprising 10%.

To investigate the relationship between code complexity and line count, we performed stratified uniform sampling, drawing 40 samples from each line-count category (280 samples total). Three authors independently annotated each sample's complexity as "Easy," "Medium," or "Hard" based on whether it involved computationally demanding operations (e.g., large-number arithmetic, complex equation solving, advanced linear algebra, combinatorial computations, or calculus). The annotation statistics are presented in Table 21.

Our analysis shows that among samples with $\leq 5$ lines of code, over 65% were labeled "Easy" while fewer than 10% were labeled "Hard." To prevent the model from learning to overuse tool invocations during training, we conservatively filtered out all samples containing 5 or fewer lines of code. In future work, we plan to employ LLMs such as Qwen3-30B for automated complexity assessment, replacing manual annotation to improve both objectivity and scalability of our filtering pipeline.

### A.13.3 SCALABILITY TO NEW DOMAINS

To demonstrate the generalizability of our approach, we extended AgentMath's automated data synthesis and cleaning pipeline to non-mathematical domains, including physics, chemistry, and biology, and evaluated the resulting model on the GPQA Diamond benchmark. GPQA Diamond assesses large language models' ability to solve graduate-level problems across biology, physics, and chemistry—a highly challenging benchmark where even PhD-level domain experts achieve only 65% accuracy.

We randomly sampled 40K examples from the open-source Science 220K dataset released by AM-Thinking and applied AgentMath's data synthesis method. Specifically, we segmented DeepSeek-R1's reasoning response into 3K-token chunks and transformed them into tool-augmented reasoning

Table 21: Statistics of code complexity distribution by lines of code.

| Lines of Code | Easy (%) | Medium (%) | Hard (%) |
|---|---|---|---|
| $< 5$ lines | 87% | 9% | 4% |
| $= 5$ lines | 66% | 27% | 7% |
| $= 6$ lines | 43% | 38% | 19% |
| $= 7$ lines | 24% | 49% | 27% |
| $= 8$ lines | 13% | 56% | 31% |
| $= 9$ lines | 7% | 51% | 42% |
| $\geq 10$ lines | 4% | 43% | 53% |

chains by replacing complex, error-prone computational steps with executable code. We then applied our automated cleaning pipeline, which includes: (1) filtering examples with $\leq 5$ lines of code, (2) correcting formatting errors, (3) removing samples with failed code execution, and (4) performing environmental feedback alignment to ensure consistency.

Using Qwen3-8B-Base as the foundation model, we conducted SFT and evaluated performance on GPQA across 8 independent runs, reporting the average accuracy (Avg@8). As shown in Table 22, AgentMath-8B-SFT-40k achieves 58.9% accuracy on GPQA, substantially outperforming DeepSeek-R1-Distill-Qwen-7B (49.1%). These results demonstrate that AgentMath's tool-augmented synthesis approach and automated cleaning pipeline generalize effectively beyond mathematics to scientific domains, highlighting the scalability and broad applicability of our method.

Table 22: GPQA accuracy comparison of different models at the 7B-8B scale.

| Model | Base Model | GPQA Acc |
|---|---|---|
| OpenThinker-7B | Qwen2.5-7B-Instruct | 42.4% |
| AReaL-boba-RL-7B | DeepSeek-R1-Distill-Qwen-7B | 47.6% |
| DeepSeek-R1-Distill-Qwen-7B | Qwen2.5-Math-7B | 49.1% |
| Light-R1-7B | DeepSeek-R1-Distill-Qwen-7B | 49.4% |
| AgentMath-8B-SFT-40k | Qwen3-8B-Base | 58.9% |

### A.13.4 COMPUTATIONAL COST ANALYSIS

We build upon the open-source AM-Thinking dataset, leveraging their existing responses generated by DeepSeek-R1-0528 to obtain 346K data samples. We employ DeepSeek-V3 for data synthesis, completing this process on 128 GPUs (each with 96GB memory) in approximately 62 hours. We then utilize Qwen3-30B as a judge model to verify consistency between actual code execution results and model-simulated outputs, requiring approximately 3 hours.

Subsequently, we perform SFT on the 316K synthetic samples using Qwen3-8B-Base as the base model, which takes approximately 44 hours. In total, the complete pipeline—encompassing data synthesis, cleaning, verification, and SFT training—requires approximately $62 + 3 + 44 = 109$ hours.

Notably, we also explore a more cost-effective alternative to address computational overhead. As illustrated in Figure 4, we apply SFT to only 20K samples starting from Qwen3-8B-Base, followed by 400 steps of large-scale reinforcement learning. This streamlined approach enables our Agent-Math model to achieve leading performance: 76.2% on AIME24 and 67.5% on AIME25, surpassing both OpenMath-Nemotron-7B (74.8% and 61.2%, respectively) and Qwen3-8B-Thinking (76.0% and 67.3%, respectively). This entire workflow, including data synthesis, cleaning, SFT, and RL training, requires only 76 hours total, providing a substantially more efficient solution.

### A.14 TOOL-AUGMENTED DATA SYNTHESIS PIPELINE

This section details our tool-augmented data synthesis methodology, including the segmentation strategy, code integration criteria, and teacher model selection.

### A.14.1 RESPONSE SEGMENTATION STRATEGY

We utilized long chain-of-thought responses generated by DeepSeek-R1 from the open-source AM-Thinking dataset. Each response is segmented into fixed-length 3k-token chunks, yielding $N$ segments $(S_1, S_2, S_3, \ldots, S_n)$. Across the 316k SFT synthetic data, the responses have an average length of 16.9k tokens, corresponding to roughly six segments per response.

Subsequently, for each segment, we applied the Tool-Augmented Data Synthesis Prompt detailed in Appendix A.6.2. We employed DeepSeek-V3-0324 to independently transform each segment by replacing complex and error-prone computational processes with executable code.

To accelerate the synthesis process, we initially instructed DeepSeek-V3-0324 to simulate code execution results based on the textual reasoning context. During post-processing, we replaced these simulated results with actual code execution outputs and employed Qwen3-30B as a judge model to assess their consistency. Samples exhibiting inconsistencies were filtered out to ensure data quality.

### A.14.2 CODE INTEGRATION CRITERIA

A key design consideration is determining when code usage is beneficial within the reasoning chain. Our Tool-Augmented Data Synthesis Prompt (Appendix A.6.2) guides DeepSeek-V3-0324 to first identify complex and error-prone computational processes, and then replace these manual calculations with code. Specifically, it covers the following types of computational tasks:

- **Complex Symbolic Algebra**: polynomial expansion, factorization, equation solving
- **Advanced Calculus**: differentiation, integration, definite integral computation
- **Probability and Combinatorics**: complex counting problems, probability distribution calculations
- **Linear Algebra**: matrix operations, matrix inversion, eigenvalue decomposition
- **Numerical Computation**: approximation calculations, large number arithmetic, geometric calculations
- **Other Error-Prone or Computation-Intensive Problems**

To control code complexity and ensure the reliability of the synthesized data, we further filter out low-complexity samples with $\leq 5$ lines of code. Furthermore, we randomly sampled 100 instances from the synthesized data and manually annotated whether DeepSeek-V3 effectively replaced complex textual computational processes with code. Results show that code replacement is beneficial and reasonable in 84% of the samples, thereby confirming the reliability of our proposed tool-augmented data synthesis approach.

### A.14.3 EFFECTIVENESS OF CODE-AUGMENTED REASONING

We demonstrate the effectiveness of code-augmented methods in replacing manual complex computations from multiple perspectives. As shown in Table 23, we systematically compare code-based and pure text-based reasoning along two dimensions: inference efficiency (measured by average token count) and performance metrics.

- **Token Analysis on Synthetic Data**: The SFT training dataset comprises 316k instances in total. Pure text-based reasoning averages 18.3k tokens per sample, while tool augmentation reduces this to 16.9k tokens. This demonstrates that substituting code for lengthy computational processes effectively reduces overall token consumption.
- **SFT Stage**: We comprehensively compare the efficiency and performance of code augmentation against pure text-based reasoning. Specifically, when trained on identical 20k SFT data, AgentMath-SFT-20k achieves accuracies of 60.5% and 53.3% on AIME24 and AIME25, respectively, significantly outperforming Text-based-SFT-20k (57.1% and 49.2%). Additionally, AgentMath-SFT-20k reduces the average token count by approximately 1.3k–3k compared to Text-based-SFT-20k.
- **RL Stage**: In tool-augmented RL training, using only 440 steps, AgentMath-RL-440-steps improves accuracy from 60.5% to 76.2% on AIME24 and from 53.3% to 67.5%

on AIME25. In contrast, pure text-based RL training requires 1600 steps (Text-based-RL-1600-steps) to achieve improvements from 57.1% to 68.7% on AIME24 and from 49.2% to 57.5% on AIME25. These results demonstrate that the code-augmented approach achieves approximately $4\times$ higher training efficiency while reducing the overall token count by 2.5k–5k compared to Text-based-RL-1600-steps.

In summary, introducing code augmentation in both SFT and RL stages yields substantial improvements in performance and efficiency over pure text-based reasoning, thereby validating that using code is more effective for complex computational segments of the reasoning chain.

Table 23: Accuracy and average inference token usage on AIME24 and AIME25 for pure text-based models and code-augmented AgentMath at both SFT and RL stages.

| Model | Base Model | AIME24 Acc | AIME24 Avg Tokens | AIME25 Acc | AIME25 Avg Tokens |
|---|---|---|---|---|---|
| Text-based-SFT-20k | Qwen3-8B-Base | 57.1% | 29k | 49.2% | 28k |
| AgentMath-SFT-20k | Qwen3-8B-Base | 60.5% | 26k | 53.3% | 26.7k |
| Text-based-RL-1600-steps | Text-based-SFT-20k | 68.7% | 32.3k | 57.5% | 31.2k |
| AgentMath-RL-440-steps | AgentMath-SFT-20k | 76.2% | 27.2k | 67.5% | 28.6k |

### A.14.4 TEACHER MODEL SELECTION

Different models may prefer distinct code-usage patterns for the same question, including variations in code injection positions, line counts, and code implementation details. To identify the optimal teacher model for our data synthesis pipeline, we conducted a comparative study between two candidate models: DeepSeek-V3-0324 and Qwen3-235B-Instruct-2507.

Specifically, we synthesized training data using each model on the 20k training set and performed supervised fine-tuning on Qwen3-8B-Base. As shown in Table 24, data synthesized by DeepSeek-V3-0324 averaged 6.5 tool calls per sample, compared to 5.3 tool calls per sample for Qwen3-235B-Instruct-2507. On the AIME24 and AIME25 benchmarks, the model trained on DeepSeek-V3-0324-synthesized data (AgentMath-20k-SFT-DeepSeek-V3) achieved accuracies of 60.5% and 53.3%, respectively, marginally outperforming the model trained on Qwen3-235B-Instruct-2507-synthesized data (59.3% and 51.7%). Notably, the former also demonstrated a higher average tool call frequency.

Based on these comprehensive performance results, we selected DeepSeek-V3-0324 as the teacher model for our data synthesis pipeline.

### A.15 FAIRNESS OF COMPARISON WITH TEACHER MODEL DISTILLATION

The dataset synthesis in AgentMath employs larger teacher models (DeepSeek R1 and V3, Qwen-30B), which may imply knowledge distillation from more powerful teacher models. To address potential concerns about the fairness of comparing such distilled models with those trained without teacher guidance, we conduct controlled experiments to isolate the contributions of our proposed method.

**Controlled Comparison with Retool.** To ensure a rigorous and fair comparison, we strictly followed the experimental setup of Retool, controlling for all key factors: the same 2k synthetic dataset, identical teacher model responses from DeepSeek-R1, the same base model (Qwen2.5-32B-Instruct), and 400 training steps during the RL phase. The sole difference lies in the data synthesis strategy: AgentMath leverages our proposed tool-augmented data synthesis method (Appendix A.6.2), whereas Retool employs its original synthesis prompt. Both approaches undergo identical two-stage training comprising SFT and RL. As shown in Table 25, the experimental results demonstrate that:

- AgentMath-32B-SFT-2k achieves 44.1% on AIME24 and 37.3% on AIME25, surpassing Retool-32B-SFT-2k (40.9% and 34.5%, respectively), thereby demonstrating the effectiveness of our tool-augmented data synthesis approach;

Table 24: Comparison of Qwen3-8B-Base models fine-tuned on the 20k AgentMath SFT dataset synthesized by DeepSeek-V3-0324 and Qwen3-235B-Instruct-2507. We report AIME24 and AIME25 accuracies (%) and the average number of tool calls per problem on each benchmark.

| Model | AIME24 | AIME24 Tool Calls Avg | AIME25 | AIME25 Tool Calls Avg |
|---|---|---|---|---|
| AgentMath-20k-SFT-DeepSeek-V3 | 60.5% | 5.1 | 53.3% | 5.7 |
| AgentMath-20k-SFT-Qwen3-235B-Instruct-2507 | 59.3% | 4.4 | 51.7% | 5.0 |

- AgentMath-32B-RL attains 74.8% on AIME24 and 56.6% on AIME25, consistently outperforming Retool-32B-RL (67.0% and 49.3%, respectively), further validating the superiority of our reinforcement learning training strategy.

These results confirm that AgentMath exhibits consistent and substantial performance gains across both training stages, validating the efficacy and robustness of our proposed method independent of teacher model choice.

Table 25: Comparison of AgentMath and Retool performance under the same experimental setup in both the SFT and RL stages.

| Model | AIME24 Acc | AIME25 Acc |
|---|---|---|
| Retool-32B-SFT-2k | 40.9% | 34.5% |
| AgentMath-32B-SFT-2k | 44.1% | 37.3% |
| Retool-32B-RL | 67.0% | 49.3% |
| AgentMath-32B-RL | 74.8% | 56.6% |

**Impact of Teacher Model Distillation on Competing Methods.** To investigate whether the performance gap would shrink if competing models were also allowed to distill from comparable teacher models, we adopted the open-source dataset from AM-Thinking, using the same 316k prompts and responses generated by DeepSeek-R1. We conducted supervised fine-tuning (SFT) on the Qwen3-8B-Base model for 6 epochs with a learning rate of 6e-5. To ensure experimental rigor, all configurations were kept identical except for the core difference in reasoning chain types: AM-Thinking uses pure textual reasoning chains, while AgentMath employs tool-augmented reasoning chains. We conducted 8 independent training runs and report pass@1 results from the checkpoint with the best validation performance.

As shown in Table 26, AgentMath-8B-SFT achieves 78.4% accuracy on AIME24 and 72.2% accuracy on AIME25, outperforming AM-Thinking-8B-SFT (74.9% and 67.4%, respectively), with an overall average performance gap of 4.1%. These results confirm that while the performance gap shrinks when competing models also distill from comparable teacher models, AgentMath consistently maintains its advantage due to the inherent benefits of tool-augmented reasoning chains.

Table 26: Comparison of pass@1 accuracy on AIME24 and AIME25 for AM-Thinking-8B-SFT and AgentMath-8B-SFT under the same training settings.

| Model | AIME24 Acc | AIME25 Acc | Avg Score |
|---|---|---|---|
| AM-Thinking-8B-SFT | 74.9% | 67.4% | 71.2% |
| AgentMath-8B-SFT | 78.4% | 72.2% | 75.3% |

A.16    REWARD DESIGN IN RL

Our reward function consists of two components: an answer correctness reward $R_{\mathrm{acc}}$ and a tool usage efficiency reward $R_{\mathrm{tool}}$. The correctness reward $R_{\mathrm{acc}}$ provides binary (0,1) feedback based on mathematical equivalence (verified using the math_verify library):

$$R_{\mathrm{acc}} = \begin{cases} 1, & \text{if is\_equivalent}(\hat{a}, a), \\ 0, & \text{otherwise}, \end{cases}$$

where $\hat{a}$ denotes the predicted answer and $a$ the ground truth answer. Conditioned on answer correctness, the tool usage reward $R_{\text{tool}}$ incentivizes efficient utilization of computational resources:

$$R_{\text{tool}} = \min\big(R_{\max},\ \alpha + \beta \cdot N_{\text{code}}\big) \quad \text{if } N_{\text{code}} > 0,$$

where $\alpha$ is the base tool usage reward, $\beta$ is the scaling factor for the number of invocations, and $R_{\max}$ is the reward upper bound. The total reward function is defined as:

$$R_{\text{total}} = R_{\text{acc}} + \mathbb{I}(R_{\text{acc}} = 1) \cdot R_{\text{tool}}.$$

In our work, we set $\alpha = 0.1$ and $\beta = 0.01$, and constrain the maximum tool reward to $R_{\max} = 1$, which encourages tool usage while preventing over-reliance. This design is motivated by the following considerations: applying $R_{\text{tool}}$ only to correct answers encourages the model to use tools appropriately to arrive at accurate solutions, avoiding rewarding erroneous tool invocations; $\alpha = 0.1$ differentiates between cases with and without tool usage, while the smaller $\beta = 0.01$ accommodates high-frequency tool calls during RL training (some exceeding 64 invocations), preventing reward inflation.

To validate the effectiveness of the reward design, we conducted controlled experiments using the SFT model trained on 20k synthetic data (AgentMath-SFT-20k), with a fixed 100 training steps, as shown in Table 27. Our experiments demonstrate:

- **Effectiveness of tool rewards:** Introducing tool rewards (all answers) outperforms the no-tool-reward baseline in both average accuracy (61.4% vs. 60.2%) and average tool calls (6.2 vs. 5.7), indicating that tool rewards effectively incentivize the model to leverage computational tools and improve performance.

- **Advantage of tool rewards (correct answers only):** Furthermore, compared to tool rewards (all answers), applying tool rewards exclusively to correct answers ($\alpha = 0.1, \beta = 0.01$) achieves the best performance in both average accuracy (62.6% vs. 61.4%) and average tool calls (6.3 vs. 6.2).

- **Optimal hyperparameter configuration:** In a grid search over $\alpha \in \{0.1, 0.2\}$ and $\beta \in \{0.01, 0.02\}$, $\alpha = 0.1, \beta = 0.01$ achieves the best accuracy-efficiency trade-off (62.6% average accuracy, 6.3 average tool calls).

Therefore, based on performance and tool call metrics, we ultimately adopt the correctness-conditioned tool reward design with parameters $\alpha = 0.1$ and $\beta = 0.01$ for RL training.

Table 27: Ablation study results on reinforcement learning reward design.

| Reward Configuration | AIME24 Accuracy | AIME24 Tool Calls | AIME25 Accuracy | AIME25 Tool Calls | Avg. Accuracy | Avg. Tool Calls |
|---|---|---|---|---|---|---|
| AgentMath-SFT-20k (Baseline) | 60.5% | 5.1 | 53.3% | 5.7 | 56.9% | 5.4 |
| RL with No tool reward | 65.2% | 5.5 | 55.1% | 5.9 | 60.2% | 5.7 |
| RL with Tool reward (all answers), $\alpha = 0.1, \beta = 0.01$ | 66.4% | 6.1 | 56.3% | 6.3 | 61.4% | 6.2 |
| **RL with Tool reward (correct answers only),** $\alpha = 0.1, \beta = 0.01$ | **67.4%** | **6.0** | **57.7%** | **6.5** | **62.6%** | **6.3** |
| RL with Tool reward (correct answers only), $\alpha = 0.1, \beta = 0.02$ | 66.7% | 6.4 | 56.8% | 6.7 | 61.8% | 6.6 |
| RL with Tool reward (correct answers only), $\alpha = 0.2, \beta = 0.01$ | 66.9% | 5.9 | 56.5% | 6.2 | 61.7% | 6.1 |

