# OpenReview forum: "AgentMath: Empowering Mathematical Reasoning for Large Language Models via Tool-Augmented Agent"
_ICLR.cc/2026/Conference — ICLR 2026 Poster_

### Official Review · Reviewer_5L8W · 2025-10-25

**Soundness:** 3
**Presentation:** 3
**Contribution:** 3
**Rating:** 4
**Confidence:** 4

**Summary:**

The paper presents AgentMath, an agentic framework integrating code interpreters with LLMs to enhance math problem solving and mathematical reasoning to be more accurate and computationally efficient through tool-augmented learning. It contains two major contributions. The first is the introduction of a three-step data synthesis pipeline for generating tool-augmented training trajectories. The second is a reinforcement learning (RL) techniques that combine natural language reasoning with symbolic code execution. To accelerate RL training, the authors design an efficient infrastructure leveraging asynchronous rollout scheduling and adaptive load balancing. The paper primarily uses math reasoning benchmarks (AIME24, AIME25, and HMMT) for experiments, and reports commendable improvements on those datasets over baseline models/approaches.

**Strengths:**

* The paper conducts an important study on integrating the Code Interpreter into LLMs to improve mathematical reasoning via symbolic computation.

* The data synthesis pipeline and training strategies (including the code execution sandbox and adaptive load balancing) are technically sound and thoughtfully designed to improve training efficiency.

* The proposed RL with Code Interpreter integration is shown to be effective through extensive experimental results.

* The experiments are extensive and systematically conducted.

* The paper is well written and easy to follow.

**Weaknesses:**

* Concern about benchmark: The math benchmarks (AIME24, AIME25, HMMT) each contain only about 30 questions. Prior work has shown that these datasets may have leaked into open-source model training corpora. Thus, it is unclear whether the reported results are entirely reliable. The authors are encouraged to provide more evidence on the independent contribution of the proposed approach.

* Training-testing overlap: The authors report 346k training questions for SFT and 42k for RL, while the total number of testing questions is under 100. It is questionable whether the training data already contains components or similar structures to the test questions. Although an n-gram filtering algorithm is mentioned, the specific implementation details are missing. Have the authors performed ablation studies to evaluate the effect of training data volume on test performance?

* Unclear heuristic data synthesis: The data synthesis process involves multiple human heuristics (e.g., computational component segmentation, code complexity filtering), and several components appear to require iterative manual adjustment. These steps may limit scalability to new benchmarks or domains. Moreover, the data synthesis and cleaning costs seem high given the repeated use of LLMs.

* The paper does not clarify how textual answers are segmented into computational parts or how the model determines when code usage is beneficial. Different models may prefer distinct code-usage patterns even for the same question.

* The dataset synthesis employs larger teacher models (DeepSeek R1 and V3, Qwen-30B). This may imply knowledge distillation from more powerful teacher models. Comparing such distilled models with those trained without teacher guidance may be unfair.

* The authors should consider moving key details, such as dataset sizes for SFT and RL, to the main paper because they are critical for comprehension.

* Many designs in the algorithms, such as the four quality refinement modules in Step 2 of the data set synthesis, and reward design in RL, need further justification or ablation studies. Otherwise the overall method seems to contain too many ad-hoc details that may limit the generalizability of the proposed approach.

* Typo: In line 1600, the figure reference is missing.

**Questions:**

1. Given that AIME24, AIME25, and HMMT each contain fewer than 30 questions and have been reported as leaked in prior work, how do the authors ensure that these benchmarks were not inadvertently seen during pretraining or fine-tuning? Have the authors considered using larger or newly constructed evaluation sets to validate the robustness of the results?
1. With 346k training questions for SFT and 42k for RL but fewer than 100 test questions, how do the authors ensure that the training data do not overlap or contain structurally similar questions to the test set?

1. What exact n-gram filtering algorithm and similarity threshold were used to remove overlapping data?

1. Have the authors conducted ablation studies to quantify how training data volume affects final test accuracy? Would performance remain stable if a smaller, filtered subset of training data were used?

1. Many steps in the data synthesis pipeline involve human heuristics such as computational component segmentation and code complexity filtering. How much manual adjustment or iteration do these steps require?

1. Can the synthesis and cleaning pipeline be generalized or automated for other benchmarks or non-math domains?

1. What is the total computational cost (e.g., GPU hours or LLM API calls) associated with synthesizing and verifying the datasets?

1. How are textual reasoning steps segmented into computational components when deciding where to inject code? What criteria or metrics determine whether using code is more effective for a given part of the reasoning chain?

1. Since different models can prefer different code usage patterns for the same question, how robust is the segmentation procedure across models?

1. The paper uses large teacher models (DeepSeek R1, DeepSeek V3, and Qwen-30B) for data synthesis. How do the authors ensure fairness when comparing with models trained without such teacher guidance? Would the performance gap shrink if competing models were also allowed to distill from comparable teacher models?

---

> ### Author Response · Authors · 2025-11-24
> **Response to Reviewer 5L8W (1/16)**
>
> Dear Reviewer 5L8W,
>
> We would like to express our sincere gratitude to you for taking the time from your busy schedule to review our paper and for providing such valuable and insightful comments. Your professional feedback is crucial for helping us improve the quality of the paper and make it more thorough and competitive. We are truly grateful for your support and constructive suggestions.
>
> To ensure we address each of your comments as thoroughly and clearly as possible, our replies are relatively detailed. We sincerely apologize for any additional time this may require and respectfully ask for your understanding. Thank you once more for your patience in considering our rebuttal. Below, we provide detailed, point-by-point responses to the Weaknesses and Questions raised in your review.
>
> To further facilitate community reproducibility and future research, we have added **the  Reproducibility Statement** to our **latest upload of revised paper (page 11, lines 540-555)**.  And we have made **available our anonymous main source code, data synthesis pipeline, and training/eval scripts in the supplementary materials and [https://anonymous.4open.science/r/AgentMath-E5BR](https://anonymous.4open.science/r/AgentMath-E5BR).**  Due to time constraints, certain parts of this codebase may currently rely on hard-coded elements or may be missing. We sincerely apologize for any inconvenience and we are committed to continuously improving this repository, enhancing code quality, and making the training process more robust, smooth, and stable.  We hope these materials will help reviewers and researchers better understand and reproduce our work.
>
> Furthermore, **in Appendix A.11--A.16 of our latest upload of revised paper (pages 41–53, lines 2201--2844)**, we also have added the below discussions with the Reviewer-5L8W on the weaknesses and questions of our paper to respond to the Reviewer-5L8W's comments and to further improve the quality of our research.
>
> **Since some Weaknesses and Questions are similar, we will discuss them together.** Please find below a detailed discussion of the points you have raised:
>
> > **Weaknesses-1**: Concern about benchmark: The math benchmarks (AIME24, AIME25, HMMT) each contain only about 30 questions. Prior work has shown that these datasets may have leaked into open-source model training corpora. Thus, it is unclear whether the reported results are entirely reliable. The authors are encouraged to provide more evidence on the independent contribution of the proposed approach.
>
> >**Questions-1**: Given that AIME24, AIME25, and HMMT each contain fewer than 30 questions and have been reported as leaked in prior work, how do the authors ensure that these benchmarks were not inadvertently seen during pretraining or fine-tuning? Have the authors considered using larger or newly constructed evaluation sets to validate the robustness of the results?
>
> We sincerely thank you for raising this important question. Considering that Questions-1 and Weaknesses-1 address similar concerns, **so I have included my response to Questions-1 in the Weaknesses-1 section below.**
>
> To systematically validate the reliability of our results and mitigate potential evaluation benchmark leakage concerns, we conducted supplementary experiments and analyses from three complementary perspectives.
>
> **(1) IMO-AnswerBench: A latest, larger and High-Difficulty Benchmark**
>
> Considering that datasets such as AIME24, AIME25, and HMMT25 may have partially appeared in the pre-training corpora of open-source models, we evaluate AgentMath on IMO-AnswerBench[1], a recent and larger mathematics olympiad benchmark released by Google DeepMind **in November 2025**. This benchmark comprises **400 carefully curated problems** spanning four domains about algebra, combinatorics, geometry, and number theory, with **100 problems per category**, all sourced from national, regional, and international olympiad competitions.
>
> **To mitigate data memorization and leakage, IMO-AnswerBench[1] systematically rewrites original competition problems** through several strategies: renaming points or lines in geometry problems, rephrasing problem statements, modifying numerical values and or introducing distractors, and reformulating problems using entirely different yet semantically equivalent expressions. This design substantially reduces the probability of verbatim matches in pre-training corpora and minimizes the potential for memorization-based advantages, thereby enhancing evaluation robustness and fairness.
>
> The performance of current frontier models on this benchmark: DeepSeek-V3 at 37.0%, Qwen3-235B at 53.8%, and DeepSeek-R1 at 60.8%, indicates that the benchmark possesses sufficient discriminative power and challenge.
>
> [1] Luong M T, Hwang D, Nguyen H H, et al. Towards Robust Mathematical Reasoning[C]//Proceedings of the 2025 Conference on Empirical Methods in Natural Language Processing. 2025: 35406-35430.

---

> ### Author Response · Authors · 2025-11-24
> **Response to Reviewer 5L8W (2/16)**
>
> **[ Continue the response to above Weaknesses-1 and Questions-1]**
>
>  Moreover, **IMO-AnswerBench's release date (November 2025) is notably later than the public release of Qwen3-series models (May 2025)**, further minimizing the likelihood of contamination in their training corpora.
>
> As detailed in the table 1 below, following the experimental setup specified in the IMO-AnswerBench paper[1], we report results averaged over 8 independent runs. At comparable parameter scales, AgentMath achieves the following performance across the four categories:
>
> **At the 1.5B--1.7B scale:** AgentMath-Qwen3-1.7B Avg score achieves 20.3%, outperforming OpenReasoning-Nemotron-1.5B (17.8%);
>
> **At the 7B--8B scale:** AgentMath-Qwen2.5-7B Avg score achieves 35.1% and AgentMath-Qwen3-8B achieves 37.9%, both substantially outperforming OpenReasoning-Nemotron-7B (33.0%);
>
> **At the 30B--32B scale:** AgentMath-Qwen3-30B-A3B Avg score achieves 51.2%, surpassing Qwen3-30B-A3B-Thinking-2507 (41.4%), Claude Sonnet 4, DeepSeek-V3, and Kimi-K2-Instruct, while approaching the performance of Qwen3-235B;
>
> **At scales larger than 32B:** AgentMath-Qwen3-235B-A22B-SFT Avg score achieves 55.4%, surpassing Qwen3-235B and approaching DeepSeek-R1.
>
> These results demonstrate that AgentMath maintains strong performance and robustness across different parameter scales on a challenging evaluation set that is demonstrably free from data leakage concerns.
>
> **Table 1: Performance of AgentMath on Google-IMO-AnswerBench, a recent mathematics olympiad benchmark released by Google DeepMind in November 2025. Our model  is compared against other leading models, with accuracy (avg@8) as the evaluation metric according to the IMO-AnswerBench[1].**
>
> | Model (Google-IMO-AnswerBench) | Algebra | Combinatorics | Geometry | Number Theory | Avg score |
> | :--- | :---: | :---: | :---: | :---: | :---: |
> | **Proprietary models** | | | | | |
> | Gemini Deep Think (IMO Gold) | 85.0% | 69.0% | 88.0% | 78.0% | 80.0% |
> | Grok 4 | 75.5% | 55.9% | 80.1% | 80.9% | 73.1% |
> | Gemini 2.5 Deep Think | 78.0% | 49.0% | 83.0% | 77.0% | 71.8% |
> | Gemini 2.5 Pro | 73.4% | 48.0% | 74.3% | 77.1% | 68.2% |
> | o4-mini (high reasoning) | 71.3% | 46.6% | 78.4% | 75.3% | 67.9% |
> | GPT-5 | 69.9% | 46.4% | 74.8% | 71.2% | 65.6% |
> | o3 | 62.8% | 43.0% | 70.6% | 68.0% | 61.1% |
> | Claude Sonnet 4 | 20.6% | 17.8% | 26.0% | 27.6% | 23.0% |
> | Claude Opus4 | 19.4% | 20.0% | 23.3% | 26.6% | 22.3% |
> | **Frontier Models (1B--2B)** | | | | | |
> | DeepSeek-R1-Distill-Qwen-1.5B | 5.5% | 9.9% | 14.5% | 8.7% | 9.7% |
> | Qwen3-1.7B Thinking | 11.4% | 12.5% | 21.5% | 17.5% | 15.7% |
> | OpenReasoning-Nemotron-1.5B | 16.3% | 14.0% | 22.6% | 18.4% | 17.8% |
> | **AgentMath-Qwen3-1.7B** | **18.6%** | **16.2%** | **24.4%** | **21.8%** | **20.3%** |
> | **Frontier Models (7B--8B)** | | | | | |
> | DeepSeek-R1-Distill-Llama3.1-8B | 10.2% | 17.4% | 21.6% | 17.0% | 16.6% |
> | DeepSeek-R1-Distill-Qwen-7B | 12.8% | 20.6% | 22.8% | 18.0% | 18.5% |
> | Qwen3-8B-Thining | 17.0% | 26.8% | 38.0% | 25.8% | 26.9% |
> | DeepSeek-R1-0528-Distill-Qwen3-8B | 21.2% | 31.7% | 43.3% | 27.3% | 30.9% |
> | OpenReasoning-Nemotron-7B | 23.6% | 35.2% | 43.2% | 29.9% | 33.0% |
> | **AgentMath-Qwen2.5-7B** | **24.9%** | **37.4%** | **45.6%** | **32.5%** | **35.1%** |
> | **AgentMath-Qwen3-8B** | **28.3%** | **39.8%** | **49.5%** | **33.8%** | **37.9%** |
> | **Frontier Models (30B--32B)** | | | | | |
> | DeepSeek-R1-Distill-Qwen-32B | 17.2% | 25.1% | 33.0% | 22.5% | 24.4% |
> | Qwen3-30B-A3B-Instruct-2507 | 27.8% | 29.9% | 36.2% | 27.4% | 30.3% |
> | AM-Thinking-v1-32B | 27.9% | 36.2% | 50.1% | 29.2% | 35.8% |
> | Qwen3-30B-A3B-Thinking-2507 | 34.7% | 44.3% | 53.2% | 33.3% | 41.4% |
> | **AgentMath-Qwen3-30B-A3B** | **43.4%** | **40.0%** | **69.6%** | **51.9%** | **51.2%** |
> | **Frontier Models (>32B)** | | | | | |
> | DeepSeek V3 | 39.0% | 26.0% | 35.0% | 48.0% | 37.0% |
> | Kimi-K2-Instruct | 45.6% | 31.1% | 49.3% | 56.9% | 45.8% |
> | Qwen3-235B | 57.6% | 37.5% | 57.6% | 62.3% | 53.8% |
> | **AgentMath-Qwen3-235B-A22B-SFT** | **49.4%** | **43.2%** | **73.5%** | **55.3%** | **55.4%** |
> | DeepSeek R1 | 65.0% | 40.0% | 73.0% | 65.0% | 60.8% |
>
> **(2) Validation Using Base Models Released Prior to Evaluation Benchmark Publication**
>
> To further mitigate concerns regarding potential evaluation data leakage into the pretraining corpus, we employ open-source base models released before the publication of evaluation benchmarks for our SFT and RL training. The timeline is as follows:
>
> *   Qwen2.5-7B-Base release: September 2024
> *   Llama3.1-8B-Base release: July 2024
> *   AIME25 publication: February 2025
> *   HMMT25 publication: February 2025
> *   Google-IMO-AnswerBench publication: November 2025
>
> [1] Luong M T, Hwang D, Nguyen H H, et al. Towards Robust Mathematical Reasoning[C]//Proceedings of the 2025 Conference on Empirical Methods in Natural Language Processing. 2025: 35406-35430.

---

> ### Author Response · Authors · 2025-11-24
> **Response to Reviewer 5L8W (3/16)**
>
> **[ Continue the response to above Weaknesses-1 and Questions-1]**
>
> **So, we use Llama3.1-8B-Base and Qwen2.5-7B-Base pre-trained models as our backbone. Because all benchmarks were thus published after model pretraining was completed, thereby precluding any possibility of data contamination.**
>
> As shown in the table 2 and table 3 below, AgentMath trained on these earlier base models achieves superior performance across multiple benchmarks:
>
> *   Building on Llama3.1-8B-Base, AgentMath-Llama-8B achieves 66.2% and 53.1% on AIME25 and HMMT25, respectively, significantly outperforming Llama3.1-Nemotron-Nano-8B-v1 (48.0% and 26.7%);
> *   Building on Qwen2.5-7B-Base, AgentMath-Qwen2.5-7B attains 79.8% and 65.9% on AIME25 and HMMT25, respectively, surpassing OpenReasoning-Nemotron-7B (78.2% and 63.5%);
> *   On the more challenging Google-IMO-AnswerBench, AgentMath-Llama-8B Avg score achieves 25.2%, outperforming DeepSeek-R1-Distill-Llama3.1-8B (16.6%), while AgentMath-Qwen2.5-7B (35.1%) Avg score also exceeds OpenReasoning-Nemotron-7B (33.0%).
>
> Therefore, by leveraging base models **released in 2024 (Qwen2.5-7B-Base and Llama3.1-8B-Base)**, AgentMath demonstrates consistent and leading performance across multiple mathematical reasoning benchmarks **released in 2025 (AIME25, HMMT25, and IMO-AnswerBench)**. These results not only eliminate data contamination risks but also establish the robustness and reliability of our model.
>
> **Table 2: Performance of AgentMath on AIME24/25, and HMMT25 based on Llama3.1-8B-Base and Qwen2.5-7B-Base. Our model is compared against other leading models, with accuracy (avg@32) as the evaluation metric.**
>
> | Models | Base Model | Tool Use | AIME25 | HMMT25 |
> | :--- | :--- | :--- | :--- | :--- |
> | **Based on Llama3.1-8B-Base/Instruct** | | | | |
> | DeepSeek-R1-Distill-Llama-8B | Llama-3.1-8B-Base | ✗ | 28.7 | 13.8 |
> | Llama3.1-Nemotron-Nano-8B-v1 | Llama-3.1-8B-Instruct | ✗ | 48.0 | 26.7 |
> | **AgentMath-Llama-8B** | **Llama-3.1-8B-Base** | **✓** | **66.2** | **53.1** |
> | **Based on Qwen2.5-7B-Base/Instruct** | | | | |
> | ZeroTIR-7B | Qwen-2.5-7B-Base | ✓ | 30.0 | 22.5 |
> | SimpleTIR-7B | Qwen2.5-7B-Base | ✓ | 30.9 | 29.7 |
> | DeepSeek-R1-Distill-Qwen-7B | Qwen2.5-Math-7B-Base | ✗ | 39.7 | 16.3 |
> | OpenR1-Distill-7B | Qwen2.5-Math-7B-Base | ✗ | 39.7 | 25.7 |
> | Light-R1-7B-DS | DeepSeek-R1-Distill-Qwen-7B | ✗ | 44.3 | 27.6 |
> | AReal-boba-7B | DeepSeek-R1-Distill-Qwen-7B | ✗ | 48.3 | 29.4 |
> | Skywork-OR1-7B | DeepSeek-R1-Distilled-Qwen-7B | ✗ | 54.6 | 35.7 |
> | AceReason-Nemotron-1.1-7B | DeepSeek-R1-Distill-Qwen-7B | ✗ | 64.8 | 42.9 |
> | OpenReasoning-Nemotron-7B | Qwen2.5-7B-Instruct | ✗ | 78.2 | 63.5 |
> | **AgentMath-Qwen2.5-7B** | **Qwen2.5-7B-Base** | **✓** | **79.8** | **65.9** |
>
> **Table 3: Performance of AgentMath on Google-IMO-AnswerBench based on Llama3.1-8B-Base and Qwen2.5-7B-Base. Our model is compared against other leading models, with accuracy (avg@8) as the evaluation metric.**
>
> | Model (Google-IMO-AnswerBench) | Base Model | Algebra | Combinatorics | Geometry | Number Theory | Avg Score |
> | :--- | :--- | :---: | :---: | :---: | :---: | :---: |
> | **Based on Llama3.1-8B-Base** | | | | | | |
> | DeepSeek-R1-Distill-Llama3.1-8B | Llama3.1-8B-Base | 10.2% | 17.4% | 21.6% | 17.0% | 16.6% |
> | **AgentMath-Llama-8b** | **Llama3.1-8B-Base** | **18.8%** | **24.6%** | **33.7%** | **23.7%** | **25.2%** |
> | **Based on Qwen2.5-7B Base/Instruct** | | | | | | |
> | OpenReasoning-Nemotron-7B | Qwen2.5-7B -Instruct | 23.6% | 35.2% | 43.2% | 29.9% | 33.0% |
> | **AgentMath-Qwen2.5-7B** | **Qwen2.5-7B -Base** | **24.9%** | **37.4%** | **45.6%** | **32.5%** | **35.1%** |
>
> **(3) Systematic Comparison of SFT and RL Improvements on Non-Contaminated Evaluation Benchmarks**
>
> To rigorously evaluate the independent contributions of SFT and RL in AgentMath while mitigating potential data contamination, **we conduct systematic comparisons exclusively on benchmarks confirmed to be released after the pre-training cutoff date of their corresponding backbone models**:
>
> *   For Llama3.1-8B-Base and Qwen2.5-7B-Base, we report results on AIME25, HMMT25, and Google-IMO-AnswerBench;
> *   For Qwen3-1.7B-Base, Qwen3-8B-Base, and Qwen3-30B-A3B-Instruct-2507, we report results on Google-IMO-AnswerBench only.

---

> ### Author Response · Authors · 2025-11-24
> **Response to Reviewer 5L8W (4/16)**
>
> **[ Continue the response to above Weaknesses-1 and Questions-1]**
>
> As shown in the **table 4** and **table 5** below, the results reveal consistent improvements across all base models. Based on Llama3.1-8B-Base, AgentMath-Llama-8b-SFT achieves an average score of 39.2% across the three benchmarks (AIME25, HMMT25, and Google-IMO-AnswerBench), while AgentMath-Llama-8b-RL reaches 48.2%. Based on Qwen2.5-7B-Base, AgentMath-Qwen2.5-7B-SFT attains 49.6%, with AgentMath-Qwen2.5-7B-RL achieving 60.3%.
>
> Based on Qwen3-series, we observe similar trends on Google-IMO-AnswerBench: Based on Qwen3-1.7B-Base, AgentMath-Qwen3-1.7B-SFT scores 11.7% while AgentMath-Qwen3-1.7B-RL scores 20.3%; Based on Qwen3-8B-Base, AgentMath-Qwen3-8B-SFT scores 28.8% while AgentMath-Qwen3-8B-RL scores 37.9%; Based on Qwen3-30B-A3B-Instruct-2507, AgentMath-Qwen3-30B-A3B-SFT scores 43.5% while AgentMath-Qwen3-30B-A3B-RL scores 51.2%.
>
> These results demonstrate that tool-augmented SFT substantially enhances mathematical reasoning capabilities across all base models, while RL consistently delivers an additional 7%–10% performance gain over SFT.
>
> **Table 4: The SFT and RL performance of AgentMath on AIME25, HMMT25, Google-IMO-AnswerBench based on Llama3.1-8B-Base and Qwen2.5-7B-Base.**
>
> | Model | Base Model | AIME25 | HMMT25 | Google-IMO-AnswerBench | Avg Score |
> | :--- | :--- | :---: | :---: | :---: | :---: |
> | **Based on Llama3.1-8B-Base** | | | | | |
> | AgentMath-Llama-8b-SFT | Llama3.1-8B-Base | 53.5% | 45.8% | 18.3% | 39.2% |
> | AgentMath-Llama-8b-RL | AgentMath-Llama-8b-SFT | 66.2% | 53.1% | 25.2% | 48.2% |
> | **Based on Qwen2.5-7B Base** | | | | | |
> | AgentMath-Qwen2.5-7B-SFT | Qwen2.5-7B -Base | 66.4% | 55.1% | 27.3% | 49.6% |
> | AgentMath-Qwen2.5-7B-RL | AgentMath-Qwen2.5-7B-SFT | 79.8% | 65.9% | 35.1% | 60.3% |
>
> **Table 5: The SFT and RL performance of AgentMath on Google-IMO-AnswerBench based on Qwen3-series base model (1.7B, 8B, 30B).**
>
> | Model | Base Model | Google-IMO-AnswerBench |
> | :--- | :--- | :---: |
> | **Based on Qwen3-1.7B-Base** | | |
> | AgentMath-Qwen3-1.7B-SFT | Qwen3-1.7B-Base | 11.7% |
> | AgentMath-Qwen3-1.7B-RL | AgentMath-Qwen3-1.7B-SFT | 20.3% |
> | **Based on Qwen3-8B-Base** | | |
> | AgentMath-Qwen3-8B-SFT | Qwen3-8B-Base | 28.8% |
> | AgentMath-Qwen3-8B-RL | AgentMath-Qwen3-8B-SFT | 37.9% |
> | **Based on Qwen3-30B-A3B-Instruct-2507** | | |
> | Qwen3-30B-A3B-Instruct-2507 | Qwen3-30B-A3B-Base | 30.3% |
> | AgentMath-Qwen3-30B-A3B-SFT | Qwen3-30B-A3B-Instruct-2507 | 43.5% |
> | AgentMath-Qwen3-30B-A3B-RL | AgentMath-Qwen3-30B-A3B-SFT | 51.2% |
>
> In summary, we mitigate potential benchmark contamination through both temporal controls, **utilizing a latest, larger and high-difficulty benchmark (Google IMO-AnswerBench, November 2025) with earlier pre-trained base models (Llama3.1-8B-Base and Qwen2.5-7B-Base, 2024) and model diversity, validating across architectures spanning 1.7B to 30B parameters**. The consistent and substantial performance improvements across all configurations validate the effectiveness of our tool-augmented data synthesis method and large-scale reinforcement learning on ultra-long sequences with massive tool-calls, while confirming the reliability and robustness of our model.
>
> **We have added the above discussions about the Weaknesses-1 to Appendix A.11 of our latest upload of revised paper (pages 41–45, lines 2202–2387).**
>
> It is worth noting that, for the detailed deduplication of AgentMath SFT and RL training data, please refer to Weaknesses-2 below.
>
> > **Weaknesses-2**: Training-testing overlap: The authors report 346k training questions for SFT and 42k for RL, while the total number of testing questions is under 100. It is questionable whether the training data already contains components or similar structures to the test questions. Although an n-gram filtering algorithm is mentioned, the specific implementation details are missing. Have the authors performed ablation studies to evaluate the effect of training data volume on test performance?
>
> We greatly appreciate your insightful questions and apologize for any confusion. Please allow me to address them one by one.

---

> ### Author Response · Authors · 2025-11-24
> **Response to Reviewer 5L8W (5/16)**
>
> >> **Weaknesses-2-Q1**: Training-testing overlap: The authors report 346k training questions for SFT and 42k for RL, while the total number of testing questions is under 100. It is questionable whether the training data already contains components or similar structures to the test questions. Although an n-gram filtering algorithm is mentioned, the specific implementation details are missing.
>
> >> **Questions-2**: With 346k training questions for SFT and 42k for RL but fewer than 100 test questions, how do the authors ensure that the training data do not overlap or contain structurally similar questions to the test set?
>
> We sincerely appreciate your valuable feedback and apologize for the lack of clarity regarding our data deduplication procedure. Considering that Questions-2 and Weaknesses-2-Q1 address similar concerns, **so I have included my response to Questions-2 in the Weaknesses-2-Q1 section below.**
> We provide a comprehensive description below to address your concerns.
>
> In Appendix A.5.1 "SUPERVISED FINE-TUNING (SFT) DATA CONSTRUCTION (Stage 1: Foundational Data Curation and Filtering)" of our paper (page 25, lines 1296--1322), we described our deduplication strategy. To rigorously prevent data contamination, we employed a three-stage progressive deduplication pipeline as follows:
>
> **Stage 1: Problem-level Exact Deduplication**
>
> We remove all training samples whose problem statements are exact matches with those in the test sets by string matching.
>
> **Stage 2: 4-gram + MinHash LSH Similarity-based Deduplication**
>
> To further eliminate problems with highly similar surface forms, we employ a combination of **4-gram** analysis and MinHash LSH:
>
> *   **4-gram construction:** For each problem, we construct a **n-gram (n=4)** set by first tokenizing the text (word-level tokenization for English text; Jieba tokenizer for Chinese text) and then extracting all consecutive 4-token sequences.
> *   **Similarity measurement:** For any test problem $A$ and training problem $B$, we compute the Jaccard similarity of their 4-gram sets:
>     $$
>     J(A,B) = \frac{|A \cap B|}{|A \cup B|}
>     $$
>     where $J(A,B) \in [0,1]$, with higher values indicating greater n-gram-level similarity.
> *   **Filtering criterion:** We set the **Jaccard similarity threshold at 0.6**. Training samples with Jaccard similarity exceeding 0.6 relative to any test problem are removed from the training set.
> *   **Computational optimization:** To efficiently handle large-scale deduplication, we employ MinHash LSH, which uses hash functions to map MinHash signatures into buckets. This enables similar texts to be clustered together, transforming the problem from global pairwise comparisons into localized bucket-wise searches and substantially reducing computational complexity.
> *   **Implementation:** This procedure is implemented using NLTK and the MinHashLSH module from the datasketch library.
>
> **Stage 3: Semantic Embedding-based Deduplication**
>
> To account for problems that exhibit substantial lexical differences yet remain semantically similar, we perform an additional semantic deduplication step:
>
> *   We generate embeddings for all problem texts in both training and test sets using the **gte-large[1]** model.
> *   We compute semantic similarity (cosine similarity) between each test problem and all training problems using the sentence_transformers library.
> *   For each test problem, we rank all training problems by descending similarity and remove the **top 5** most semantically similar training samples.
>
> Through this comprehensive three-stage pipeline (**exact matching + 4-gram/MinHashLSH-based filtering + semantic embedding-based filtering**), we removed approximately **8.3k** samples from the training set. This rigorous deduplication ensures that AgentMath's SFT and RL training data contain no overlapping or highly similar samples with respect to any evaluation benchmark (AIME24, AIME25, HMMT25), thereby effectively preventing data contamination and guaranteeing the reliability and integrity of our evaluation results.
>
> [1] Li Z, Zhang X, Zhang Y, et al. Towards general text embeddings with multi-stage contrastive learning[J]. arXiv preprint arXiv:2308.03281, 2023.

---

> ### Author Response · Authors · 2025-11-24
> **Response to Reviewer 5L8W (6/16)**
>
> >> **Weaknesses-2-Q2**: Have the authors performed ablation studies to evaluate the effect of training data volume on test performance?
>
> >> **Questions-4**: Have the authors conducted ablation studies to quantify how training data volume affects final test accuracy? Would performance remain stable if a smaller, filtered subset of training data were used?
>
> Thank you sincerely for raising these thoughtful and valuable questions. Considering that Questions-4 and Weaknesses-2-Q2 address similar concerns, **so I have included my response to Questions-4 in the Weaknesses-2-Q2 section below.**
>
> **(1)  Regarding the scaling law of SFT training data**, we provide the analysis In Section 3.5 "Synthetic Data Refinement and Scaling Law" (Page 10, Lines 508--526), and Appendix A.7.3 (Pages 32--34, Lines 1721--1790, Figure 7) of our paper
>
> Specifically, we conducted systematic scalability experiments using the Qwen3-8B-Base model. As shown in the **table 6** below, when the training data scales from 2k to 300k, the model exhibits significant and consistent performance gains across all benchmarks:
>
> *   **AIME24**: Accuracy increases from 27.2% to 78.4%
> *   **AIME25**: Accuracy increases from 21.1% to 72.2%
> *   **Google-IMO-AnswerBench**: Accuracy increases from 5.3% to 28.8%
> *   **Overall Average**: Performance improves from 17.9% to 59.8% (+41.9%)
>
> These results demonstrate that our proposed tool-augmented data synthesis approach exhibits strong scalability and effectiveness.
>
> **Table 6: Performance across benchmarks about the scaling law of SFT training data.**
>
> | SFT Training Data Volume | AIME24(%) | AIME25(%) | Google-IMO-AnswerBench(%) | Avg Score(%) |
> | :--- | :---: | :---: | :---: | :---: |
> | AgentMath-8B-SFT-2k | 27.2 | 21.1 | 5.3 | 17.9 |
> | AgentMath-8B-SFT-20k | 60.5 | 53.3 | 12.8 | 42.2 |
> | AgentMath-8B-SFT-40k | 66.8 | 57.9 | 17.6 | 47.4 |
> | AgentMath-8B-SFT-80k | 69.2 | 61.8 | 21.4 | 50.8 |
> | AgentMath-8B-SFT-160k | 73.8 | 66.4 | 25.6 | 55.3 |
> | AgentMath-8B-SFT-300k | 78.4 | 72.2 | 28.8 | 59.8 |
>
> **(2) Regarding the scaling law of RL training data**, we conducted systematic scalability experiments based on the best-performing AgentMath-8B-SFT-300k model. To ensure fair comparison, all experiments were conducted using the same number of training steps (200 steps) and hyperparameter configurations, varying only the RL training data size (10k, 20k, 30k, 42k).
>
> As shown in the **table 7** below, when the RL training data scales from 10k to 42k, the model demonstrates consistent improvements across all benchmarks:
>
> *   **AIME24**: Accuracy increases from 80.8% to 85.3%
> *   **AIME25**: Accuracy increases from 75.5% to 80.4%
> *   **Google-IMO-AnswerBench**: Accuracy increases from 30.9% to 34.7%
> *   **Overall Average**: Performance improves from 62.4% to 66.8%
>
> These results demonstrate the effectiveness and scalability of our RL training data.
>
> **Table 7: Performance across benchmarks about the scaling law of RL training data.**
>
> | RL Training Data Volume | AIME24(%) | AIME25(%) | Google-IMO-AnswerBench(%) | Avg Score(%) |
> | :--- | :---: | :---: | :---: | :---: |
> | AgentMath-8B-SFT-300k | 78.4 | 72.2 | 28.8 | 59.8 |
> | AgentMath-8B-RL-10k | 80.8 | 75.5 | 30.9 | 62.4 |
> | AgentMath-8B-RL-20k | 82.7 | 77.8 | 32.4 | 64.3 |
> | AgentMath-8B-RL-30k | 84.2 | 79.0 | 33.7 | 65.6 |
> | AgentMath-8B-RL-42k | 85.3 | 80.4 | 34.7 | 66.8 |
>
> In summary, combining the scaling law experiments from both the SFT and RL stages, we demonstrate that as the tool-augmented training data size increases, AgentMath exhibits sustained and consistent performance improvements across all evaluation benchmarks, further validating the good scalability of AgentMath.
>
> Moreover, even when using a smaller and filtered subset of training data, the performance remains stable. For example, with 40k SFT training data, AgentMath-8B-SFT-40k achieves 66.8% on AIME24, 57.9% on AIME25, 17.6% on Google-IMO-AnswerBench, and an Avg Score of 47.4%; similarly, with 80k SFT training data, AgentMath-8B-SFT-80k achieves 69.2% on AIME24, 61.8% on AIME25, 21.4% on Google-IMO-AnswerBench, and an Avg Score of 50.8%.
>
> **We have added the above discussions about the Weaknesses-2 to Appendix A.12 of our latest upload of revised paper (pages 45–47, lines 2388–2497).**

---

> ### Author Response · Authors · 2025-11-24
> **Response to Reviewer 5L8W (7/16)**
>
> > **Weaknesses-3**: Unclear heuristic data synthesis: The data synthesis process involves multiple human heuristics (e.g., computational component segmentation, code complexity filtering), and several components appear to require iterative manual adjustment. These steps may limit scalability to new benchmarks or domains. Moreover, the data synthesis and cleaning costs seem high given the repeated use of LLMs.
>
> Thank you sincerely for raising these thoughtful and valuable questions. Please allow me to address them one by one.
>
> >> **Weaknesses-3-Q1**: Unclear heuristic data synthesis: The data synthesis process involves multiple human heuristics (e.g., computational component segmentation, code complexity filtering), and several components appear to require iterative manual adjustment.
>
> >> **Questions-5:** Many steps in the data synthesis pipeline involve human heuristics such as computational component segmentation and code complexity filtering. How much manual adjustment or iteration do these steps require?
>
> We sincerely  thank you for your valuable feedback. Considering that Questions-5 and Weaknesses-3-Q1 address similar concerns, **so I have included my response to Questions-5 in the Weaknesses-3-Q1 section below**. In response to your questions regarding computational component segmentation and code-complexity filtering, we provide a detailed clarification below and apologize for the insufficient clarity in presenting these critical design choices in our paper.
>
> **1. Computational Component Segmentation.**
>
> This strategy segments each long chain-of-thought response generated by DeepSeek-R1 into fixed-length chunks (e.g., 3k tokens), yielding $N$ segments $(S_1, S_2, S_3, \ldots, S_N)$. We then use DeepSeek-V3-0324 to independently perform tool-augmented rewriting on each segment. In our 346k-example synthetic SFT dataset, responses have an average length of 18.3k tokens; thus, each response is on average split into 6 segments.
>
> **Motivation.** In early experiments, we attempted to rewrite the full DeepSeek-R1 response without segmentation. This led to several issues:
>
> *   DeepSeek-R1 responses are excessively long.
> *   The instruction-following ability of DeepSeek-V3-0324 degrades markedly under ultra-long inputs, and the model becomes more prone to hallucinations.
> *   Consequently, the rewritten data exhibits a low frequency of tool usage, a high code execution failure rate, and extensive abbreviation or omission of intermediate natural language reasoning steps (which we refer to as the "textual reasoning omission rate").
>
> As shown in the **table 8** below, to quantitatively assess the effect of segment length on data quality and downstream performance, we conducted controlled experiments on 20k synthetic examples and performed SFT on Qwen3-8B-Base. The results are summarized below:
>
> *   **No segmentation (AgentMath-SFT-20k-No-Segment):** The average number of tool calls is 1.6; the code execution failure rate is 31.8%; the textual reasoning omission rate (the proportion of intermediate reasoning steps omitted) is 78.5%; and AIME24/AIME25 accuracies are 40.3%/35.6%.
> *   **Segmentation with 9k tokens (AgentMath-SFT-Segment-with-9k-token):** This setting improves over no segmentation, but segments remain relatively long, so tool usage and evaluation performance are still limited.
> *   **Segmentation with 3k tokens (AgentMath-SFT-Segment-with-3k-token):** The average number of tool calls increases to 8.3; the code execution failure rate drops to 6.2%; the textual reasoning omission rate decreases to 2.3%; and AIME24/AIME25 accuracies reach 60.5%/53.3%.
> *   **Segmentation with 2k tokens (AgentMath-SFT-Segment-with-2k-token):** Although the number of tool calls further increases, downstream performance does not improve significantly and appears to saturate.
>
> Based on this analysis, we **adopt a 3k-token segment length** as the default configuration for computational component segmentation, as it strikes a favorable balance: it effectively controls data-quality metrics while maximizing downstream task performance.
>
> **Table 8: Results on 20k AgentMath synthetic data about Computational Component Segmentation.**
>
> | 20k AgentMath synthetic data | Mean Tool Calls | Code Execution Failure Rate | Textual Reasoning Omission Rate | AIME24 Acc | AIME25 Acc |
> | :--- | :---: | :---: | :---: | :---: | :---: |
> | AgentMath-SFT-20k-No-Segment | 1.6 | 31.8% | 78.5% | 40.3% | 35.6% |
> | AgentMath-SFT-Segment-with-9k-token | 2.1 | 28.4% | 67.9% | 44.8% | 38.7% |
> | AgentMath-SFT-Segment-with-6k-token | 3.2 | 23.3% | 38.6% | 49.2% | 44.3% |
> | AgentMath-SFT-Segment-with-4k-token | 5.7 | 12.5% | 8.8% | 57.3% | 51.6% |
> | AgentMath-SFT-Segment-with-3k-token | 8.3 | 6.2% | 2.3% | 60.5% | 53.3% |
> | AgentMath-SFT-Segment-with-2k-token | 10.1 | 5.9% | 2.1% | 59.4% | 51.9% |

---

> ### Author Response · Authors · 2025-11-24
> **Response to Reviewer 5L8W (8/16)**
>
> **[ Continue the response to above Weaknesses-3-Q1 and Questions-5]**
>
> **2. Code Complexity Filtering.**
>
> At an early stage, we conducted a statistical analysis of code-length distributions over 20k synthetic instances. The results showed that samples with fewer than 5 lines of code constituted 5% of the dataset, with exactly 5 lines representing 7%, 6 lines 13%, 7 lines 24%, 8 lines 29%, 9 lines 13%, and 10 or more lines comprising 10%.
>
> To investigate the relationship between code complexity and line count, we performed stratified uniform sampling, drawing 40 samples from each line-count category (280 samples total). Three authors independently annotated each sample's complexity as "Easy," "Medium," or "Hard" based on whether it involved computationally demanding operations (e.g., large-number arithmetic, complex equation solving, advanced linear algebra, combinatorial computations, or calculus). The annotation statistics are presented in the **table 9** below.
>
> Our analysis shows that among samples with $\leq 5$ lines of code, over 50% were labeled "Easy" while fewer than 10% were labeled "Hard." To prevent the model from learning to overuse tool invocations during training, **we conservatively filtered out all samples containing 5 or fewer lines of code.** In future work, we plan to employ LLMs such as Qwen3-30B for automated complexity assessment, replacing manual annotation to improve both objectivity and scalability of our filtering pipeline.
>
> **Table 9: The statistics of code complexity by lines of code.**
>
> | Lines of Code | Easy (%) | Medium (%) | Hard (%) |
> | :--- | :---: | :---: | :---: |
> | $< 5$ lines | 87% | 9% | 4% |
> | $= 5$ lines | 66% | 27% | 7% |
> | $= 6$ lines | 43% | 38% | 19% |
> | $= 7$ lines | 24% | 49% | 27% |
> | $= 8$ lines | 13% | 56% | 31% |
> | $= 9$ lines | 7% | 51% | 42% |
> | $\geq 10$ lines | 4% | 43% | 53% |
>
>
> >> **Weaknesses-3-Q2**: These steps may limit scalability to new benchmarks or domains.
>
> >> **Questions-6**: Can the synthesis and cleaning pipeline be generalized or automated for other benchmarks or non-math domains?
>
> Thank you so much for your constructive question. Considering that Questions-6 and Weaknesses-3-Q2 address similar concerns, **so I have included my response to Questions-6 in the Weaknesses-3-Q2 section below.**
>
> In response, we have extended AgentMath's automated data synthesis and cleaning pipeline to **non-mathematical domains, including physics, chemistry, and biology**, and evaluated the resulting model on the **GPQA Diamond benchmark**. GPQA Diamond assesses large language models' ability to solve graduate-level problems across biology, physics, and chemistry--a highly challenging benchmark where even PhD-level domain experts achieve only 65% accuracy.
>
> We randomly sampled 40k examples from the open-source Science 220k dataset released by AM-Thinking[1] and applied AgentMath's data synthesis method. Specifically, we segmented DeepSeek-R1's reasoning response into 3k-token chunks and transformed them into tool-augmented reasoning chains by replacing complex, error-prone computational steps with executable code. We then applied our automated cleaning pipeline, which includes: (1) filtering examples with $\leq 5$ lines of code, (2) correcting formatting errors, (3) removing samples with failed code execution, and (4) performing environmental feedback alignment to ensure consistency.
>
> Using Qwen3-8B-Base as the foundation model, we conducted SFT and evaluated performance on GPQA Diamond across 8 independent runs, reporting the average accuracy (Avg@8). As shown in the **table 10** below, AgentMath-8B-SFT-40k achieves 58.9% accuracy on GPQA Diamond, substantially outperforming DeepSeek-R1-Distill-Qwen-7B (49.1%).
>
> These results demonstrate that **AgentMath's tool-augmented synthesis approach and automated cleaning pipeline generalize effectively beyond mathematics to scientific domains (physics, chemistry, and biology)**, highlighting the scalability and broad applicability of our method.
>
> **Table 10: The GPQA accuracy of different models and AgentMath-8B models at the 7B-8B scale.**
>
> | Model | Base Model | GPQA Diamond Acc |
> | :--- | :--- | :---: |
> | OpenThinker-7B | Qwen2.5-7B-Instruct | 42.4% |
> | AReaL-boba-RL-7B | DeepSeek-R1-Distill-Qwen-7B | 47.6% |
> | DeepSeek-R1-Distill-Qwen-7B | Qwen2.5-Math-7B | 49.1% |
> | Light-R1-7B | DeepSeek-R1-Distill-Qwen-7B | 49.4% |
> | AgentMath-8B-SFT-40k | Qwen3-8B-Base | 58.9% |
>
> [1] https://huggingface.co/datasets/a-m-team/AM-DeepSeek-R1-0528-Distilled

---

> ### Author Response · Authors · 2025-11-24
> **Response to Reviewer 5L8W (9/16)**
>
> >> **Weaknesses-3-Q3**: Moreover, the data synthesis and cleaning costs seem high given the repeated use of LLMs.
>
> >> **Questions-7**: What is the total computational cost (e.g., GPU hours or LLM API calls) associated with synthesizing and verifying the datasets?
>
> Thank you for pointing out this valuable question. Considering that Questions-7 and Weaknesses-3-Q3 address similar concerns, **so I have included my response to Questions-7 in the Weaknesses-3-Q3 section below.** Below, we present a detailed analysis about the total computational cost (e.g., GPU hours or LLM API calls) associated with synthesizing and verifying the datasets
>
> We build upon the open-source AM-Thinking dataset, leveraging their existing responses generated by DeepSeek-R1-0528 to obtain 346K data samples. We employ DeepSeek-V3 for data synthesis, completing this process on 128 GPUs (each with 96GB memory) in approximately 62 hours. We then utilize Qwen3-30B as a judge model to verify consistency between actual code execution results and model-simulated outputs, requiring approximately 3 hours.
>
> Subsequently, we perform SFT on the 316K synthetic samples using Qwen3-8B-Base as the base model, which takes approximately 44 hours. In total, the complete pipeline---encompassing data synthesis, cleaning, verification, and SFT training, requires **approximately 62 + 3 + 44 = 109 hours.**
>
> Notably, we also explore a more cost-effective alternative to address computational overhead. As illustrated in Figure 4 (page 10, lines 486–499) and detailed in Section 3.5 (page 9, lines 453–473) of our paper, we apply SFT to only 20K samples starting from Qwen3-8B-Base, followed by 400 steps of large-scale reinforcement learning. This streamlined approach enables our AgentMath model to achieve the leading performance: 76.2% on AIME24 and 67.5% on AIME25, surpassing both OpenMath-Nemotron-7B (74.8% and 61.2%, respectively) and Qwen3-8B-Thinking (76.0% and 67.3%, respectively). This entire workflow, including data synthesis, cleaning, SFT, and RL training, requires only about 76 hours total, providing a substantially more efficient solution.
>
> **We have added the above discussions about the Weaknesses-3 to Appendix A.13 of our latest upload of revised paper (pages 47–49, lines 2498–2624).**
>
> > **Weaknesses-4**: The paper does not clarify how textual answers are segmented into computational parts or how the model determines when code usage is beneficial. Different models may prefer distinct code-usage patterns even for the same question.
>
> We sincerely appreciate your attention to our work and your careful and responsible review and thank you for your insightful questions.  Please allow me to address them one by one.
>
> >> **Weaknesses-4-Q1**: The paper does not clarify how textual answers are segmented into computational parts
>
> >> **Questions-8-Q1**: How are textual reasoning steps segmented into computational components when deciding where to inject code?
>
> Thank you for raising this important question. We sincerely apologize for any confusion this may have caused and provide a detailed clarification below. Considering that Questions-8-Q1 and Weaknesses-4-Q1 address similar concerns, **so I have included my response to Questions-8-Q1 in the Weaknesses-4-Q1 section below.**
>
> We utilized the long chain-of-thought responses generated by DeepSeek-R1 from the open-source AM-Thinking dataset. Each response is segmented into fixed-length fix-token length(i.e., 3k token length) chunks, yielding $N$ segments $(S_1, S_2, S_3, \ldots, S_n)$. Across the 316k SFT synthetic data, the responses have an average length of 16.9k tokens, corresponding to roughly six segments per response.
>
> Subsequently, for each segment, we applied **the Tool-Augmented Data Synthesis Prompt detailed in Appendix A.6.2 (page 29, lines 1512–1565) of our paper.** We employed DeepSeek-V3-0324 to independently transform each segment by replacing complex and error-prone computational processes with executable code.
>
> To accelerate the synthesis process, we initially instructed DeepSeek-V3-0324 to simulate code execution results based on the textual reasoning context. During post-processing, we replaced these simulated results with actual code execution outputs and employed Qwen3-30B as a judge model to assess their consistency. Samples exhibiting inconsistencies were filtered out to ensure data quality.
>
> **For a more detailed elaboration on computational component segmentation, please refer to our response to Weaknesses-3-Q1 regarding Computational Component Segmentation above.**
>
>
> >> **Weaknesses-4-Q2**: how the model determines when code usage is beneficial.
>
> Thank you for this valuable question. We present the design of the Tool-Augmented Data Synthesis Prompt in Appendix A.6.2 (page 29, lines 1512–1565) of our paper. This prompt guides DeepSeek-V3-0324 to first identify complex and error-prone computational processes, and then replace these manual calculations with code.

---

> ### Author Response · Authors · 2025-11-24
> **Response to Reviewer 5L8W (10/16)**
>
> **[ Continue the response to above Weaknesses-4-Q2]**
>
> Specifically, it covers the following types of **complex computational tasks**:
>
> *   **Complex Symbolic Algebra**: polynomial expansion, factorization, equation solving
> *   **Advanced Calculus**: differentiation, integration, definite integral computation
> *   **Probability and Combinatorics**: complex counting problems, probability distribution calculations
> *   **Linear Algebra**: matrix operations, matrix inversion, eigenvalue decomposition
> *   **Numerical Computation**: approximation calculations, large number arithmetic, geometric calculations
> *   **Other Error-Prone or Computation-Intensive Problems**
>
> To control code complexity and ensure the reliability of the synthesized data, we further filter out low-complexity samples with $\leq 5$ lines of code. Further details are provided in the "Weaknesses-3-Q1: Code Complexity Filtering" section above.
>
> Furthermore, we randomly sampled 100 instances from the synthesized data and manually annotated whether DeepSeek-V3 effectively replaced complex textual computational processes with code based on the principles of complex computing tasks.
>
> The results show that **code replacement is beneficial and reasonable in 84% of the samples**, thereby confirming the reliability of our proposed tool-augmented data synthesis approach.
>
> >> **Questions-8-Q2**: What criteria or metrics determine whether using code is more effective for a given part of the reasoning chain?
>
> We sincerely thank you for raising this insightful question. Below, we demonstrate the effectiveness of code-augmented methods in replacing text-based complex computations from **two perspectives:**
>
> **(1)** We randomly sampled 100 instances from the synthetic data and annotated whether DeepSeek-V3 effectively employs code to replace manual complex computations based on the following criteria. The evaluation reveals that **code-based computation proves more effective in 84% of cases.** The evaluation criteria encompass the following mathematical domains:
>
> *   Complex symbolic algebra: polynomial expansion, factorization, equation solving
> *   Advanced calculus: differentiation, integration, definite integral computation
> *   Complex probability and combinatorics: complex counting, probability distributions
> *   Complex linear algebra: matrix operations, matrix inversion, eigenvalue decomposition
> *   Complex numerical computation: approximation, large number arithmetic, geometric calculations
> *   Any error-prone or computationally intensive operations
>
> **(2)** We systematically compare code-based and pure text-based reasoning along two dimensions: **inference efficiency (measured by average token count) and performance metrics**, as shown in the **table 11** below:
>
> *   **Token Analysis on Synthetic Data**: The SFT training dataset comprises 316k instances in total. Pure text-based reasoning averages 18.3k tokens per sample, while tool augmentation reduces this to 16.9k tokens. This demonstrates that substituting code for lengthy computational processes effectively reduces overall token consumption.
> *   **SFT Stage**: As detailed in Section 3.5 (page 9, lines 453–472) and Appendix A.7.1 (page 31, lines 1644–1673) in our paper, we comprehensively compare the efficiency and performance of code augmentation against pure text-based reasoning. Specifically, when trained on identical 20k SFT data, AgentMath-SFT-20k achieves accuracies of 60.5% and 53.3% on AIME24 and AIME25, respectively, significantly outperforming Text-based-SFT-20k (57.1% and 49.2%). Additionally, AgentMath-SFT-20k **reduces the average token count by approximately 1.3k–3k** compared to Text-based-SFT-20k.
> *   **RL Stage**: In tool-augmented RL training, using only 440 steps, AgentMath-RL-440-steps improves accuracy from 60.5% to 76.2% on AIME24 and from 53.3% to 67.5% on AIME25. In contrast, pure text-based RL training requires 1600 steps (Text-based-RL-1600-steps) to achieve improvements from 57.1% to 68.7% on AIME24 and from 49.2% to 57.5% on AIME25. These results demonstrate that the code-augmented approach achieves **approximately $4\times$ higher training efficiency** while **reducing the overall token count by 2.5k–5k** compared to Text-based-RL-1600-steps.
>
> **Table 11: Accuracy and average inference token usage on AIME24 and AIME25 for pure text-based models and code-augmented AgentMath at both SFT and RL stages.**
>
> | Model | Base Model | AIME24 Acc | AIME24 Avg Tokens | AIME25 Acc | AIME25 Avg Tokens |
> | :--- | :--- | :---: | :---: | :---: | :---: |
> | Text-based-SFT-20k | Qwen3-8B-Base | 57.1% | 29k | 49.2% | 28k |
> | AgentMath-SFT-20k | Qwen3-8B-Base | 60.5% | 26k | 53.3% | 26.7k |
> | Text-based-RL-1600-steps | Text-based-SFT-20k | 68.7% | 32.3k | 57.5% | 31.2k |
> | AgentMath-RL-440-steps | AgentMath-SFT-20k | 76.2% | 27.2k | 67.5% | 28.6k |

---

> ### Author Response · Authors · 2025-11-24
> **Response to Reviewer 5L8W (11/16)**
>
> **[Continue the response to above Questions-8-Q2]**
>
> **In summary, introducing code augmentation in both SFT and RL stages yields substantial improvements in performance and efficiency over pure text-based reasoning, validating that using code is more effective for a given part of the reasoning chain.**
>
> >> **Weaknesses-4-Q3**: Different models may prefer distinct code-usage patterns even for the same question.
>
> >> **Questions-9**: Since different models can prefer different code usage patterns for the same question, how robust is the segmentation procedure across models?
>
> Thank you so much for this insightful comment. We agree that different models can prefer different code usage patterns for the same question, including variations in code injection positions, line counts, and code implementation details. Considering that Questions-9 and Weaknesses-4-Q3 address similar concerns, **so I have included my response to Questions-9 in the Weaknesses-4-Q3 section below.**
>
> In our early experiments, we conducted a comparative study between two candidate teacher models: DeepSeek-V3-0324 and Qwen3-235B-Instruct-2507. Specifically, we synthesized training data using each model on the 20k training set and performed supervised fine-tuning on Qwen3-8B-Base. As shown in the **table 12** below, the experimental results are as follows:
>
> *   Data synthesized by DeepSeek-V3-0324 averaged 6.5 tool calls per sample, compared to 5.3 tool calls per sample for Qwen3-235B-Instruct-2507.
> *   On the AIME24 and AIME25 benchmarks, the model trained on DeepSeek-V3-0324-synthesized data (AgentMath-20k-SFT-DeepSeek-V3) achieved accuracies of 60.5% and 53.3%, respectively, marginally outperforming the model trained on Qwen3-235B-Instruct-2507-synthesized data (59.3% and 51.7%). Notably, the former also demonstrated a higher average tool call frequency.
>
> Based on these comprehensive performance results, we selected DeepSeek-V3-0324 as the teacher model for our data synthesis pipeline. **Meanwhile, using different models (i.e., Qwen3-235B-Instruct-2507) for segmentation also achieves good and robust performance.**
>
> **We have added the above discussions about the Weaknesses-4 to Appendix A.14 of our latest upload of revised paper (pages 49–51, lines 2625–2732).**
>
> **Table 12: Comparison of Qwen3-8B-Base fine-tuned on the 20k AgentMath SFT dataset synthesized by DeepSeek-V3-0324 and Qwen3-235B-Instruct-2507.**
>
> | Model | AIME24 | AIME24 Tool Calls Avg | AIME25 | AIME25 Tool Calls Avg |
> | :--- | :---: | :---: | :---: | :---: |
> | AgentMath-20k-SFT-DeepSeek-V3 | 60.5% | 5.1 | 53.3% | 5.7 |
> | AgentMath-20k-SFT-Qwen3-235B-Instruct-2507 | 59.3% | 4.4 | 51.7% | 5.0 |
>
> > **Weaknesses-5**: The dataset synthesis employs larger teacher models (DeepSeek R1 and V3, Qwen-30B). This may imply knowledge distillation from more powerful teacher models. Comparing such distilled models with those trained without teacher guidance may be unfair.
>
> > **Questions-10-Q1**: The paper uses large teacher models (DeepSeek R1, DeepSeek V3, and Qwen-30B) for data synthesis. How do the authors ensure fairness when comparing with models trained without such teacher guidance?
>
> Thank you for your valuable advice. Considering that Questions-10-Q1 and Weaknesses-5 address similar concerns, **so I have included my response to Questions-10-Q1 in the Weaknesses-5 section below.**
>
> To ensure a rigorous and fair comparison, we strictly followed the same experimental setup of Retool: the same 2k synthetic dataset, same teacher model responses from DeepSeek-R1, the same base model (Qwen2.5-32B-Instruct), and 400 training steps during the RL phase. The sole difference lies in the data synthesis strategy: AgentMath leverages our proposed tool-augmented data synthesis method, whereas Retool employs its original synthesis prompt. Both approaches undergo identical two-stage training comprising SFT and RL. As shown in the **table 13** below, the experimental results are summarized as follows:
>
> *   AgentMath-32B-SFT-2k achieves 44.1% on AIME24 and 37.3% on AIME25, surpassing Retool-32B-SFT-2k (40.9% and 34.5%, respectively), thereby demonstrating the effectiveness of our tool-augmented data synthesis approach;
> *   AgentMath-32B-RL attains 74.8% on AIME24 and 56.6% on AIME25, consistently outperforming Retool-32B-RL (67.0% and 49.3%, respectively), further validating the superiority of our reinforcement learning training strategy.
>
> **In summary, compared to Retool, AgentMath exhibits consistent and substantial performance gains across both training stages when using the same teacher model, confirming the efficacy and robustness of our proposed method.**
>
> **Table 13: Comparison of AgentMath and Retool performance under the same experimental setup in both the SFT and RL stages.**
>
> | Model | AIME24 Acc | AIME25 Acc |
> | :--- | :---: | :---: |
> | Retool-32B-SFT-2k | 40.9% | 34.5% |
> | AgentMath-32B-SFT-2k | 44.1% | 37.3% |
> | Retool-32B-RL | 67.0% | 49.3% |
> | AgentMath-32B-RL | 74.8% | 56.6% |

---

> ### Author Response · Authors · 2025-11-24
> **Response to Reviewer 5L8W (12/16)**
>
> > **Questions-10-Q2**: Would the performance gap shrink if competing models were also allowed to distill from comparable teacher models?
>
> Thank you so much for your important question. We adopted the open-source dataset from AM-Thinking[1], using the same 316k prompts and responses generated by DeepSeek-R1. We conducted supervised fine-tuning (SFT) on the Qwen3-8B-Base model for 6 epochs with a learning rate of 6e-5. To ensure experimental rigor, all configurations were kept identical except for the core difference in reasoning chain types: AM-Thinking[1] uses pure textual reasoning chains, while AgentMath employs tool-augmented reasoning chains. We conducted 8 independent training runs and report pass@1 results from the checkpoint with the best validation performance.
>
> As shown in **table 14**, AgentMath-8B-SFT achieves 78.4% accuracy on AIME24 and 72.2% accuracy on AIME25, outperforming AM-Thinking-8B-SFT (74.9% and 67.4%, respectively), with an overall average performance gap of 4.1%. **These results confirm that the performance gap will shrink if competing models were also allowed to distill from comparable teacher models.**
>
> **We have added the above discussions about the Weaknesses-5 to Appendix A.15 of our latest upload of revised paper (pages 51–52, lines 2733–2795).**
>
> **Table 14: Comparison of pass@1 accuracy on AIME24 and AIME25 for AM-Thinking-8B-SFT and AgentMath-8B-SFT under the same training settings.**
>
> | Model | AIME24 Acc | AIME25 Acc | Avg Score |
> | :--- | :---: | :---: | :---: |
> | AM-Thinking-8B-SFT | 74.9% | 67.4% | 71.2% |
> | AgentMath-8B-SFT | 78.4% | 72.2% | 75.3% |
>
> [1] https://huggingface.co/datasets/a-m-team/AM-DeepSeek-R1-0528-Distilled
>
> > **Weaknesses-6**：The authors should consider moving key details, such as dataset sizes for SFT and RL, to the main paper because they are critical for comprehension.
>
> We sincerely thank you for your thoughtful and constructive suggestions. **In response, we have moved the SFT and RL data construction process, data sizes, and training details to Sections 3.1, 3.2, and 3.3 in our latest upload of revised paper (Page 7, Lines 326–377).** We hope this reorganization provides you and other researchers with a more comprehensive and clear understanding of the technical details of our work.
>
> > **Weaknesses-7**: Many designs in the algorithms, such as the four quality refinement modules in Step 2 of the data set synthesis, and reward design in RL, need further justification or ablation studies. Otherwise the overall method seems to contain too many ad-hoc details that may limit the generalizability of the proposed approach.
>
> Thank you very much for bringing up this valuable suggestion, and we sincerely apologize for any inconvenience caused by the algorithm design details. Below, we  present detailed ablation studies on the four quality refinement modules in Step 2 of the dataset synthesis and the reward design in RL.
>
> **1. Four Quality Refinement Modules For Data Synthesis**
>
> We sincerely thank you for your valuable feedback. Below, we present a detailed ablation analysis of the four quality refinement modules in our data synthesis process to elucidate the individual contribution of each component.
>
> **In Section 3.5 "Synthetic Data Refinement and Scaling Law" (Page 10, Lines 508–526), Table 3, and Appendix A.7.3 (Pages 32–34, Lines 1721–1790) of our paper**, we present the discussion of the four quality refinement modules employed in our data synthesis pipeline. Our experiments are conducted using the Qwen3-8B-Base model on 20k synthetic data samples, with results shown in the **table 15** below.
>
> The initial unrefined synthetic data (20k samples) achieves accuracies of 35.3% and 25.7% on AIME24 and AIME25, respectively, with performance primarily limited by issues such as format inconsistencies and non-executable code. We then progressively introduce the following refinement modules:
>
> *   **Format consistency correction**: Accuracy improves to 47.4% on AIME24 and 40.1% on AIME25;
> *   **Code executability verification**: Accuracy further increases to 52.8% on AIME24 and 44.8% on AIME25;
> *   **Environmental feedback alignment**: Accuracy reaches 56.3% on AIME24 and 48.3% on AIME25;
> *   **Tool-usage rationality assessment**: Accuracy achieves 57.2% on AIME24 and 48.9% on AIME25.
>
> **Table 15: Ablation study of quality refinement modules showing progressive accuracy improvements on AIME24 and AIME25 benchmarks.**
>
> | Models / Refinement Steps | AIME24 | AIME25 |
> | :--- | :---: | :---: |
> | Initial Unrefined CI-Synthetic Data (20k) | 35.3% | 25.7% |
> | + Format consistency correction | 47.4% | 40.1% |
> | + Code executability verification | 52.8% | 44.8% |
> | + Environmental feedback alignment | 56.3% | 48.3% |
> | + Tool-usage rationality assessment | 57.2% | 48.9% |
> | + Self-correction capability injection | 58.6% | 50.8% |
> | + SFT with selective feedback masking | 60.5% | 53.3% |

---

> ### Author Response · Authors · 2025-11-24
> **Response to Reviewer 5L8W (13/16)**
>
> **[ Continue the response to above Weaknesses-7]**
>
> Finally, by incorporating self-correction capability injection and supervised fine-tuning with selective feedback masking, the model achieves accuracies of 60.5% and 53.3% on AIME24 and AIME25, respectively. Overall, through this multi-dimensional quality refinement process, accuracies improve from 35.3% to 60.5% on AIME24 and from 25.7% to 53.3% on AIME25, which clearly demonstrates the critical role and effectiveness of each refinement module in optimizing data quality and establishes a solid foundation for deriving scaling laws for synthetic data.
>
> **2. Reward Design in RL**
>
> We thank you for your valuable suggestions. Below, we provide more detailed explanations of the reward design and corresponding experimental validation.
>
> Our reward function consists of two components: an answer correctness reward $R_{\\text{acc}}$ and a tool usage efficiency reward $R_{\\text{tool}}$. The correctness reward $R_{\\text{acc}}$ provides binary (0,1) feedback based on mathematical equivalence (verified using the math_verify library):
>
> $$
> R_{\\text{acc}} =
> \\begin{cases}
> 1, & \\text{if } \\mathrm{is\\_equivalent}(\\hat{a}, a),\\\\
> 0, & \\text{otherwise},
> \\end{cases}
> $$
>
> where $\\hat{a}$ denotes the predicted answer and $a$ the ground truth answer. Conditioned on answer correctness, the tool usage reward $R_{\\text{tool}}$ incentivizes efficient utilization of computational resources:
>
> $$
> R_{\\text{tool}}=\\min\\!\\big(R_{\\text{max}},\\, \\alpha+\\beta\\cdot N_{\\text{code}}\\big)
> \\quad\\text{if } N_{\\text{code}}>0,
> $$
>
> where $\\alpha$ is the base tool usage reward, $\\beta$ is the scaling factor for the number of invocations, and $R_{\\text{max}}$ is the reward upper bound. The total reward function is defined as:
>
> $$
> R_{\\text{total}}=R_{\\text{acc}}+\\mathbb{I}(R_{\\text{acc}}=1)\\cdot R_{\\text{tool}}.
> $$
>
> In our work, we set $\\alpha=0.1$ and $\\beta=0.01$, and constrain the maximum tool reward to $R_{\\text{max}}=1$, which encourages tool usage while preventing over-reliance. This design is motivated by the following considerations: applying $R_{\\text{tool}}$ only to correct answers encourages the model to use tools appropriately to arrive at accurate solutions, avoiding rewarding erroneous tool invocations; and $\\alpha=0.1$ differentiates between cases with and without tool usage, while the smaller $\\beta=0.01$ accommodates high-frequency tool calls during RL training (some exceeding 64 invocations), preventing reward inflation.
>
> In early experiments, to validate the effectiveness of the reward design, we conducted controlled experiments using the SFT model trained on 20k synthetic data (AgentMath-SFT-20k), with a fixed 100 training steps, as shown in the **table 16** below. Our experiments demonstrate:
>
> * **Effectiveness of tool rewards:** Introducing tool rewards (all answers) outperforms the no-tool-reward baseline in both average accuracy (61.4% vs. 60.2%) and average tool calls (6.2 vs. 5.7), indicating that tool rewards effectively incentivize the model to leverage computational tools and improve performance.
> * **Advantage of tool rewards (correct answers only):** Furthermore, compared to tool rewards (all answers), applying tool rewards exclusively to correct answers ($\\alpha=0.1, \\beta=0.01$) achieves the best performance in both average accuracy (62.6% vs. 61.4%) and average tool calls (6.3 vs. 6.2).
> * **Optimal hyperparameter configuration:** In a grid search over $\\alpha \\in \\{0.1, 0.2\\}$ and $\\beta \\in \\{0.01, 0.02\\}$, $\\alpha=0.1, \\beta=0.01$ achieves the best accuracy-efficiency trade-off (62.6% average accuracy, 6.3 average tool calls).
>
> Therefore, based on performance and tool call metrics, we ultimately **adopt the correctness-conditioned tool reward design with parameters $\\alpha=0.1$ and $\\beta=0.01$ for RL training.**
>
> **We have added the above discussions about the Weaknesses-7 to Appendix A.16 of our latest upload of revised paper (pages 52–53, lines 2796–2844).**
>
> **Table 16: Ablation study results on reinforcement learning reward design.**
>
> | Reward Configuration | AIME24 Accuracy | AIME24 Tool Calls | AIME25 Accuracy | AIME25 Tool Calls | Avg. Accuracy | Avg. Tool Calls |
> | :--- | :---: | :---: | :---: | :---: | :---: | :---: |
> | AgentMath-SFT-20k (Baseline) | 60.5% | 5.1 | 53.3% | 5.7 | 56.9% | 5.4 |
> | RL with No tool reward | 65.2% | 5.5 | 55.1% | 5.9 | 60.2% | 5.7 |
> | RL with Tool reward (all answers), $\\alpha=0.1$, $\\beta=0.01$ | 66.4% | 6.1 | 56.3% | 6.3 | 61.4% | 6.2 |
> | **RL with Tool reward (correct answers only), $\\alpha=0.1$, $\\beta=0.01$** | **67.4%** | **6.0** | **57.7%** | **6.5** | **62.6%** | **6.3** |
> | RL with Tool reward (correct answers only), $\\alpha=0.1$, $\\beta=0.02$ | 66.7% | 6.4 | 56.8% | 6.7 | 61.8% | 6.6 |
> | RL with Tool reward (correct answers only), $\\alpha=0.2$, $\\beta=0.01$ | 66.9% | 5.9 | 56.5% | 6.2 | 61.7% | 6.1 |

---

> ### Author Response · Authors · 2025-11-24
> **Response to Reviewer 5L8W (14/16)**
>
> > **Weaknesses-8**: Typo: In line 1600, the figure reference is missing.
>
> We sincerely appreciate you pointing out the typos in our original paper and for your thorough and meticulous review. **The typo regarding the missing figure reference has been corrected in Appendix A.7.2 (page 32, line 1708) in our latest upload of revised paper.**
>
> > **Questions-1**: Given that AIME24, AIME25, and HMMT each contain fewer than 30 questions and have been reported as leaked in prior work, how do the authors ensure that these benchmarks were not inadvertently seen during pretraining or fine-tuning? Have the authors considered using larger or newly constructed evaluation sets to validate the robustness of the results?
>
> Thank you so much for your valuable question. I've noticed that Questions-1 and Weaknesses-1 address similar concerns. To avoid redundancy, **I've integrated the discussion of Questions-1 into the section for Weaknesses-1 above.** Please feel free to **search for "Weaknesses-1" or "Questions-1" tag** in the full text to find the above detailed response.
>
> > **Questions-2**: With 346k training questions for SFT and 42k for RL but fewer than 100 test questions, how do the authors ensure that the training data do not overlap or contain structurally similar questions to the test set?
>
> I'm grateful for your thoughtful question. I realized that Questions-2 and Weaknesses-2-Q1 touch upon similar points. To avoid redundancy, **I've integrated the discussion of Questions-2 into the section for Weaknesses-2-Q1 above.** Please feel free to **search for "Weaknesses-2-Q1" or "Questions-2" tag** in the full text to find the above detailed response.
>
> > **Questions-3**: What exact n-gram filtering algorithm and similarity threshold were used to remove overlapping data?
>
> Thank you for raising this important question. We employed the **4-gram(N=4)** filtering algorithm and **set the similarity threshold to 0.6** to remove overlapping data. More detailed deduplication procedures have been thoroughly described in Weaknesses-2-Q1 above and please feel free to **search for "Weaknesses-2-Q1" tag** in the full text. We hope this can address your concern.
>
> > **Questions-4**: Have the authors conducted ablation studies to quantify how training data volume affects final test accuracy? Would performance remain stable if a smaller, filtered subset of training data were used?
>
> Thank you very much for this excellent question. I found that Questions-4 and Weaknesses-2-Q2 address similar concerns, so **I have included my response to Questions-4 in the Weaknesses-2-Q2 section above.** Please **search for "Weaknesses-2-Q2" or "Questions-4" tag** in the full text for the above detailed discussion.
>
> Specifically, **as the volume of tool-augmented synthetic data scales from 2k to 300k, the accuracy across all test sets continues to improve. Moreover, even when using a smaller and filtered subset of training data, the performance remains stable.** For example, with 40k training data, AgentMath-8B-SFT-40k achieves 66.8% on AIME24, 57.9% on AIME25, 17.6% on Google-IMO-AnswerBench, and an Avg Score of 47.4%; similarly, with 80k training data, AgentMath-8B-SFT-80k achieves 69.2% on AIME24, 61.8% on AIME25, 21.4% on Google-IMO-AnswerBench, and an Avg Score of 50.8%. This demonstrates that AgentMath exhibits good scalability.
>
> > **Questions-5**: Many steps in the data synthesis pipeline involve human heuristics such as computational component segmentation and code complexity filtering. How much manual adjustment or iteration do these steps require?
>
> Thank you for raising this pertinent question. Recognizing the similarity between Questions-5 and Weaknesses-3-Q1,  **I have incorporated Questions-5 into the Weaknesses-3-Q1 section above for convenience.** Please **search "Weaknesses-3-Q1" or "Questions-5" tag** in the full text for the above detailed discussion.
>
> > **Questions-6**: Can the synthesis and cleaning pipeline be generalized or automated for other benchmarks or non-math domains?
>
> Yes,  **the synthesis and cleaning pipeline can be generalized or automated for other benchmarks or non-math domains(i.e., physics, chemistry, and biology).**
>
> To demonstrate this, we extended AgentMath's automated data synthesis and cleaning pipeline to non-mathematical domains, including physics, chemistry, and biology, and evaluated the resulting model on the GPQA Diamond benchmark. The results demonstrate that AgentMath's tool-augmented synthesis approach and automated cleaning pipeline generalize effectively beyond mathematics to scientific domains (physics, chemistry, and biology), highlighting the scalability and broad applicability of our method.
>
> Note: As Questions-6 and Weaknesses-3-Q2 concern similar points, **I have addressed them together by incorporating my response to Questions-6 into the Weaknesses-3-Q2 section above.** For the more detailed discussion, please **search for "Weaknesses-3-Q2" or "Questions-6" tag** in the full text.

---

> ### Author Response · Authors · 2025-11-24
> **Response to Reviewer 5L8W (15/16)**
>
> > **Questions-7**: What is the total computational cost (e.g., GPU hours or LLM API calls) associated with synthesizing and verifying the datasets?
>
> Thank you very much for this excellent question. We build upon the open-source AM-Thinking[1] dataset, leveraging their existing responses generated by DeepSeek-R1-0528 to obtain 346K data samples. We employ DeepSeek-V3 for data synthesis, completing this process on 128 GPUs (each with 96GB memory) in approximately 62 hours. We then utilize Qwen3-30B as a judge model to verify consistency between actual code execution results and model-simulated outputs, requiring approximately 3 hours.
>
> Subsequently, we perform SFT on the 316K synthetic samples using Qwen3-8B-Base as the base model, which takes approximately 44 hours. In total, the complete pipeline, encompassing data synthesis, cleaning, verification, and SFT training, requires **approximately 62 + 3 + 44 = 109 GPU hours.**
>
> **Notably, we also explore a more cost-effective alternative to address computational overhead.** As illustrated in Figure 4 (page 10, lines 486–499) and detailed in Section 3.5 (page 9, lines 453–473) of our paper, we apply SFT to only 20K samples starting from Qwen3-8B-Base, followed by 400 steps of large-scale reinforcement learning. This streamlined approach enables our AgentMath model to achieve the leading performance: 76.2% on AIME24 and 67.5% on AIME25, surpassing both OpenMath-Nemotron-7B (74.8% and 61.2%, respectively) and Qwen3-8B-Thinking (76.0% and 67.3%, respectively). This entire workflow, including data synthesis, cleaning, SFT, and RL training, requires only about 76 GPU hours total, providing a substantially more efficient solution.
>
> [1] https://huggingface.co/datasets/a-m-team/AM-DeepSeek-R1-0528-Distilled
>
> > **Questions-8**: How are textual reasoning steps segmented into computational components when deciding where to inject code? What criteria or metrics determine whether using code is more effective for a given part of the reasoning chain?
>
> Thank you for raising this important question. Given that Questions-8 and Weaknesses-4 address similar concerns, **I responded to them jointly by placing my answer to Questions-8 under the Weaknesses-4 section above.** For a comprehensive discussion, please **search for "Questions-8-Q1" and "Questions-8-Q2" tags** in the full text.
>
> > **Questions-9**: Since different models can prefer different code usage patterns for the same question, how robust is the segmentation procedure across models?
>
> Thank you for bringing up this valuable question. I observed that Questions-9 and Weaknesses-4-Q3 raise similar concerns, **so I have integrated my answer to Questions-9 into the Weaknesses-4-Q3 discussion for coherence.** Please **search for "Weaknesses-4-Q3" or "Questions-9" tag** in the full text to find the complete response.
>
> > **Questions-10**: The paper uses large teacher models (DeepSeek R1, DeepSeek V3, and Qwen-30B) for data synthesis. How do the authors ensure fairness when comparing with models trained without such teacher guidance? Would the performance gap shrink if competing models were also allowed to distill from comparable teacher models?
>
> I sincerely appreciate your important question. **The performance gap would shrink if competing models were also allowed to distill from comparable teacher models.**
>
> Considering that Questions-10 and Weaknesses-5 address similar concerns, so to streamline our discussion, **I have incorporated my answer to Questions-10 into the Weaknesses-5 section.** Please **search for "Questions-10-Q1" and "Questions-10-Q2" tags** in the full text  to access the detailed discussion.

---

> ### Author Response · Authors · 2025-11-24
> **Response to Reviewer 5L8W (16/16)**
>
> **Note:** It is worth noting that we have added a **Reproducibility Statement** to **our latest upload of revised paper (page 11, lines 540-555)**. To further facilitate community reproducibility and future research, we have **made available our anonymous source code, data synthesis pipeline, and training/eval scripts in the supplementary materials and [https://anonymous.4open.science/r/AgentMath-E5BR](https://anonymous.4open.science/r/AgentMath-E5BR).**
>
> Due to time constraints, parts of this codebase may currently rely on hard-coded elements (e.g., in the synthesis pipeline, RL training, and data paths), and some code components may be incomplete or missing. If you encounter any issues during execution, please share feedback actively. We sincerely apologize for any inconvenience and appreciate your understanding. We are committed to continuously improving this repository, enhancing code quality, and making the training process more robust, smooth, and stable.
>
> Regarding the synthesized data and model checkpoints, their release is subject to our company's open-source policy review process to ensure compliance. We are actively advancing the company review process. Once approval is granted, we will immediately release the synthesized data and models to further support research in the Agent LLM community.
>
> We hope these materials will help reviewers and researchers better understand and reproduce our work.
>
> We sincerely hope the above response adequately addresses your concerns. We truly value your input and we are looking forward to receiving any additional feedback you may have and are very happy to engage in any follow-up discussions or address any additional comments. Please know that we remain fully committed to continuing this dialogue and addressing any further questions or comments you might raise. We deeply apologize for any inconvenience caused by the additional time this process has taken, and we humbly ask for your patience and understanding. Your invaluable contribution to our work means a great deal to us, and we are truly thankful for your engagement.
>
> Respectfully,
>
> Paper 18148 Authors.

---

> > ### Author Response · Authors · 2025-11-27
> > **Follow-up on Our Response to Your Review (Paper 18148)**
> >
> > Dear Reviewer 5L8W,
> >
> > Thank you for your detailed and thoughtful review of our work. We genuinely appreciate the time and effort you have dedicated to evaluating our paper.
> >
> > With the discussion period now open, we are writing to follow up.  We would be grateful if you could let us know whether our response has adequately addressed your concerns or if you have any remaining questions. We welcome any additional feedback you may have.
> >
> > We would be happy to continue the discussion or address any further comments. Thank you once again for your valuable contributions to improving our work.
> >
> > Respectfully,
> >
> > Paper 18148 Authors

---

### Official Review · Reviewer_xGwU · 2025-10-31

**Soundness:** 3
**Presentation:** 2
**Contribution:** 2
**Rating:** 4
**Confidence:** 4

**Summary:**

The paper proposes AgentMath, a tool-augmented agentic framework for competition-level math reasoning. It tackles three pain points: (i) scarcity and noisiness of tool-use data; (ii) lack of agentic RL that interleaves generation with code execution; (iii) infrastructure bottlenecks for ultra-long contexts and many tool invocations. Concretely, it introduces an automated tool-augmented trajectory synthesis pipeline (code injection → executability verification → environment-feedback alignment → self-correction examples), an agentic RL scheme (GRPO-style optimization with interleaved code execution and selective masking of environment outputs; a reward combining correctness with tool-usage efficiency), and a scalable training system (request-level async scheduling, agentic partial rollout, prefix-aware weighted load balancing). On AIME24/25 and HMMT25, the approach reports strong results; e.g., AgentMath-30B-A3B = 90.6/86.4/73.8%, competitive with proprietary frontier models and surpassing open counterparts of comparable size. The system ablations claim 4–5× end-to-end RL throughput gains from the engineering stack

**Strengths:**

1. The refinement pipeline (format consistency, executability checks, feedback alignment, self-correction) shows sizable SFT gains and scaling trends before RL.
2. The async scheduler + partial rollout + prefix-aware balancing reportedly lift RL throughput 4–5×, and the paper provides a breakdown and sensitivity to segment count.
3. The 30B model achieves 90.6/86.4/73.8% on AIME24/25/HMMT25

**Weaknesses:**

1. Novelty over prior tool-augmented reasoning may feel incremental.
The community already has multiple tool-augmented RL frameworks (e.g., ReTool-style RL teaching strategic tool calls) and mature RL infra with asynchronous rollout and truncation/partial-trajectory techniques (e.g. AREAL, ROLL). Much of AgentMath’s lift appears to stem from more elaborate SFT data synthesis and a careful infra implementation rather than a fundamentally new RL principle. From a novelty lens, the async scheduler and partial rollout echo patterns well-established in modern RL systems; similarly, the use of truncated/segmented rollouts conceptually aligns with prior truncated policy optimization ideas (e.g. Truncated Proximal Policy Optimization). The work’s value is therefore primarily systems consolidation + scale rather than conceptual RL novelty.
2. I could not find a code repository or an explicit release statement in the current manuscript (PDF). Given that a large portion of the contribution is engineering (decoupled async architecture, agentic partial rollout, prefix-aware balancing, data-synthesis tooling), lack of code release significantly limits reproducibility and adoption. Please clarify whether code, data pipelines, and training/eval scripts will be released (e.g., upon acceptance) and provide a URL if available.

**Questions:**

See my comments on Weaknesses.

---

> ### Author Response · Authors · 2025-11-24
> **Response to Reviewer xGwU (1/6)**
>
> Dear Reviewer xGwU,
>
> We thank you for your valuable comments and the time you spent reviewing our work! Your professional feedback provides valuable guidance for writing a more comprehensive and competitive paper. Below, we provide detailed responses to the Weaknesses raised in your review of our paper, addressing each point systematically.
>
> > **Weaknesses-1**: Novelty over prior tool-augmented reasoning may feel incremental. The community already has multiple tool-augmented RL frameworks (e.g., ReTool-style RL teaching strategic tool calls) and mature RL infra with asynchronous rollout and truncation/partial-trajectory techniques (e.g. AREAL, ROLL). Much of AgentMath’s lift appears to stem from more elaborate SFT data synthesis and a careful infra implementation rather than a fundamentally new RL principle. From a novelty lens, the async scheduler and partial rollout echo patterns well-established in modern RL systems; similarly, the use of truncated/segmented rollouts conceptually aligns with prior truncated policy optimization ideas (e.g. Truncated Proximal Policy Optimization). The work’s value is therefore primarily systems consolidation + scale rather than conceptual RL novelty.
>
> We would like to express our sincere gratitude for your thoughtful and constructive feedback. It is our sincere pleasure to have the opportunity to further illustrate the novelty of our work and to carefully address the points you have raised. Our response is organized around the following **three main aspects**:
>
> **1. The Distinctions Between AgentMath and multiple tool-augmented RL frameworks (e.g., ReTool-style RL teaching strategic tool calls)**
>
> We commend Retool's important contributions to tool-augmented reinforcement learning. It employs 2k synthetic tool-augmented samples for cold-start initialization and leverages large-scale RL training to enhance reasoning capabilities. However, we note that Retool provides limited disclosure regarding specific implementation details of data synthesis and RL training procedures, leaving substantial room for further investigation. **AgentMath and Retool differ across the following five aspects:**
>
> **1.1 Data Synthesis Strategy**
>
> Through systematic analyzing Retool's open-sourced 2k synthetic dataset and responses generated by their released models (ReTool-Qwen-32B-SFT and ReTool-Qwen-32B), we observe phenomena of **tool overuse** and **redundant verification**: code is frequently invoked even for simple calculations, and verification via code is repeatedly performed even when the model has already computed the correct answer manually, leading to substantial generation of unnecessary tokens.
>
> In contrast, **AgentMath's synthesis strategy focuses on applying tools to complex and error-prone computational scenarios**, specifically: complex equation solving, high-dimensional linear algebra operations, advanced calculus derivations, large-number arithmetic, and intricate probability and combinatorics problems. Furthermore, we employ multi-dimensional quality refinement and systematically filter low-complexity code snippets.
>
> **1.2 Multi-dimensional Quality Refinement Pipeline**
>
> AgentMath synthesizes data through a segmented approach: we partition the chain-of-thought traces produced by DeepSeek-R1 into fixed-token (i.e., 3k) chunks and transforms each segment into a tool-augmented chain-of-thought. We apply an automated refinement pipeline that includes: (1) filtering samples with ≤5 lines of code, (2) correcting formatting errors, (3) excluding samples with code execution failures, (4) performing environmental feedback alignment, and (5) injecting self-correction capabilities. In Section 3.5 "Synthetic Data Refinement and Scaling Law" (Page 10, Lines 508–526), Table 3, and Appendix A.7.3 (Pages 32–34, Lines 1721–1790, Figure 7) of our paper provides the discussions of the six quality refinement modules employed in our data synthesis pipeline.

---

> ### Author Response · Authors · 2025-11-24
> **Response to Reviewer xGwU (2/6)**
>
> **[Continue the response to above Weaknesses-1]**
>
> As shown in **Table 1**, our multi-dimensional quality refinement process improves accuracy on AIME24 from 35.3% (using initial unrefined synthetic data) to 60.5%, and on AIME25 from 25.7% to 53.3%. These results demonstrate the critical role and effectiveness of each refinement module in optimizing data quality, thereby establishing a foundation for investigating scaling laws of synthetic data. Consequently, we **regard the multi-dimensional quality refinement pipeline as an essential component of AgentMath and one of our key contributions.** We prioritize transparency and clarity in our synthesis and refinement procedures; in contrast, **Retool provides limited disclosure of data synthesis and post-processing details, which poses challenges for reproducibility.**
>
> **Table 1: Ablation Study of Multi-dimensional Quality Refinement Pipeline**
>
> | Models / Refinement Steps | AIME24 | AIME25 |
> | :--- | :---: | :---: |
> | Initial Unrefined CI-Synthetic Data (20k) | 35.3% | 25.7% |
> | + Format consistency correction | 47.4% | 40.1% |
> | + Code executability verification | 52.8% | 44.8% |
> | + Environmental feedback alignment | 56.3% | 48.3% |
> | + Tool-usage rationality assessment | 57.2% | 48.9% |
> | + Self-correction capability injection | 58.6% | 50.8% |
> | + SFT with selective feedback masking | 60.5% | 53.3% |
>
> **1.3 Reinforcement Learning Training Strategy**
>
> AgentMath employs a request-based asynchronous rollout strategy combined with Agent Partial Rollout, implementing a three-stage progressive training pipeline for ultra-long sequences with large-scale tool invocations (sequence length: 48k→72k→96k; tool invocation count: 48→72→96), achieving stable performance improvements throughout the RL training process. In contrast, Retool has not disclosed detailed specifications of its RL training procedure.
>
> **1.4 Data Synthesis Performance Comparison**
>
> To ensure a fair comparison, we strictly follow Retool's experimental setup: the same 2k synthetic problems from Retool's official release, the same DeepSeek-R1 teacher model responses, the same base model (Qwen2.5-32B-Instruct), and the same number of RL training steps (400 steps). The sole difference lies in the data synthesis strategy: AgentMath employs our proposed tool-augmented synthesis method, while Retool uses its original synthesis prompts. The results are presented in **Table 2**:
>
> - AgentMath-32B-SFT-2k achieves 44.1% on AIME24 and 37.3% on AIME25, surpassing Retool-32B-SFT-2k (40.9%, 34.5%), demonstrating **the effectiveness of our tool-augmented data synthesis approach.**
> - AgentMath-32B-RL attains 74.8% on AIME24 and 56.6% on AIME25, likewise surpassing Retool-32B-RL (67.0%, 49.3%), further confirming **the effectiveness of our reinforcement learning training strategy.**
>
> **Overall, compared to Retool, AgentMath exhibits consistent performance advantages across the SFT and RL stages when using the same experimental setup, underscoring the efficiency and robustness of our proposed method.**
>
> **Table 2: Performance comparison between AgentMath and Retool on AIME24 and AIME25 during SFT and RL stages.**
>
> | Model | AIME24 Acc | AIME25 Acc |
> | :--- | :---: | :---: |
> | Retool-32B-SFT-2k | 40.9% | 34.5% |
> | AgentMath-32B-SFT-2k | 44.1% | 37.3% |
> | Retool-32B-RL | 67.0% | 49.3% |
> | AgentMath-32B-RL | 74.8% | 56.6% |
>
> **1.5 Scaling Laws of Synthetic Data**
>
> **We systematically investigate the scaling laws of tool-augmented synthetic data in our paper**, Section 3.5 (Page 10, Lines 508–526) and Appendix A.7.3 (Pages 32–34, Lines 1721–1790, Figure 7). Using the Qwen3-8B-Base model, we observe consistent performance improvements as the training data scales from 2k to 300k examples.
>
> Specifically, as shown in Table 3, AIME24 accuracy increases from 27.2% to 78.4%, AIME25 accuracy from 21.1% to 72.2%, and Google-IMO-AnswerBench[1] accuracy from 5.3% to 28.8% (+23.5 points), yielding an overall average performance gain of 41.9 points (from 17.9% to 59.8%). These results validate both the scalability and effectiveness of our synthesis approach. However, **recent works such as Retool focus exclusively on small-scale data (e.g., 2k samples) without delving into the scaling laws of synthetic data.**
>
> **Table 3: Scaling laws of tool-augmented synthetic data on AgentMath-8B-SFT models.**
>
> | SFT Training Data Volume | AIME24 | AIME25 | Google-IMO-AnswerBench | Avg Score |
> | :--- | :---: | :---: | :---: | :---: |
> | AgentMath-8B-SFT-2k | 27.2 | 21.1 | 5.3 | 17.9 |
> | AgentMath-8B-SFT-20k | 60.5 | 53.3 | 12.8 | 42.2 |
> | AgentMath-8B-SFT-40k | 66.8 | 57.9 | 17.6 | 47.4 |
> | AgentMath-8B-SFT-80k | 69.2 | 61.8 | 21.4 | 50.8 |
> | AgentMath-8B-SFT-160k | 73.8 | 66.4 | 25.6 | 55.3 |
> | AgentMath-8B-SFT-300k | 78.4 | 72.2 | 28.8 | 59.8 |
>
> [1] Luong M T, Hwang D, Nguyen H H, et al. Towards Robust Mathematical Reasoning[C]//Proceedings of the 2025 Conference on Empirical Methods in Natural Language Processing. 2025: 35406-35430.

---

> ### Author Response · Authors · 2025-11-24
> **Response to Reviewer xGwU (3/6)**
>
> **[Continue the response to above Weaknesses-1]**
>
> In summary, AgentMath distinguishes itself from Retool across five critical aspects: **data synthesis strategy, multi-dimensional quality refinement, reinforcement learning training strategy, data synthesis performance, and scaling laws of synthetic data**, thereby demonstrating the effectiveness and distinctive contributions our approach.
>
> **2. Regarding that mature RL infra with asynchronous rollout and truncation/partial-trajectory techniques (e.g. AREAL, ROLL) and the async scheduler and partial rollout echo patterns well-established in modern RL systems**
>
> We have conducted a thorough investigation of excellent open-source RL frameworks such as AREAL and ROLL, which have made significant contributions to advancing the development of LLMs in the RL domain.
>
> We note that the AREAL and ROLL frameworks supported the asynchronous rollout and asynchronous scheduling functionalities **on July 31, 2025.**
>
> Regarding truncation/partial-trajectory techniques, ROLL Flash and RollPacker introduced these approaches to mitigate long-tail rollout issues and accelerate asynchronous RL training. Specifically, ROLL Flash was **made available on arXiv on October 11, 2025**, while RollPacker was **released on arXiv on September 21, 2025.**
>
> It is worth noting that **the submission deadline for our paper was September 24, 2025 AOE**, resulting in **a time difference of less than two months from AREAL, ROLL, ROLL Flash, and RollPacker.**
>
> **According to ICLR submission guidelines, papers published within the most recent two months may be considered concurrent work.** Therefore, these excellent frameworks and works can **be regarded as concurrent work with AgentMath.**
>
> We fully recognize the value of these works and their substantial contributions to advancing LLMs in the RL domain. **We have cited and acknowledged them in the related work section of our latest upload of revised paper.**
>
> **3. Regarding the use of truncated/segmented rollouts conceptually aligns with prior truncated policy optimization ideas (e.g. Truncated Proximal Policy Optimization).**
>
> I agree the use of truncated or segmented rollouts in our method is conceptually aligned with prior work on truncated policy optimization, such as Truncated Proximal Policy Optimization (T-PPO). T-PPO (Truncated Proximal Policy Optimization) is a valuable contribution that effectively improves training efficiency by optimizing the policy update process and constraining generation length. This work was released on arXiv on June 15, 2025, however, **since implementation code and associated models are not yet publicly available, gaining an in-depth understanding of certain method details remains somewhat limited.**
>
> During our implementation of the Partial Rollout mechanism, we encountered several practical challenges. While Partial Rollout effectively alleviates long-tail latency issues, it simultaneously introduces requests with longer prefixes, resulting in increased KV cache memory consumption and higher prefill costs. So, to mitigate this issue, we **propose a prefix-aware weighted load balancing strategy** that dynamically assigns weights based on prefix length and routes requests to the least-loaded inference engine, **in contrast T-PPO does not mention this**.
>
> Moreover T-PPO's focus on Partial Rollout along the generation length dimension, we extend this concept to address challenges specific to agent reinforcement learning scenarios. In Agent RL training with large-scale tool calls (e.g., 64 or 96 invocations), we frequently encounter significant imbalance in the number of tool invocations across prompts within a batch. **To address this, we introduce Partial Rollout along the tool invocation dimension, capping the maximum number of tool calls per segment at 32 and deferring incomplete calls to subsequent batches.** This approach enables effective handling of Agent RL training tasks involving extremely long sequences with extensive tool invocations.
>
> **We have explicitly cited T-PPO in the related work section of our latest upload of revised paper** to commend its precedence and contributions. It is important to note that we do not claim to introduce conceptual RL novelty or a fundamentally new RL principle. Rather, our contribution lies in extending and adapting existing truncated rollout concept to address the unique challenges that arise in Agent RL training. Therefore, we refine our contribution regarding Partial Rollout as follows: **building upon T-PPO, we extend the concept of truncated/segmented rollouts to Agent RL and propose Tool Partial Rollout for ultra-long and large-scale tool invocations scenarios.**

---

> ### Author Response · Authors · 2025-11-24
> **Response to Reviewer xGwU (4/6)**
>
> **[Continue the response to above Weaknesses-1]**
>
> **4. In summary, we highlight the core contributions of AgentMath from four main aspects:**
>
> **4.1 Efficient Tool-Augmented Data Synthesis Method:**
>
> Compared to existing method (e.g., Retool), AgentMath achieves substantial improvements in data synthesis quality. We observe that prior methods (e.g., Retool) suffer from tool overuse, including invoking tools for simple calculations and performing redundant verification of already-correct results, which unnecessarily increases token consumption and inefficiency. To address this issue, **AgentMath focuses on applying tool calls specifically to complex computational scenarios that genuinely require them (e.g., solving intricate equations, performing complex large-number operations, and advanced linear algebra and calculus problems)**. We propose **a segmented synthesis strategy coupled with multi-dimensional quality refinement mechanisms (including code executability verification and environment feedback alignment)** to ensure transparency and clarity throughout the synthesis pipeline. Experimental results demonstrate that under identical training configurations, AgentMath significantly outperforms Retool on both AIME24 and AIME25 across both SFT and RL stages,  validating the effectiveness of our synthesis method and the superior quality of the synthesized data.
>
> **4.2 Scaling Laws for Tool-Augmented Data:**
>
> We **systematically validate the scaling law of tool-augmented synthetic data.** Through AgentMath's segmented data synthesis pipeline and multi-dimensional quality refinement, we scale the dataset from 2K to 300K samples. Building upon the Qwen3-8B-Base model, we observe substantial improvements: accuracy on AIME24 increases from 27.2% to 78.4%, and on AIME25 from 21.1% to 72.2%. These results demonstrate the effectiveness of the scaling law for tool-augmented data. By comparison, recent approaches such as Retool utilize limited data (e.g., 2K samples) and do not fully explore the benefits of data scaling.
>
> **4.3 Tool Partial Rollout and Prefix-Aware Load Balancing:**
>
> We extend the advanced concept of truncated/segmented rollouts introduced by T-PPO to the Agent RL training, and **propose Tool Partial Rollout, enabling partial rollout training for large-scale tool invocations.** Specifically, we limit each segment to a maximum of 32 tool calls, deferring incomplete invocations to subsequent batches.
>
> To address the substantial KV cache memory and prefill overhead incurred by ultra-long sequences, we **propose a Prefix-Aware Weighted Load Balancing strategy** that dynamically allocates weights according to prefix length and routes requests to the inference engine with the lowest load. Our overall framework achieves a 4–5× speedup in training, making efficient Agent RL training feasible for scenarios with ultra-long sequences and large-scale tool calls (e.g., 96k tokens with 96 tool calls).
>
> **4.4 Large-Scale Agent RL Training and State-of-the-Art Performance:**
>
> AgentMath successfully achieves large-scale RL training with **ultra-long sequences (e.g., 96k tokens) and extensive tool invocations (e.g., 96 tool calls)** across **diverse model backbones ranging from 1.5B to 32B parameters**, yielding consistent and substantial performance improvements. **AgentMath achieves state-of-the-art results on challenging mathematical competition benchmarks, including AIME24, AIME25, and HMMT25, significantly outperforming leading open-source models of comparable scale size. Specifically, AgentMath-30B-A3B achieves accuracies of 90.6%, 86.4%, and 73.8%, respectively, surpassing OpenAI-o3-mini and Claude-Opus-4.0-Thinking while remaining competitive with OpenAI-o3, Gemini-2.5-Pro, and DeepSeek-R1-671B-0528.** These results validate the effectiveness and scalability of our data synthesis methodology and RL training strategy, paving the way for building more sophisticated math reasoning agents. We commit to open-sourcing our code, data pipeline, and models to advance the LLM reasoning community.
>
> It is worth noting that we **commend many excellent works(i.e., ROLL[1], AREAL[2],  Truncated Proximal Policy Optimization[3], Roll Flash[4], RollPacker[5], Realhf[7])**. These research efforts are highly complementary to our work and collectively advance the field.  **In the related work section of our latest upload of revised paper, we have incorporated citations to these latest works below and extend our appreciation to their contributors**, including but not limited to the following works (If there are any relevant works that have been overlooked, we sincerely apologize and would greatly appreciate your suggestions.):
>
> We sincerely hope our responses have addressed your concerns. We are deeply grateful for your patience and thoughtful engagement with our work. We warmly welcome any further feedback or discussion regarding the implementation details of our work, and apologize for any confusion or inconvenience caused.

---

> ### Author Response · Authors · 2025-11-24
> **Response to Reviewer xGwU (5/6)**
>
> **[Continue the response to above Weaknesses-1]**
>
> [1] Wang W, Xiong S, Chen G, et al. Reinforcement Learning Optimization for Large-Scale Learning: An Efficient and User-Friendly Scaling Library[J]. arXiv preprint arXiv:2506.06122, 2025.
>
> [2] Fu W, Gao J, Shen X, et al. AReaL: A Large-Scale Asynchronous Reinforcement Learning System for Language Reasoning[J]. arXiv preprint arXiv:2505.24298, 2025.
>
> [3] Fan T, Liu L, Yue Y, et al. Truncated Proximal Policy Optimization[J]. arXiv preprint arXiv:2506.15050, 2025.
>
> [4] Lu H, Liu Z, Xiong S, et al. Part II: ROLL Flash--Accelerating RLVR and Agentic Training with Asynchrony[J]. arXiv preprint arXiv:2510.11345, 2025.
>
> [5] Gao W, Zhao Y, An D, et al. RollPacker: Mitigating Long-Tail Rollouts for Fast, Synchronous RL Post-Training[J]. arXiv preprint arXiv:2509.21009, 2025.
>
> [6] Liu Z, Liu J, He Y, et al. Part i: Tricks or traps? a deep dive into rl for llm reasoning[J]. arXiv preprint arXiv:2508.08221, 2025.
>
> [7] Mei Z, Fu W, Li K, et al. Realhf: Optimized rlhf training for large language models through parameter reallocation[J]. arXiv e-prints, 2024: arXiv: 2406.14088.
>
> [8] Mei Z, Fu W, Li K, et al. Real: Efficient rlhf training of large language models with parameter reallocation[J]. arXiv preprint arXiv:2406.14088, 2024.
>
> [9] Hu J, Wu X, Zhu Z, et al. Openrlhf: An easy-to-use, scalable and high-performance rlhf framework[J]. arXiv preprint arXiv:2405.11143, 2024.
>
> [10] Shen G, Wang Z, Delalleau O, et al. Nemo-aligner: Scalable toolkit for efficient model alignment[J]. arXiv preprint arXiv:2405.01481, 2024.
>
> [11] Wang Z, Wang K, Wang Q, et al. Ragen: Understanding self-evolution in llm agents via multi-turn reinforcement learning[J]. arXiv preprint arXiv:2504.20073, 2025.
>
> [12] Zilin Zhu, Chengxing Xie, Xin Lv, and slime Contributors. slime: An llm post-training framework for rl scaling. https://github.com/THUDM/slime, 2025. GitHub repository. Corre-sponding author: Xin Lv.
>
> [13] Gao J, Fu W, Xie M, et al. Beyond ten turns: Unlocking long-horizon agentic search with large-scale asynchronous rl[J]. arXiv preprint arXiv:2508.07976, 2025.
>
> [14] Liu J, Obando-Ceron J, Lu H, et al. Asymmetric Proximal Policy Optimization: mini-critics boost LLM reasoning[J]. arXiv preprint arXiv:2510.01656, 2025.
>
> [15] Li Y, Dong Z, Sun Y, et al. Attention Illuminates LLM Reasoning: The Preplan-and-Anchor Rhythm Enables Fine-Grained Policy Optimization[J]. arXiv preprint arXiv:2510.13554, 2025.
>
> > **Weaknesses-2**: I could not find a code repository or an explicit release statement in the current manuscript (PDF). Given that a large portion of the contribution is engineering (decoupled async architecture, agentic partial rollout, prefix-aware balancing, data-synthesis tooling), lack of code release significantly limits reproducibility and adoption. Please clarify whether code, data pipelines, and training/eval scripts will be released (e.g., upon acceptance) and provide a URL if available.
>
> We sincerely thank you for your valuable suggestions. We place great importance on the reproducibility of our research and deeply apologize for any inconvenience regarding code availability.
>
> To address your concerns and ensure community reproducibility, we **have provided additional details on reproducibility below, along with our anonymous code in the supplementary materials and [https://anonymous.4open.science/r/AgentMath-E5BR](https://anonymous.4open.science/r/AgentMath-E5BR)**
>
> **1.  Reproducibility Statement**
>
> To ensure the reproducibility of AgentMath, we provide comprehensive details on algorithms, data synthesis procedures, Agent RL training, and experimental configurations throughout the main paper and appendices.
>
> - Data Synthesis and Quality Refinement: The data synthesis pipeline, multi-dimensional quality refinement mechanisms, and the design of synthesis and judge prompts are detailed in Sections 2.1 and 2.2, and Appendices A.2 and A.6.
> - Agent RL Training: The RL training framework and implementation details are presented in Section 2.3 and Appendix A.3.
> - RL Training Efficiency and Algorithmic Details: Optimization strategies for RL training efficiency and the design of Algorithm 1 are described in Section 2.4 and Appendix A.4.
> - Module Contributions and Ablation Studies: We systematically evaluate the individual contributions of tool-augmented data synthesis, multi-stage RL training, scaling laws of synthetic data, and multi-dimensional quality refinement through ablation experiments in Section 3.5 and Appendix A.7.
> - Case Studies: Case analyses are provided in Appendix A.8.
> - Training and Evaluation Configurations: Detailed experimental settings, including SFT and RL training data, training strategies, and evaluation settings, are documented in Sections 3.1 , 3.2, 3.3, 3.4 and Appendix A.5.
>
> **It is worth noting that we have added the above Reproducibility Statement to our latest upload of revised paper (page 11, lines 540-555).**

---

> ### Author Response · Authors · 2025-11-24
> **Response to Reviewer xGwU (6/6)**
>
> **[Continue the response to above Weaknesses-2]**
>
> To further facilitate community reproducibility and future research, we **have made available our anonymous source code, data synthesis pipeline, and training/eval scripts in the supplementary materials and [https://anonymous.4open.science/r/AgentMath-E5BR](https://anonymous.4open.science/r/AgentMath-E5BR)**.
>
> We hope these materials will help reviewers and researchers better understand and reproduce our work.
>
> **2. Codebase Overview**
>
> To facilitate a clearer understanding of our code repository, we offer a concise overview of the core functionalities implemented in AgentMath below:
>
> - Data Synthesis Pipeline: Tool-augmented data synthesis code located in **agentmath/data_synthesis**
> - RL Training Modules: Efficiency optimization components including Request-Level Asynchronous Rollout Scheduling, Agentic Partial Rollout, and Prefix-Aware Weighted Load Balancing in **agentmath/agent_dapo_ray_trainer_patial_rollout.py and agentmath/agent_loop**
> - Running Scripts: Training scripts in **agentmath/train_scripts**, evaluation scripts in **agentmath/eval_scripts**
>
> **Note**: Due to time constraints, parts of this codebase may currently rely on hard-coded elements (e.g., in the synthesis pipeline, RL training, and data paths), and some code components may be incomplete or missing. If you encounter any issues during execution, please share feedback actively. We sincerely apologize for any inconvenience and appreciate your understanding. We are committed to continuously improving this repository, enhancing code quality, and making the training process more robust, smooth, and stable.
>
> **Regarding the synthesized data and model checkpoints, their release is subject to our company's open-source policy review process to ensure compliance. We are actively advancing the company review process.** Once approval is granted, we will immediately release the synthesized data and models to further support research in the Agent LLM community.
>
> For convenience, we have uploaded the anonymous code, data pipelines, and training/eval scripts to **the supplementary materials for now and [https://anonymous.4open.science/r/AgentMath-E5BR](https://anonymous.4open.science/r/AgentMath-E5BR)**.
>
> Please feel free to review them there. We fully intend to share a clear URL on our GitHub repository in the future.
>
> We hope that the responses above can address your concerns. We sincerely appreciate your patience and interest and apologize for any inconvenience this may cause. We are looking forward to receiving any additional feedback you may have and are very happy to engage in any follow-up discussions or address any additional comments. Thank you once again for your valuable contributions to our work.
>
> Respectfully,
>
> Paper 18148 Authors.

---

> > ### Author Response · Authors · 2025-11-27
> > **Follow-up on Our Response to Your Review (Paper 18148)**
> >
> > Dear Reviewer xGwU,
> >
> > Thank you for your detailed and thoughtful review of our work. We genuinely appreciate the time and effort you have dedicated to evaluating our paper.
> >
> > With the discussion period now open, we are writing to follow up.  We would be grateful if you could let us know whether our response has adequately addressed your concerns or if you have any remaining questions. We welcome any additional feedback you may have.
> >
> > We would be happy to continue the discussion or address any further comments. Thank you once again for your valuable contributions to improving our work.
> >
> > Respectfully,
> >
> > Paper 18148 Authors

---

### Official Review · Reviewer_iSnP · 2025-11-03

**Soundness:** 3
**Presentation:** 3
**Contribution:** 2
**Rating:** 6
**Confidence:** 3

**Summary:**

This work introduces AgentMath, a framework that enhances the mathematical reasoning ability of large language models by combining their natural‐language chain‐of‐thought with precise external tool use (e.g., code execution). The authors propose (1) an automated method to convert natural language reasoning traces into structured tool-augmented interaction trajectories (for supervised fine-tuning), (2) a multi-turn agentic reinforcement‐learning paradigm that interleaves language reasoning and real‐time code/tool invocation to learn optimal strategies for tool use and error correction, and (3) a highly efficient training infrastructure (with asynchronous rollout scheduling, partial rollouts, and prefix‐aware load balancing) enabling meaningful scaling on ultra-long sequences with many tool calls. The result is a model better able to tackle complex mathematical problems beyond pure language reasoning—off-loading heavy computation to tools while retaining robust step‐by‐step reasoning in language.

**Strengths:**

- This paper did a good work in presentation. All the components in the pipeline are illustrated in good details and intuitions. This provides a good recipe for open-source math prover training.

- A very comprehensive comparison between the model trained from this work and other open/closed-source models are provided, making the results very convincing.

**Weaknesses:**

- The paper's contribution is mostly on the development of the entire pipeline from my perspective. As many technical innovations claimed in the paper are either standard or mostly from engineering aspects, as agentic RL training is not a fresh concept nowadays (including coding-assisted math reasoning). I value a lot of the paper's efforts on developing such a comprehensive pipeline (SFT data collection and large-scale RL system building), which I understand is very challenging and helpful for the open-source community; however, the lack of technical contribution is the major factor preventing me providing a higher score.

**Questions:**

My major concern, as mentioned in weakness, is on the technical innovation of this paper, which I would love to hear more from the authors.

---

> ### Author Response · Authors · 2025-11-25
> **Response to Reviewer iSnP (1/6)**
>
> Dear Reviewer iSnP,
>
> We sincerely thank you for your insightful comments and the time you dedicated to reviewing our work. Your expert feedback has been invaluable in guiding us towards refining our paper and making it more comprehensive and competitive. We greatly appreciate your support and constructive suggestions. In the following, we offer detailed responses to the Weaknesses and Questions raised in your review, addressing each point in a systematic manner.
>
> > **Weaknesses-1**: The paper's contribution is mostly on the development of the entire pipeline from my perspective. As many technical innovations claimed in the paper are either standard or mostly from engineering aspects, as agentic RL training is not a fresh concept nowadays (including coding-assisted math reasoning). I value a lot of the paper's efforts on developing such a comprehensive pipeline (SFT data collection and large-scale RL system building), which I understand is very challenging and helpful for the open-source community; however, the lack of technical contribution is the major factor preventing me providing a higher score.
>
> > **Questions-1**: My major concern, as mentioned in weakness, is on the technical innovation of this paper, which I would love to hear more from the authors.
>
> We sincerely thank you very much for your recognition of our work. We are delighted that you find this paper makes comprehensive and challenging efforts in pipeline development (including SFT data collection and large-scale RL system building), and sincerely appreciate your comments that "This paper did a good work in presentation. All the components in the pipeline are illustrated in good details and intuitions. This provides a good recipe for open-source math prover training" and "A very comprehensive comparison between the model trained from this work and other open/closed-source models are provided, making the results very convincing." Your positive feedback is tremendously encouraging to us.
>
> We agree with your observation that agentic RL training is not a fresh concept nowadays (including coding-assisted math reasoning), as prior works such as Retool and ToRL tried to explore this direction. However, we believe the field currently still faces two critical challenges that remain inadequately addressed:
>
> **1. Lack of efficient tool-augmented data synthesis methods and scaling law studies for tool-augmented long-chain reasoning scenarios**
>
> Existing approaches face substantial limitations when synthesizing tool-augmented data in long-chain reasoning scenarios. For instance, CoRT relies on manual annotation, making it costly and difficult to scale. While Retool employs LLM-based synthesis, it produces only 2k samples and provides insufficient details about the synthesis process. Moreover, existing research lacks systematic investigation of scaling laws for agent synthesized data. In code-assisted mathematical reasoning, what is the relationship between data scale and performance? Can superior performance be achieved through data scaling law? These fundamental questions have rarely been systematically studied in the open-source community.
>
> **2. Agent RL training encounters severe efficiency bottlenecks in ultra-long sequence and large-scale tool invocation scenarios**
>
> When models perform well on code-assisted mathematical reasoning through SFT training, can performance be continuously improved through large-scale Agent RL to achieve adaptive tool invocation? However, under conditions involving ultra-long reasoning chains and large-scale tool invocations (e.g., 96k token length, 96 tool calls), training efficiency becomes a critical bottleneck. Frequent environment interactions and ultra-long trajectory generation cause traditional batch synchronous rollout training on Qwen3-8B to require 3600-4000 seconds per training step, with rollout generation accounting for 70%-80% of total training time. This inefficiency severely constrains researchers' exploration of Agent RL training in long-horizon and large-scale tool-calling scenarios.
>
> We note that **training efficiency is crucial for RL training.** The ability to train for thousands of steps and break through performance plateaus directly impacts final results. For instance, DeepSeek-R1-Zero requires 4000-8000 training steps to observe the model's "Aha moment" phenomenon and continuous performance improvements. To address this core bottleneck, our work introduces innovative optimization that **reduce per-step training time from 3600-4000 seconds to 700-900 seconds, achieving a 4–5× speedup.** This provides critical support for advancing large-scale Agent RL in LLMs.

---

> ### Author Response · Authors · 2025-11-25
> **Response to Reviewer iSnP (2/6)**
>
> **[Continue the response to above Weaknesses-1 and Questions-1]**
>
> Furthermore, while leading proprietary models (GPT-5, Gemini 3.0, Claude 4.5, etc.) demonstrate exceptional performance on agent tasks such as code-assisted mathematical reasoning, the lack of technical details and training methods hinders the open-source community's ability to reproduce and advance this research. **By providing a clear, transparent, and complete pipeline for data synthesis and large-scale agent RL training that achieve competitive performance with proprietary models (e.g., Gemini 2.5-Pro, OpenAI-o3)**, our work aims to offer the open-source community a reproducible technical roadmap, thereby advancing the development of the Agent LLM field.
>
> Therefore, we highlight our core technical contributions across **four key dimensions**: (1) an efficient tool-augmented data synthesis method, (2) exploration of scaling laws for tool-augmented synthetic data, (3) efficient agent RL training techniques for ultra-long sequences and large-scale tool invocation scenarios, and (4) state-of-the-art performance.
>
> **1. Efficient Tool-Augmented Data Synthesis Method**
>
> **(1.1)** Compared to existing work (e.g., Retool), AgentMath achieves significant improvements in data synthesis. We observe that previous methods (e.g., Retool) **suffer from tool overuse issues**, including invoking tools for simple calculations and performing redundant verification of already-verified correct results, which leads to an increase in ineffective tokens. To address this issue, AgentMath focuses on **applying tool invocations to genuinely necessary complex computational scenarios** (e.g., solving complex equations, performing complex large-number arithmetic, and handling advanced linear algebra and calculus operations). We propose **a segmented synthesis strategy** that partitions DeepSeek-R1's long chain-of-thought process into fixed-length segments (e.g., 3K tokens) and transforms them into tool-augmented chains-of-thought, and incorporates automated **multi-dimensional quality refinement** to ensure high data quality.
>
> Under identical training settings, we use the same 2k synthetic data problems from Retool's official open-source release, the same DeepSeek-R1 teacher model responses, the same base model (Qwen2.5-32B-Instruct), and the same number of RL training steps (400 steps). The only difference lies in the data synthesis strategy: AgentMath employs our paper's proposed tool-augmented synthesis method (Appendix A.6.2, page 29, lines 1512–1564), while Retool uses its original synthesis prompts.
>
> As shown in the **table 1** below, AgentMath-32B-SFT-2k achieves 44.1% on AIME24 and 37.3% on AIME25, outperforming Retool-32B-SFT-2k (40.9%, 34.5%); meanwhile, AgentMath-32B-RL achieves 74.8% on AIME24 and 56.6% on AIME25, similarly outperforming Retool-32B-RL (67.0%, 49.3%). These experimental results demonstrate that AgentMath significantly outperforms Retool in both SFT and RL performance on AIME24 and AIME25, validating the effectiveness of our data synthesis method and the superior quality of the synthesized data.
>
> **Table 1: Performance comparison between AgentMath and Retool on AIME24 and AIME25 under identical training configurations, demonstrating the superiority of AgentMath's tool-augmented data synthesis method over Retool's approach in both SFT and RL stages.**
>
> | Model | AIME24 Acc | AIME25 Acc |
> | :--- | :---: | :---: |
> | Retool-32B-SFT-2k | 40.9% | 34.5% |
> | AgentMath-32B-SFT-2k | 44.1% | 37.3% |
> | Retool-32B-RL | 67.0% | 49.3% |
> | AgentMath-32B-RL | 74.8% | 56.6% |

---

> ### Author Response · Authors · 2025-11-25
> **Response to Reviewer iSnP (3/6)**
>
> **[Continue the response to above Weaknesses-1 and Questions-1]**
>
> **(1.2)** Furthermore, we emphasize the **multi-dimensional quality refinement** pipeline in our data synthesis process, which includes correcting format inconsistencies, excluding samples with failed code execution, performing Environmental Feedback Alignment, filtering low-complexity code (e.g., ≤5 lines of code), and injecting self-correction capabilities. In Section 3.5 "Synthetic Data Refinement and Scaling Law" (Page 10, Lines 508–526), Table 3, and Appendix A.7.3 (Pages 32–34, Lines 1721–1790) of our paper, we provide the discussion of six quality refinement modules employed in the data synthesis process.
>
> As shown in the **table 2** below, **through our multi-dimensional quality refinement pipeline, accuracy on AIME24 improves from 35.3% with initially unrefined synthetic data to 60.5%, and on AIME25 from 25.7% to 53.3%**, substantially demonstrating the critical role and effectiveness of each refinement module in optimizing data quality. We therefore consider the multi-dimensional quality refinement pipeline in data synthesis to be of paramount importance, which constitutes **one of our key contributions**. We are committed to maintaining transparency and clarity in our synthesis and refinement procedures, whereas some recent works provide limited details on this crucial aspect, thereby posing challenges for research reproducibility.
>
> **Table 2: Progressive improvement in model accuracy on AIME24 and AIME25 through multi-dimensional quality refinement pipeline. Each refinement step is applied cumulatively, demonstrating substantial accuracy gains from 35.3% to 60.5% on AIME24 and from 25.7% to 53.3% on AIME25.**
>
> | Models / Refinement Steps | AIME24 | AIME25 |
> | :--- | :---: | :---: |
> | Initial Unrefined CI-Synthetic Data (20k) | 35.3% | 25.7% |
> | + Format consistency correction | 47.4% | 40.1% |
> | + Code executability verification | 52.8% | 44.8% |
> | + Environmental feedback alignment | 56.3% | 48.3% |
> | + Tool-usage rationality assessment | 57.2% | 48.9% |
> | + Self-correction capability injection | 58.6% | 50.8% |
> | + SFT with selective feedback masking | 60.5% | 53.3% |
>
> **2. Scaling Laws for Tool-Augmented Synthetic Data**
>
> As shown in the **table 3** below, we systematically validate the scaling laws associated with tool-augmented synthetic data. Through AgentMath's segmented data synthesis strategy and multi-dimensional quality refinement, we progressively scale the dataset from 2k to 300k samples. Fine-tuning the Qwen3-8B-Base model, we observe **consistent performance improvements: accuracy on AIME24 increases from 27.2% to 78.4%, and on AIME25 from 21.1% to 72.2%.** Performance continues to improve with data scale, ultimately achieving excellent results. These findings robustly demonstrate that scaling laws remain effective for tool-augmented synthetic data. In contrast, some recent studies (e.g., Retool) utilize only limited data (e.g., 2k samples) and do not sufficiently explore the potential benefits of data scaling.
>
> **Table 3: Performance data scaling law of AgentMath-8B models across different benchmarks as a function of SFT training data volume, demonstrating consistent improvements from 2k to 300k samples on AIME24, AIME25, and Google-IMO-AnswerBench[1].**
>
> | SFT Training Data Volume | AIME24 | AIME25 | Google-IMO-AnswerBench | Avg Score |
> | :--- | :---: | :---: | :---: | :---: |
> | AgentMath-8B-SFT-2k | 27.2 | 21.1 | 5.3 | 17.9 |
> | AgentMath-8B-SFT-20k | 60.5 | 53.3 | 12.8 | 42.2 |
> | AgentMath-8B-SFT-40k | 66.8 | 57.9 | 17.6 | 47.4 |
> | AgentMath-8B-SFT-80k | 69.2 | 61.8 | 21.4 | 50.8 |
> | AgentMath-8B-SFT-160k | 73.8 | 66.4 | 25.6 | 55.3 |
> | AgentMath-8B-SFT-300k | 78.4 | 72.2 | 28.8 | 59.8 |
>
> [1] Luong M T, Hwang D, Nguyen H H, et al. Towards Robust Mathematical Reasoning[C]//Proceedings of the 2025 Conference on Empirical Methods in Natural Language Processing. 2025: 35406-35430.

---

> ### Author Response · Authors · 2025-11-25
> **Response to Reviewer iSnP (4/6)**
>
> **[Continue the response to above Weaknesses-1 and Questions-1]**
>
> **3. Efficient Agent RL Training for Ultra-Long Sequences and Large-Scale Tool Calling Scenarios**
>
> In Agent RL, the combination of long-context generation and frequent external tool interactions creates heterogeneous computational workloads, posing significant challenges to training efficiency. To address this, we have designed targeted optimization strategies comprising three core improvements:
>
> - **Request-Level Asynchronous Rollout Scheduling:** We replace the conventional static batch synchronous processing with **a coroutine-driven, request-level asynchronous scheduler.** Each trajectory rollout is treated as an independent long-running request, where the inference engine (server) and the agent (client) are fully decoupled through asynchronous communication. This effectively mitigates the efficiency bottleneck caused by synchronous waiting.
>
> - **Agentic Partial Rollout:** To alleviate the long-tail latency issues arising from ultra-long sequences and large-scale tool calls, we design an agentic partial rollout mechanism, including **Length Partial Rollout and Tool Partial Rollout**. This mechanism decomposes each trajectory $\tau$ into budget-constrained segments, where each segment is limited to a fixed maximum sequence length (e.g., 32k tokens) and a fixed maximum number of tool calls (e.g., 32). Incomplete sequences or tool calls are carried over to subsequent batches for continued execution, while completed trajectories immediately participate in training, thereby significantly improving training efficiency.
>
> - **Prefix-Aware Weighted Load Balancing:** Since Partial Rollout introduces ultra-long sequences, leading to a substantial increase in KV cache memory consumption and prefill computational overhead, we design a **prefix-aware weighted load balancing** strategy. This strategy dynamically assigns weights based on the prefix length of requests and intelligently routes them to the least-loaded inference engine instances, effectively alleviating memory and computational pressure.
>
> We systematically evaluate the efficiency improvements of the AgentMath training framework.
>
> As shown in **Table 4**, the traditional static batch synchronous rollout method **requires 3600–4000 seconds** per training step. With the introduction of request-level asynchronous rollout scheduling, latency is reduced to 2100–2500 seconds (a 1.5–1.8× speedup), effectively mitigating head-of-line blocking caused by tool calls. Further incorporating the Agentic Partial Rollout mechanism reduces latency to 1100–1300 seconds (a 3.0–3.3× speedup). Finally, with the incorporation of prefix-aware weighted load balancing, the per-step latency is **only 750–900 seconds**, achieving an **overall training speedup of 4.0–5.0×**.
>
> This framework enables efficient Agent RL training in extreme scenarios (e.g., 96k sequence length, 96 tool calls), for which training with such ultra-long sequences and large-scale tool calls has **few precedents in the open-source community**. We believe this work provides an important technical reference for training complex agent systems.
>
> **Table 4: Progressive training efficiency improvements in the AgentMath framework through cumulative optimization strategies, demonstrating up to 4.0–5.0× speedup over baseline static batch synchronous rollout.**
>
> | **Method** | **Time per step (s)** | **Speedup** |
> | :--- | :---: | :--- |
> | Static Batch Synchronous Rollout | 3600–4000 | — |
> | + Request-Level Asynchronous Rollout | 2100–2500 | 1.5–1.8× |
> | + Agentic Partial Rollout | 1100–1300 | 3.0–3.3× |
> | + Prefix-Aware Weighted Load Balancing | 750–900 | 4.0–5.0× |

---

> ### Author Response · Authors · 2025-11-25
> **Response to Reviewer iSnP (5/6)**
>
> **[Continue the response to above Weaknesses-1 and Questions-1]**
>
> **4. Large-Scale Agent RL Training Achieving State-of-the-Art Performance**
>
> AgentMath undergoes a three-stage curriculum learning process, with **sequence lengths progressively increasing from 48k to 72k to 96k tokens and tool invocations scaling from 48 to 72 to 96**. Across **diverse model backbones spanning 1.5B to 32B parameters**, we successfully conduct stable Agent RL training on ultra-long sequences with large-scale tool invocations, achieving consistent performance improvements.
>
> As detailed in **Table 5**, we achieve state-of-the-art performance on challenging math competition benchmarks including AIME24, AIME25, and HMMT25, significantly outperforming leading open-source models of comparable size. Specifically, **AgentMath-Qwen3-30B-A3B achieves accuracies of 90.6%, 86.4%, and 73.8% on AIME24, AIME25, and HMMT25 respectively, surpassing OpenAI-o3-mini and Claude-Opus-4.0-Thinking while remaining competitive with OpenAI-o3, Gemini-2.5-Pro, and DeepSeek-R1-671B-0528.** These results validate the effectiveness and scalability of our data synthesis method and Agent RL training strategy, paving the way for building more sophisticated and scalable math reasoning agents.
>
> **Table 5: Performance comparison of AgentMath against proprietary and frontier open-source models on AIME24, AIME25, and HMMT25.**
>
> | Models | AIME24 | AIME25 | HMMT25 |
> | :--- | :---: | :---: | :---: |
> | **Proprietary models** | | | |
> | OpenAI-o4-mini-w/tools | 98.7 | 99.5 | - |
> | Gemini-2.5-Pro | 92.0 | 88.0 | 82.5 |
> | OpenAI-o3 | 91.6 | 88.9 | 77.5 |
> | OpenAI-o3-mini | 87.3 | 86.3 | 53.0 |
> | Claude-Opus-4.0-Thinking | 83.0 | 72.0 | 58.3 |
> | **Frontier Models (1B ~ 2B)** | | | |
> | DeepSeek-R1-Distill-Qwen-1.5B | 28.8 | 21.8 | 15.3 |
> | Qwen3-1.7B Thinking | 52.0 | 35.3 | 23.3 |
> | OpenReasoning-Nemotron-1.5B | 55.5 | 45.6 | 31.5 |
> | **AgentMath-Qwen3-1.7B** | **59.6** | **48.1** | **40.2** |
> | **Frontier Models (7B ~ 8B)** | | | |
> | DeepSeek-R1-Distill-Qwen-7B | 55.0 | 39.7 | - |
> | Qwen3-8B Thinking | 76.0 | 67.3 | 44.7 |
> | OpenReasoning-Nemotron-7B | 84.7 | 78.2 | 63.5 |
> | DeepSeek-R1-0528-Qwen3-8B | 86.0 | 76.3 | 61.5 |
> | **AgentMath-Qwen3-8B** | **89.8** | **84.7** | **71.3** |
> | **Frontier Models (30B ~ 32B)** | | | |
> | ReTool-32B | 67.0 | 49.3 | - |
> | DeepSeek-R1-Distill-Qwen-32B | 72.9 | 59.0 | 33.0 |
> | Qwen3-30B-A3B-Thinking-2507 | 87.7 | 85.0 | 71.4 |
> | OpenReasoning-Nemotron-32B | 89.2 | 84.0 | 73.8 |
> | **AgentMath-Qwen3-30B-A3B** | **90.6** | **86.4** | **73.8** |
> | **Frontier Models (>32B)** | | | |
> | DeepSeek-R1-671B-0528 | 91.4 | 87.5 | 77.0 |
> | Qwen3-235B-A22B-Thinking-2507 | 94.2 | 92.3 | 83.9 |
> | **AgentMath-Qwen3-235B-A22B-SFT** | **93.4** | **90.8** | **81.7** |
>
> **In summary, we highlight our core contributions as follows:**
>
> 1. We propose an efficient tool-augmented data synthesis method that focuses on **applying tool invocation to genuinely demanding complex computational scenarios (e.g., solving complex equations, large-number arithmetic, advanced linear algebra, and calculus)**. By combining a **segmented synthesis strategy with a multi-dimensional quality refinement mechanism**, AgentMath significantly outperforms Retool in both SFT and RL stages on AIME24 and AIME25 under identical training configurations.
>
> 2. We **systematically validate the scaling laws associated with tool-augmented synthetic data**, scaling the dataset from 2K to 300K samples. Building on the Qwen3-8B-Base model, accuracy on AIME24 improves from 27.2% to 78.4%, and on AIME25 from 21.1% to 72.2%, demonstrating consistent performance gains with increased data scale.
>
> 3. To alleviate the training efficiency bottleneck in Agent RL under ultra-long sequences and large-scale tool invocation scenarios, we **introduce request-level asynchronous rollout scheduling, propose Agent Partial Rollout, and design prefix-aware weighted load balancing**. Through these technical innovations, our overall framework **achieves a 4-5× speedup in training efficiency**, reducing the time per training step **from 3600-4000 seconds to 750-900 seconds**, thereby enabling efficient agent RL training in extreme scenarios (e.g., **96K sequence length with 96 tool invocations**). To the best of our knowledge, such exploration of large-scale Agent RL training with extremely long sequences and massive tool invocations **is rarely conducted in the open-source community**.
>
> 4. Through three-stage curriculum Agent RL training, AgentMath **achieves state-of-the-art** performance on challenging math competition benchmarks including AIME24, AIME25, and HMMT25 across **various model backbones ranging from 1.5B to 32B parameters**, significantly outperforming leading open-source models of comparable size. Notably, **AgentMath-Qwen3-30B-A3B surpasses OpenAI-o3-mini while remaining competitive with Gemini-2.5-Pro and DeepSeek-R1-671B-0528**.

---

> ### Author Response · Authors · 2025-11-25
> **Response to Reviewer iSnP (6/6)**
>
> **[Continue the response to above Weaknesses-1 and Questions-1]**
>
> We are committed to open-sourcing our code, data synthesis pipeline, and model training workflow to advance the LLM reasoning community.
>
> **We have added the above discussions about the technical contribution weakness to Appendix A.10 of our latest upload of revised paper (pages 37–41, lines 1944–2200).**
>
>
> **Note:** It is worth noting that we have added the **Reproducibility Statement** to our latest upload of revised paper (page 11, lines 540-555). To further facilitate community reproducibility and future research, we have **made available our anonymous source code, data synthesis pipeline, and training/eval scripts in the supplementary materials and [https://anonymous.4open.science/r/AgentMath-E5BR](https://anonymous.4open.science/r/AgentMath-E5BR).**
>
> Due to time constraints, parts of this codebase may currently rely on hard-coded elements (e.g., in the synthesis pipeline, RL training, and data paths), and some code components may be incomplete or missing. If you encounter any issues during execution, please share feedback actively. We sincerely apologize for any inconvenience and appreciate your understanding. We are committed to continuously improving this repository, enhancing code quality, and making the training process more robust, smooth, and stable.
>
> **Regarding the synthesized data and model checkpoints, their release is subject to our company's open-source policy review process to ensure compliance. We are actively advancing the company review process.** Once approval is granted, we will immediately release the synthesized data and models to further support research in the Agent LLM community.
>
> We hope these materials will help reviewers and researchers better understand and reproduce our work.
>
> We hope that the responses above can address your concerns. We eagerly await any further feedback you may have and would be more than happy to engage in additional discussions or respond to any further comments. Thank you once again for your invaluable contributions to our work and for your careful and thorough review of our paper.
>
> Respectfully,
>
> Paper 18148 Authors.

---

> > ### Author Response · Authors · 2025-11-27
> > **Follow-up on Our Response to Your Review (Paper 18148)**
> >
> > Dear Reviewer iSnP,
> >
> > Thank you for your detailed and thoughtful review of our work. We genuinely appreciate the time and effort you have dedicated to evaluating our paper.
> >
> > With the discussion period now open, we are writing to follow up.  We would be grateful if you could let us know whether our response has adequately addressed your concerns or if you have any remaining questions. We welcome any additional feedback you may have.
> >
> > We would be happy to continue the discussion or address any further comments. Thank you once again for your valuable contributions to improving our work.
> >
> > Respectfully,
> >
> > Paper 18148 Authors

---

### Meta-Review · Area_Chair_NovX · 2026-01-08

**Summary:**

The paper proposes AgentMath, a tool-augmented agentic framework for competition-level math reasoning. Reviewer iSnP acknowledges that many of the claimed technical innovations are not novel, as they are either standard practices or engineering-focused implementations. Reviewer xGwU considers the novelty is limited as many techniques (async rollout, partial trajectories) already exist in prior RL systems and that lack of code release limits reproducibility.

**Reviewer Concerns:**

Reviewer 5L8W have several concerns: (1) the small, potentially leaked test sets (AIME24/25, HMMT) cast doubt on result reliability; (2) significant training-test data imbalance and unclear overlap filtering raise questions about data contamination; (3) key processes (data synthesis heuristics, answer segmentation, code-use decisions) are unclear, manually intensive, and may not generalize; (4) critical details (e.g., dataset sizes) should be put in the main paper and justification/ablation should be provided for many ad-hoc algorithmic designs.

**Reviewer Scores:**

These are major weaknesses of the original paper, but the author rebuttal addresses these concerns sufficiently. While the proposed method may be mostly "engineering," it is noteworthy that AgentMath achieves significant and consistent improvements, and the method is scalable and efficient. It is also commendable that the authors have now released the code, and compared to Retool, a similar approach brought up by the reviewers, this paper presents clear and reproducible techniques. The new experiments using IMO data, as well as older models, are also helpful in understanding the effectiveness of AgentMath.

---

### Decision · Program_Chairs · 2026-01-26

Accept (Poster)